# *A Winning Hand*: Compressing Deep Networks Can Improve Out-Of-Distribution Robustness

**James Diffenderfer, Brian R. Bartoldson,**\* **Shreya Chaganti,**\* **Jize Zhang, Bhavya Kailkhura**
Lawrence Livermore National Laboratory
{diffenderfer2, bartoldson, chaganti1, zhang64, kailkhura1}@llnl.gov

## Abstract

Successful adoption of deep learning (DL) in the wild requires models to be: (1) compact, (2) accurate, and (3) robust to distributional shifts. Unfortunately, efforts towards simultaneously meeting these requirements have mostly been unsuccessful. This raises an important question: *"Is the inability to create Compact, Accurate, and Robust Deep neural networks (`CARDs`) fundamental?"* To answer this question, we perform a large-scale analysis of popular model compression techniques which uncovers several intriguing patterns. Notably, in contrast to traditional pruning approaches (e.g., fine tuning and gradual magnitude pruning), we find that "lottery ticket-style" approaches can surprisingly be used to produce `CARDs`, including binary-weight `CARDs`. Specifically, we are able to create extremely compact `CARDs` that, compared to their larger counterparts, have similar test accuracy and matching (or better) robustness—simply by pruning and (optionally) quantizing. Leveraging the compactness of `CARDs`, we develop a simple domain-adaptive test-time ensembling approach (`CARD-Deck`) that uses a gating module to dynamically select appropriate `CARDs` from the `CARD-Deck` based on their spectral-similarity with test samples. The proposed approach builds a "winning hand" of `CARDs` that establishes a new state-of-the-art [8] on CIFAR-10-C accuracies (i.e., *96.8% standard and 92.75% robust*) and CIFAR-100-C accuracies (i.e., *80.6% standard and 71.3% robust*) with better memory usage than non-compressed baselines (pretrained `CARDs` available at [8]). Finally, we provide theoretical support for our empirical findings.

## 1 Introduction

Deep Neural Networks (DNNs) have achieved unprecedented success in a wide range of applications due to their remarkably high accuracy [15]. However, this high performance stems from significant growth in DNN model size; i.e., massive overparameterization. Furthermore, these highly overparameterized models are known to be susceptible to the out-of-distribution (OOD) shifts encountered during their deployment in the wild [5]. This resource-inefficiency and OOD brittleness of state-of-the-art (SOTA) DNNs severely limits the potential applications DL can make an impact on.

For example, consider the "Mars rover mission" that uses laser-induced breakdown spectroscopy (LIBS) to search for microbial life. It is well accepted that endowing the rover with DNNs to analyze complex LIBS spectra could produce scientific breakthroughs [3]. However, employing DNNs in such circumstances is challenging: 1) as these devices are battery operated, the model has to be lightweight so it consumes less memory with reduced power consumption, and 2) the model must be able to efficiently handle domain shifts in spectra caused by environmental noise. These requirements are not specific only to the aforementioned use case but arise in any resource-limited application using DL in the wild. The fact that SOTA DNNs do not satisfy *compactness* and *OOD robustness* requirements is holding us back from leveraging advances in DL to make scientific discoveries.

---

\*equal contribution

35th Conference on Neural Information Processing Systems (NeurIPS 2021).

This work is driven by two questions around this crucial problem. **Q1.** *Can we show the existence of compact, accurate, and robust DNNs (`CARDs`)?* **Q2.** If yes, *can we apply existing robustness improvement techniques to `CARDs` to further amplify performance while maintaining compactness?*

Notably, there have been some recent successes in addressing each of the challenges `CARDs` present in isolation. The authors in [24, 27] developed data augmentation methods for achieving high OOD robustness without sacrificing accuracy on the clean data. The authors in [12, 9] developed pruning and quantization approaches for achieving high accuracy at extreme levels of model compression. However, efforts towards achieving model compactness, high accuracy, and OOD robustness simultaneously have mostly been unsuccessful. For example, [26] and [32] recently showed that compressed DNNs achieve accuracies similar to the original networks' but are far more brittle when faced with OOD data. Perhaps unsurprisingly, the current solution in the robust DL community to improve the OOD robustness (and accuracy) is to increase the model size (e.g., [25, 16, 8]).

In this paper, we demonstrate that these negative results are a byproduct of inapt compression strategies, and the inability to create `CARDs` is not fundamental (answering **Q1** in the affirmative). Specifically, we perform a large-scale comparison by varying architectures, training methods, and pruning rates for a range of compression techniques. We find that in contrast to traditional pruning methods (e.g., fine tuning [18] and gradual magnitude pruning [60]), "lottery ticket-style" compression approaches [12, 43, 41, 9] can surprisingly be used to create `CARDs`. In other words, we are able to create extremely compact (i.e., sparse and, optionally, binary) `CARDs` that are significantly more robust compared to their larger and full-precision counterparts while having comparable test accuracy. Our results are in sharp contrast to the existing observation that compression is harmful to OOD robustness. In fact, we show that compression is capable of providing improved robustness.

We subsequently explore the possibility of using existing robustness-improvement techniques in conjunction with compression strategies to further improve the performance of `CARDs`. Empirically, we show the compatibility of `CARDs` with popular existing strategies, such as data augmentation and model size increase. We also propose a new robustness-improvement strategy that leverages the compactness of `CARDs` via ensembling—this ensembling approach is referred to as a domain-adaptive `CARD-Deck` and uses a gating module to dynamically choose appropriate `CARDs` for each test sample such that the spectral-similarity of the chosen `CARDs` and test data is maximized. This proposed adaptive ensembling approach builds a "winning hand" of `CARDs` that establishes a new state-of-the-art robustness (and accuracy) on the popular OOD benchmark datasets CIFAR-10-C and CIFAR-100-C with a compact ensemble [8] (answering **Q2** in the affirmative).

Broad implications of our findings are as follows. First, there exist sparse networks early in training (sometimes at initialization) that can be trained to become `CARDs` (i.e., we extend the lottery ticket hypothesis [12] to robust neural networks via our `CARD` hypothesis). Second, within a random-weight neural network, there exist `CARDs` which (despite having untrained weights) perform comparably to more computationally expensive DNNs (i.e., we extend the *strong* lottery ticket hypothesis [41] to robust neural networks via our `CARD` hypothesis). Third, compression can be complementary with existing robustness-improving strategies, which suggests that appropriate compression approaches should be considered whenever training models that may be deployed in the wild.

To summarize our main contributions:

- Contrary to most prior results in the literature, we show that compression can improve robustness, providing evidence via extensive experiments on benchmark datasets, supporting our `CARD` hypothesis. Our experiments suggest "lottery ticket-style" pruning methods and a sufficiently overparameterized model are key factors for producing `CARDs` (Section 2.2).

- As corruptions in benchmark datasets could be limited (or biased) to certain frequency ranges, we tested the ability of compression to improve robustness to Fourier basis perturbations. This analysis corroborates findings on CIFAR-10/100-C and further highlights that models compressed via different methods have different robustness levels (Section 2.3).

- Leveraging the compactness of `CARDs`, we develop a test-time adaptive ensembling method, called a domain-adaptive `CARD-Deck`, that utilizes `CARDs` trained with existing techniques for improving OOD robustness. Resulting models set a new SOTA performance [8] on CIFAR-10-C and CIFAR-100-C while maintaining compactness (Section 3).

- Finally, we provide theoretical support for the `CARD` hypothesis and the robustness of the domain-adaptive `CARD-Deck` ensembling method (Section 4).

## 2 Is the inability to create CARDs fundamental?

Recent studies of the effects of model compression on OOD robustness have been mostly negative. For instance, Hooker et al. [26] showed that gradual magnitude pruning [60] of a ResNet-50 [19] trained on ImageNet [44] caused accuracy to decrease by as much as 40% on corrupted images from ImageNet-C [23], while performance on the non-corrupted validation images remained strong. Liebenwein et al. [32] reported similar findings for different pruning approaches [43, 2] applied to a ResNet-20 model [19] tested on an analogously corrupted dataset, CIFAR-10-C. Consistent with these findings, Hendrycks et al. [25] found that increasing model size tended to improve robustness.[2]

Critically, these studies suggest that model compression may be at odds with the simultaneous achievement of high accuracy and OOD (natural corruption) robustness. However, it's possible that these negative results are a byproduct of inapt compression strategies and/or insufficient overparameterization of the network targeted for compression. As such, to motivate the scientific question of interest and our empirical/theoretical analyses, we propose the following alternative hypothesis.

> **CARD Hypothesis.** *Given a sufficiently overparameterized neural network, a suitable model compression (i.e., pruning and binarization) scheme can yield a compact network with comparable (or higher) accuracy and robustness than the same network when it is trained without compression.*

### 2.1 Model compression approaches

For a comprehensive analysis of existing pruning methods, we introduce a framework inspired by those in [43, 51] that covers traditional-through-emerging pruning methodologies. Broadly, this framework places a pruning method into one of three categories: (a) traditional, (b) rewinding-based lottery ticket, and (c) initialization-based (strong) lottery ticket. Specific pruning methods considered in these respective categories are: (a) fine-tuning and gradual magnitude pruning, (b) weight rewinding and learning rate rewinding, and (c) edgepopup and biprop. Precise definitions of these pruning methods and discussion of differences are available in Appendix A.1.

Briefly, fine-tuning (FT) [18] prunes models once at the end of normal training, then fine-tunes the models for a given number of epochs to recover accuracy lost due to pruning; while gradual magnitude pruning (GMP) [60] prunes models throughout training. Weight rewinding (LTH) [12, 13] is iterative like GMP but fully trains the network, prunes, rewinds the unpruned weights (and learning rate schedule) to their values early in training, then fully trains the subnetwork before pruning again; learning rate rewinding (LRR) [43] is identical to LTH, except only the learning rate schedule is rewound, not the unpruned weights. Finally, edgepopup (EP) [41] does not weight-train the network but instead prunes a randomly initialized network, using training data to find weights whose removal improves accuracy (notably, EP can, and does here, operate on signed initialization); biprop (BP) [9] proceeds similarly but incorporates a binarization scheme resulting in a binary-weight network regardless of the initialization used. For all of these methods, we make use of global unstructured pruning, which allows for different pruning percentages at each layer of the network. For BP and EP, we additionally consider layerwise pruning, which prunes the same percentage across all layers. We use the hyperparameters specifically tuned for each approach; see Appendix B for additional details.

### 2.2 Accuracy-robustness comparison of global pruning methods

To test the CARD hypothesis, we use: five models (VGG [12, 46] and ResNet [19] style architectures of varying size), five sparsity levels (50%, 60%, 80%, 90%, 95%), and six model compression methods (FT, GMP, LTH, LRR, EP, BP). For each model, sparsity level, and compression method, five realizations are trained on the CIFAR-10 training set [28]. Model accuracy and robustness are measured using top-1 accuracy on the CIFAR-10 and CIFAR-10-C test sets, respectively. CIFAR-10-C contains 15 different common corruptions from four categories: noise, blur, weather, and digital corruptions [23]. As a baseline, we train 5 realizations of each model without compression.

In Figure 1, we plot our experimental results. Accuracy and robustness values are averaged over the five realizations and plotted relative to the average non-compressed baseline performance. The y-axis measures relative difference in percentage points. The mean baseline accuracy and robustness for each architecture is listed as the reference accuracy in each plot. The first row of plots indicate

---

[2]For additional discussion of this and more related work, please see Appendix A.

accuracy (top-1 accuracy on CIFAR-10 relative to baseline) while the second row indicate robustness (top-1 accuracy on CIFAR-10-C relative to baseline). At each sparsity level, error bars extend to the minimum and maximum relative percentage point difference across all realizations.

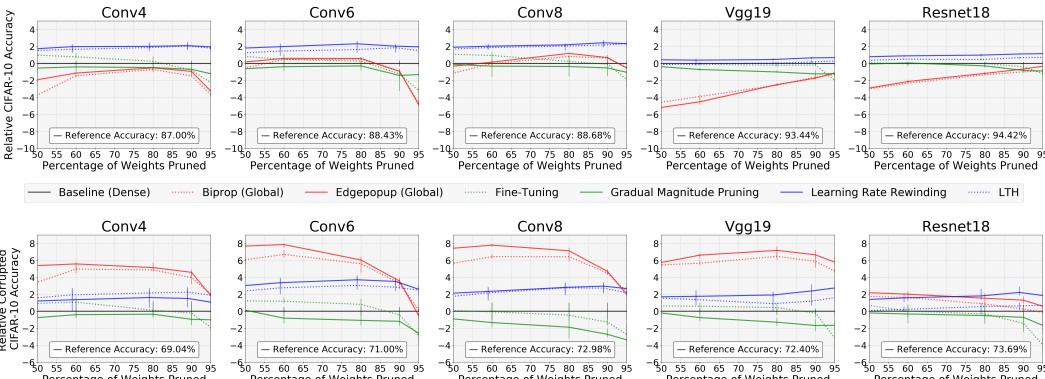

Figure 1: **Suitable pruning approaches can improve robustness over dense models**: Comparing the Top-1 accuracy of pruned models relative to the average of dense baseline models on CIFAR-10 and CIFAR-10-C demonstrates that `CARDs` exist and can be produced using LRR, LTH, BP, or EP.

Our results for traditional methods, i.e., Fine-Tuning and Gradual Magnitude Pruning, are consistent with previous works [26, 32] as the robustness of models pruned using these methods degrades relative to the dense models', particularly in higher pruning regimes. However, we find that rewinding and initialization based pruning approaches consistently produce notable gains in robustness relative to dense baselines while matching (and sometimes surpassing) the accuracy of the dense baseline. In particular, the rewinding class of methods provide a consistent, moderate improvement to both accuracy and robustness while the initialization class provides more substantial gains in robustness even when the accuracy is slightly below the baseline accuracy. The significance of overparameterization to finding highly compact `CARDs` using initialization methods is evident for all architecture types, as the robustness of these models in higher pruning regimes improves at increasing levels of parameterization for a given architecture class. However, even in models with fewer parameters, we find that initialization methods are able to provide notable robustness gains.

Additional experiments involving initialization methods are provided in Appendix C. Specifically, a comparison of the performance EP and BP using layerwise and global pruning is performed in Section C.1 and a comparison of full-precision and binary-weight EP models in Section C.2. Empirical results in Section C.2 indicate that robustness gains provided by EP- and BP-pruned models may be a feature of initialization pruning methods and not solely due to weight-binarization.

### 2.3 Viewing the effect of compression on OOD robustness through a spectral lens

As CIFAR-10-C corruptions are limited to certain frequency ranges and combinations [55], it is of interest to validate if the robustness effects of different pruning methods observed in Section 2.2 hold on a broader ranges of frequencies. To this end, we perform a frequency-domain analysis by utilizing the Fourier sensitivity method [55], which we briefly summarize below.

Given a model and a test dataset, each image in the test dataset is perturbed using additive noise in the form of 2D Fourier basis matrices, denoted by $U_{i,j} \in \mathbb{R}^{d1 \times d2}$. Specifically, for an image $X$ and a 2D Fourier basis matrix $U_{i,j}$ a perturbed image is computed by $X_{i,j} = X + r\varepsilon U_{i,j}$, where $r$ is chosen uniformly at random from $\{-1, 1\}$ and $\varepsilon > 0$ is used to scale the norm of the perturbation. Note that each channel of the image is perturbed independently. Given a set of test images, each Fourier basis matrix can be used to generate a perturbed test set of images on which the test error for the model is measured. Plotting the error rates as a function of frequencies $(i, j)$ yields the Fourier error heatmap of a model – a visualization of the sensitivity of a model to different frequency perturbations in the Fourier domain. Informally, the center of the heat map contains perturbations corresponding to the lowest frequencies and the edges correspond to the highest frequencies.

We generate heatmaps for models corresponding to each pruning method as well as layerwise pruned models using EP and BP. The norm of the perturbation, $\varepsilon$, is varied over the set $\{3, 4, 6\}$ to represent

low, medium, and high levels of perturbation severity. As a reference, we include heatmaps for the dense (non-compressed) baseline model. Fourier heatmaps for the Conv8 architecture at 80% prune percentage are provided in Figure 2 while additional heatmaps can be found in Section D.

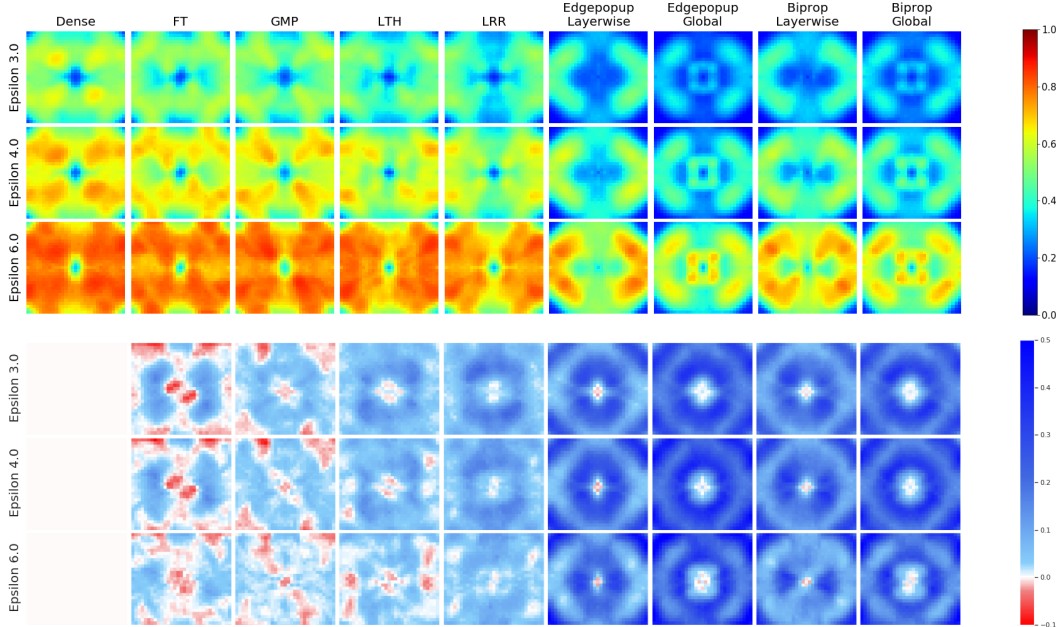

Figure 2: **Visualizing the response of compressed models to perturbations at different frequencies**: The top three rows are Fourier heatmaps for Conv8 trained on CIFAR-10 with 80% of weights pruned. The bottom three rows are difference to the baseline with blue regions indicating lower error rate than baseline. Init. methods provide up to a 50 percentage point improvement in some instances.

Figure 2 illustrates that initialization pruning methods reduce the error rate across nearly the full spectrum of Fourier perturbations relative to the dense model. Additionally, initialization pruning methods using layerwise pruning present a different response, or error rate, at certain frequency corruptions when compared to heatmaps of global initialization pruning methods. The difference heatmaps show that rewinding methods offer mild to moderate improvements across much of the frequency spectrum with LRR outperforming LTH in a few regions of the heatmap. The difference heatmaps also highlight that traditional methods degrade the robustness to more Fourier perturbations than other compression methods and result in an increased error rate of 10 percentage points (relative to the dense baseline) in some cases. These findings further suggest that the robustness of a compressed model is dependent on the compression method used or the resulting structure of the sparsity. In Appendix E, we provide additional heatmaps to examine the impact on robustness when varying the number of rewinding steps used by rewinding methods.

To summarize, we have empirically verified our `CARD` hypothesis by demonstrating that "lottery ticket-style" compression methods can produce compact models with accuracy and robustness comparable to (or higher than) their dense counterparts.

## 3    Creating a winning hand of CARDs

Having demonstrated that certain model compression techniques are capable of producing `CARDs`, we explore using existing techniques for improving model robustness in conjunction with compression strategies to produce `CARDs` that further improve robustness. We consider three popular existing strategies for improving model robustness and, further, propose a test-time adaptive ensembling strategy, called a domain-adaptive `CARD-Deck`, that leverages these strategies to maintain compactness and efficiency while improving accuracy and robustness over individual `CARDs`.

## 3.1 Popular strategies for improving model robustness

**Data augmentation.**    A popular approach for improving robustness involves data augmentations. We consider two augmentation techniques that are (at the time of writing) leading methods on Robust-Bench [8]. The first is AugMix [24] which can provide improved robustness without compromising accuracy by randomly sampling different augmentations, applying them to a training image, then "mixing" the resulting augmented image with the original. The second method independently adds Gaussian noise $\mathcal{N}(\mu = 0, \sigma = 0.1)$ to all the pixels with probability $p = 0.5$ [27].

**Larger models.**    Another popular strategy for improving OOD robustness (and accuracy) in the robust DL community is to increase the model size (e.g., [25, 16, 8]). Hence, we also consider this strategy to investigate if the performance of CARDs can be amplified by compressing larger models.

**Model Ensembling.**    It is natural to consider exploiting CARD compactness to amplify accuracy and robustness by ensembling [40] CARDs. For example, an ensemble of two to six CARDs pruned to 95% sparsity only uses 10% to 30% of the parameter count required by a single dense model.

## 3.2 Playing the right CARD to improve accuracy-robustness performance

Ensembling CARDs trained with state-of-the-art data augmentation techniques has the potential to provide additional robustness gains. We call such ensembles CARD-Decks and propose two strategies: (1) domain-agnostic CARD-Decks and (2) domain-adaptive CARD-Decks. In both strategies, the ensemble consists of CARDs that have been trained on the same dataset under different augmentation schemes. The domain-adaptive CARD-Deck utilizes a spectral-similarity metric to select a subset of CARDs from the CARD-Deck that should be used to make predictions based on the current test data. We first define this metric then provide formal definitions for both CARD-Deck methods.

**A spectral-similarity metric.**    Let $x_{train} \in \mathbb{R}^{D_1 \times D_2 \times N}$ denote the N unaugmented training images of dimension $D_1 \times D_2$, $A = \{a_k\}_{k=1}^m$ denote a set of $m$ different augmentation schemes, and $\hat{S}_{a,P}(x)$ denote a sampling of $P$ images from $x$ where augmentation $a \in A$ has been applied to $x$. Motivated by our analysis using Fourier heatmaps, we propose a spectral-similarity metric to compare representatives from augmented versions of the training sets, $\{\hat{S}_{a,P}(x_{train})\}_{a \in A}$, to the test data. First, we define $F(\cdot)$ as a map that computes the 1D radially-averaged power spectrum for images of dimension $D_1 \times D_2$ then takes the reciprocal of each component. Our spectral-similarity metric is a map $d_{ss} : \mathbb{R}^{D_1 \times D_2 \times P} \times \mathbb{R}^{D_1 \times D_2 \times M} \to \mathbb{R}$ defined by $d_{ss}(\boldsymbol{X}, \boldsymbol{Y}) = \min_{1 \leq i \leq P} \|(F(X_i)/\|F(X_i)\|) - \frac{1}{M} \sum_{j=1}^{M} (F(Y_j)/\|F(Y_j)\|)\|$. In practice, we found that the 1D power spectra for different augmentation types were more separable in the higher frequencies of the power spectrum leading to the use of the reciprocal in the definition of $F(\cdot)$.

**A "winning hand" of CARDs by test-time ensembling.**    An $n$-CARD-Deck ensemble is composed of $n$ CARDs given by $f^{Deck} = \{f^{a_k}\}_{k=1}^n$ where $a_k$ is one of the $m$ augmentation schemes from $A$ and the superscript in $f^{a_k}$ denotes that this CARD was trained used data from the distribution $S_{a_k}(x_{train})$. Our domain-agnostic $n$-CARD-Deck averages the prediction of all $n$ CARDs in the deck. Supposing that the output of each CARD in $f^{Deck}$ is softmax vectors, then the output of the domain-agnostic $n$-CARD-Deck can be expressed as $\frac{1}{n} \sum_{k=1}^{n} f^{a_k}(x_{test})$. In our domain-adaptive CARD-Deck, a gating module uses the spectral-similarity metric $d_{ss}$ to determine which augmentation method is most similar to a batch of $M$ test images $x_{test} \in \mathbb{R}^{D_1 \times D_2 \times M}$ provided to the ensemble. When an augmentation scheme, say $a \in A$, is identified as the most similar to the incoming test data, the domain-adaptive CARD-Deck utilizes only the CARDs that were trained using the data from the distribution $S_a(x_{train})$. The set of the most similar augmentations is given by $a^* \in \arg\min_{a \in A} d_{ss}(\hat{S}_{a,P}(x_{train}), x_{test})$. We note that $a^*$ is likely to be a singleton set indicating that a single augmentation scheme is most similar. If the domain-adaptive CARD-Deck contains multiple CARDs trained using the same data augmentation scheme, prediction averaging is used on these CARDs and returned as the CARD-Deck prediction. Given $a^*$ and letting $\mathcal{I}(a) = \{k : a_k \in a\}$, the output of the domain-adaptive CARD-Deck can be expressed as $\frac{1}{|\mathcal{I}(a^*)|} \sum_{k \in \mathcal{I}(a^*)} f^{a_k}(x_{test})$. As computing the spectral-similarity scheme is independent of CARD evaluation, the domain-adaptive CARD-Deck provides reduced inference time over the domain-agnostic CARD-Deck by only evaluating the CARDs necessary for prediction. Figure 3 provides an illustration of the CARD-Deck design.

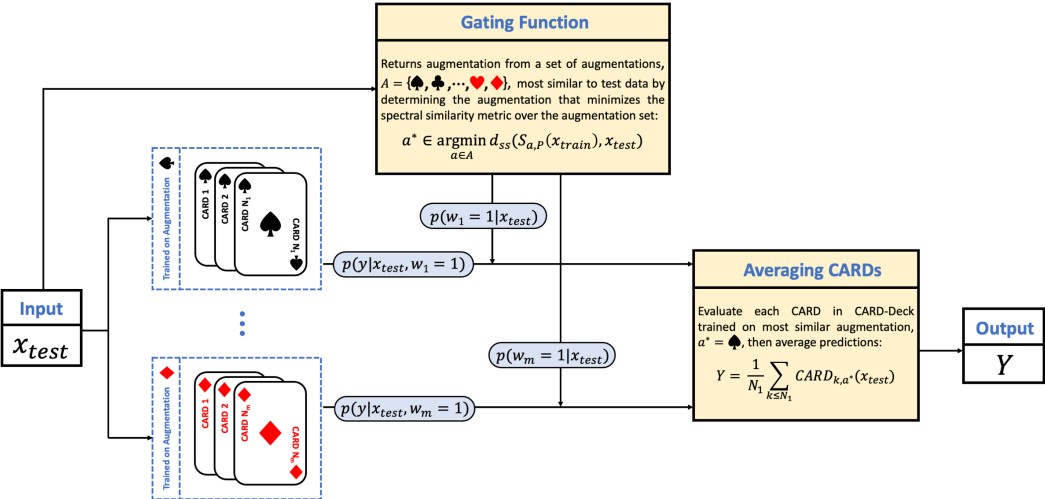

Figure 3: **Selecting a "winning hand" using a domain-adaptive** `CARD-Deck`: `CARDs` are grouped based on the data augmentation scheme used during training. At test time, a gating function identifies the augmentation scheme, $a^*$ with data most similar to the test data. The `CARDs` trained using $a^*$ are used and the average prediction is returned as the output.

## 3.3 Experimental results

We experiment with four models of increasing size (ResNeXt-29, ResNet-18, ResNet-50, WideResNet-18-2), three data augmentation methods (clean, AugMix, Gaussian), two sparsity levels (90%, 95%), and six compression methods (LTH, LRR, EP (layerwise and global), BP (layerwise and global)). For each model, sparsity level, data augmentaion method, and compression method, three realizations are trained on CIFAR-10 [28] and robustness is measured using CIFAR-10-C. Model compactness is measured by calculating the memory usage [54]. Similar experiments are performed for CIFAR-100 and CIFAR-100-C, however only WideResNet-18-2 and four model compression methods (LTH, LRR, EP (global), BP (global)) are used. As a baseline, three realizations of each model are trained without compression for each data augmentation method. Visualizations of key results are provided in this section and detailed ablation studies are in Appendix F.

In addition to measuring the performance of `CARDs` for each configuration (i.e. model, data augmentation, compression method, sparsity level), we also formed domain-agnostic and domain-adaptive $n$-`CARD-Decks` of size $n \in \{2, 4, 6\}$ comprised of models using the same compression method and sparsity level. For each $n$-`CARD-Deck`, half of the `CARDs` were trained using AugMix and the other half were trained using the Gaussian augmentation. To facilitate computation of the spectral-similarity metric in domain-adaptive `CARD-Decks`, for each augmentation method $a \in \{AugMix, Gaussian\}$ we statically created a KD-Tree containing $F(X)/\|F(X)\|$, for all $X \in \hat{S}_{a,P}(x_{train})$. In our experiments, we took $P = 5000$ and these KD-Trees were generated once and saved (separate from inference process). At test time, batches of $M = 100$ test images were used in the spectral-similarity metric to determine which augmentation method best represented the corrupted test data.

**Test-time ensembling can provide a "winning hand".** Figure 4 provides a visualization of the performance (accuracy, robustness, and memory usage) of several `CARDs` and `CARD-Decks` as well as dense baselines and the previous SOTA model. This figure highlights our findings that both `CARD-Deck` methods, domain-agnostic and adaptive, are capable of improving the performance beyond the dense baselines while maintaining reduced memory usage. Notably, we found a single LRR `CARD` (a WideResNet-18 at 96% sparsity) trained with AugMix can attain 91.24% CIFAR-10-C accuracy, outperforming dense ResNeXt-29 trained with AugMix (a state-of-the-art among methods that do not require non-CIFAR-10 training data) by more than 2 percentage points simply by pruning a larger model, i.e., WideResNet-18. Our best performing 6-`CARD-Deck` using LRR WideResNet-18 models (53.58 MB) sets a new state-of-the-art for CIFAR-10 and CIFAR-10-C accuracies of 96.8% and 92.75%, respectively. In contrast, the previous best method [6] achieves accuracies (94.93%,

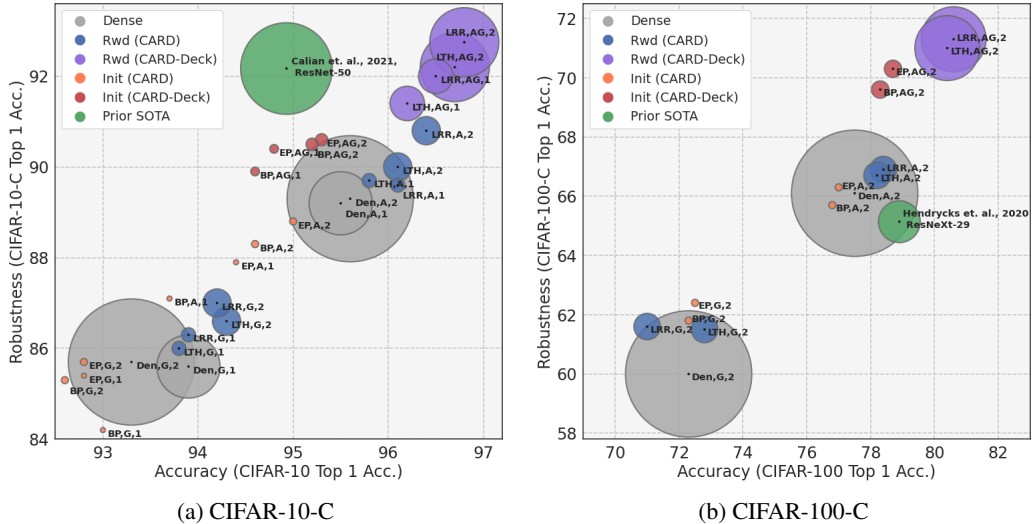

(a) CIFAR-10-C          (b) CIFAR-100-C

Figure 4: **Accuracy, robustness, and memory trends**. `CARDs` and `CARD-Decks` reduce memory usage (as indicated by the area of each circle) while achieving comparable or improved accuracy (x-axis) and robustness (y-axis). Annotation indicates {Compression Method, Data Augmentation, ResNet-18 Width} with **A** for AugMix, **G** for Gaussian, and **AG** for both (used by `CARD-Decks`).

92.17%) using increased memory (ResNet-50 with 94.12 MB), extra data (a super resolution network was pre-trained with non-CIFAR-10 data), and a computationally expensive adversarial training procedure. More impressively, our computationally lighter binary-weight `CARD-Decks` provide comparable accuracy and robustness to the dense baseline with memory usage as low as 1.67 MB. Similar trends hold on CIFAR-100-C where rewinding domain-adaptive `CARD-Decks` set a new SOTA performance (80.6%, 71.3%) compared to the previous best (78.90%, 65.14%) [24]. Note that the binary-weight `CARD-Decks` provide almost 5 percentage point robustness gain over the previous best at only 9% of the memory usage. Note that the performance of EP and BP `CARD-Decks` can be further improved by leveraging more computationally expensive training procedures, e.g., tuning batchnorm parameters [9] or searching for EP and BP `CARDs` in pretrained neural nets.

To summarize, `CARD-Decks` can maintain compactness while leveraging additional robustness improvement techniques, LRR `CARD-Decks` set a new SOTA on CIFAR-10-C and CIFAR-100-C in terms of accuracy and robustness, binary-weight `CARD-Decks` can provide up to ∼105x reduction in memory while providing comparable accuracy and robustness, and the domain-adaptive `CARD-Decks` used here are ∼2x faster than the domain-agnostic `CARD-Decks` as only half of the `CARDs` are used at inference. Additionally, for 2-`CARD-Decks` our domain-adaptive method provides an average robustness gain of 1-2 percentage points over the domain-agnostic method (see Appendix F).

## 4 Theoretical justifications

This section provides (1) theoretical results that provide support for the `CARD` hypothesis beyond what we demonstrated empirically and (2) robustness analysis for domain-adaptive `CARD-Deck` strategy.

### 4.1 Function approximation view of CARDs

By leveraging existing theoretical analyses of the Strong and Multi-Prize Lottery Ticket Hypotheses [39, 38, 9], we can provide theoretical support for the `CARD` hypothesis. While we were able to empirically produce `CARDs` within the same architecture used by the dense model, to prove theoretical results supporting the `CARD` hypothesis using existing techniques requires that the compressed network be searched for within an architecture larger than the architecture used for the dense model. An informal version of this result for binary-weight `CARDs` is provided here relevant to models produced by joint pruning and binarization compression strategies such as multi-prize tickets [9].

**Theorem 1.** *Given a non-compressed network $F$ with depth $\ell$ and width $w$ with bounded weights that achieves a desired target accuracy and robustness, a random binary network of depth $2\ell$ and width $O\left((\ell w^{3/2}/\varepsilon) + \ell w \log(\ell w/\delta)\right)$ contains with probability $(1-\delta)$ a binary-weighted `CARD` that approximates the target non-compressed network with error at most $\varepsilon$, for any $\varepsilon, \delta > 0$.*

We note that Theorem 1 follows immediately from Theorem 2 in [9] and, thereby, refer the reader to Theorem 2 in [9] for a formal statement.[3] This result provides a level of confidence with which one can expect to find a binary-weight `CARD` that is an $\varepsilon$-approximation of a target (i.e. trained and non-compressed) network. For full-precision weight `CARD`s, tighter bounds on the depth and width of a fully-connected network with ReLU activations containing a `CARD` that is an $\varepsilon$-approximation of a target network follow from Theorem 3 in [38] which also utilizes a more relaxed hypothesis set. Hence, theoretical results supporting the existence of both full-precision and binary-weight `CARD`s, with high probability, provided that a sufficiently overparameterized network is used.

Leveraging these theoretical results, we provide a corollary on the approximation capabilities of `CARD-Deck`. We denote by $F(\ell, \boldsymbol{w})$ a fully-connected neural network with ReLU activations where $\ell$ denotes the depth of the network and $\boldsymbol{w} = [w_0, w_1, \ldots, w_\ell] \in \mathbb{N}^{\ell+1}$ is a vector where component $i \in \{1, \ldots, \ell\}$ denotes the width of layer $i$ in $F$ and $w_0$ denotes the input dimension of $F$.

**Corollary 2** (`CARD-Deck` Approximation Theorem)**.** *Let $\varepsilon > 0$, $\delta > 0$, $n \geq 1$, and $\boldsymbol{\lambda} = [\lambda_1, \ldots, \lambda_n]$ satisfying $\sum_{k=1}^{n} \lambda_k = 1$ and $\lambda_k \geq 0$, for all $k \in [n]$, be given. Let $\mathcal{F} = \{F_k(\ell_k, \boldsymbol{w}_k)\}_{k=1}^{n}$ be a deck of non-compressed fully-connected networks with ReLU activations. If the input space $\mathcal{X}$ and each network in the collection $\mathcal{F}$ satisfies the hypotheses of Theorem 3 in [38] (Theorem 2 in [9]), then with probability $(1 - \delta)^n$ there exists a deck of $n$ full-precision (binary-weight) `CARD`s denoted $f^{Deck} = \{f_k\}_{k=1}^{n}$ of depth and width specified by Theorem 3 in [38] (Theorem 2 in [9]) such that*

$$\sup_{x \in \mathcal{X}} \left\| \sum_{k=1}^{n} \lambda_k f_k(x) - \sum_{k=1}^{n} \lambda_k F_k(\ell, \boldsymbol{w})(x) \right\| \leq \varepsilon. \tag{1}$$

A proof of Corollary 2 is provided in Appendix G. Note that the target non-compressed networks in Corollary 2 could be trained on data sampled from augmented distributions, such as augmented distributions using the AugMix and Gaussian methods, provided that the weights of the resulting networks satisfy the hypothesis required from the existing results in [38, 9]. Additionally, the appropriate choice of $\boldsymbol{\lambda}$ in Corollary 2 can yield a domain-agnostic or domain-adaptive `CARD-Deck`.

## 4.2 Robustness analysis of CARD-Deck

To provide the theoretical justification behind our `CARD-Deck` approach over a single classifier, we first define a robustness measure for a given classifier ensemble trained on a set of augmentations w.r.t. a corruption set encountered at the test-time. We assume that each test sample may encounter a specific corruption type $c$ from a given OOD set and be transformed to a corrupted test sample $x_c$. Let us assume $f^a$ is learnt using a learning algorithm $L$ using the augmented training data $S_a$ sampled from distribution $\mathcal{D}_a$, thus, we have $f^{Deck} = \{f^a = L(S_a) | a \in \mathbb{N}^A\}$ where $\mathbb{N}^A = \{1, \cdots, A\}$. Let us denote by $\hat{S}_a$ an empirical distribution w.r.t. sampled dataset $S_a$.

**Definition 1** (Average OOD Robustness)**.** *Let $f^{Deck} = \{f^a | a \in \mathbb{N}^A\}$ denote a `CARD-Deck` trained using an augmentation set $S^A = \{S_a \sim \mathcal{D}_a | a \in \mathbb{N}^A\}$. We define the average out-of-distribution robustness for a `CARD-Deck` w.r.t. corruption set $\mathcal{D}^C = \{\mathcal{D}_c | c \in \mathbb{N}^C\}$ as*

$$Rob(\mathcal{D}^C, f^{Deck}) = \sum_{a=1}^{|A|} \sum_{c=1}^{|C|} Rob(\mathcal{D}_c, f^a) w_c^a, \tag{2}$$

*where $Rob(\mathcal{D}_c, f^a) = \mathbb{E}_{(x_c, y) \sim \mathcal{D}_c} \left[ \inf_{f^a(x_c') \neq y} d(x_c', x_c) \right]$ with $x_c'$ being a perturbed version of $x_c$, $d$ corresponds to a distance metric, and $w_c^a$ denotes the probability of $f^{Deck}$ gating module selecting the classifier $f^a$ to make a prediction on test data coming from corruption type $c$.*

This definition refers to the expectation of the distance to the closest misclassified corrupted sample for a given test sample. Note that this is a stronger notion of robustness then the generalization error corresponding to a corrupted data distribution. Having this definition, our goal is to provide a lower bound on the average OOD robustness of $f^{Deck}$ and show that the use of domain-adaptive classifier

---

[3]Following the acceptance of this paper, improved bounds on the depth and width have been established [47].

ensemble achieves a better OOD robustness compared to the case where we use just a single classifier $f^a$. To understand this quantity better, we derive the following decomposition (see Appendix G):

$$Rob(\mathcal{D}^C, f^{Deck}) \geq \sum_{a,c} w_c^a [\underbrace{Rob(\hat{S}_a, f^a)}_{(a)} - \underbrace{\|Rob(\mathcal{D}_a, f^a) - Rob(\hat{S}_a, f^a)\|}_{(b)} - \underbrace{\|Rob(\mathcal{D}_c, f^a) - Rob(\mathcal{D}_a, f^a)\|}_{(c)}].$$

This shows that the average OOD robustness can be bounded from below in terms of the following three error terms for a classifier-corruption pair weighted by their selection probabilities: (a) empirical robustness, (b) generalization gap, and (c) out-of-distribution-shift. This implies that in order to bound the average OOD robustness, we need to bound both the generalization gap and the OOD-shift. Next, we provide a bound on the OOD-shift penalty that is independent of the classifier $f^a$ and is only related to the closeness of the augmented data distribution and corrupted data distribution. The closeness is defined in terms of Wasserstein distance $W(\cdot, \cdot)$ (see Definition 2 in Appendix G).

**Theorem 3** (Average OOD-Shift Bound). *For any* `CARD-Deck`*, the average OOD-shift (i.e.,* $ADS = \sum_{a,c} w_c^a \|Rob(\mathcal{D}_c, f^a) - Rob(\mathcal{D}_a, f^a)\|$*) can be bounded as follows* $ADS \leq \sum_{a=1}^{|A|} \sum_{c=1}^{|C|} w_c^a \times W(\mathcal{D}_c, \mathcal{D}_a)$.

*Proof.* This result can be proved by applying Theorem 1 in [45] to ADS. □

**Key insights.** Theorem 3 provides some key insights into the OOD robustness of classifiers trained on augmented datasets. First, unlike the generalization gap, the OOD-shift does not converge to zero with more augmentation data. This imposes a fundamental limit on the OOD robustness in terms of the distance between augmented train data distribution and corrupted test data distribution. Second, having diverse augmentations is critical to improving the OOD robustness. Also, it highlights that existing solutions trained with a single augmentation scheme might just be getting lucky or overfitting to the corrupted test data. Finally, the domain-adaptive `CARD-Deck` with a suitable gating function is provably better than using a single classifier because it can achieve the minimum conditional Wasserstein distance (or best achievable OOD robustness) over given augmentation-corruption pairs.

## 5  Limitations and future directions

In this paper, we showed that model compression and high robustness (and accuracy) are not necessarily conflicting objectives. We found that compression, if done properly (e.g., using "lottery ticket-style" objectives), can improve the OOD robustness compared to a non-compressed model. Leveraging this finding, we proposed a simple domain-adaptive ensemble of `CARD`s that outperformed existing SOTA in terms of the clean accuracy and the OOD robustness (at a fraction of the original memory usage). Our results are consistent with past results in that we also show that the use of test accuracy alone to evaluate the quality/deployability of a compressed model in the wild is not sufficient—one needs to adopt harder metrics such as OOD robustness. However, as opposed to the existing works in this direction, we present a construction that satisfies these "harder" requirements.

There are still many interesting questions that remain to be explored. First, while we were able to produce `CARD`s it remains unclear (i) why only certain pruning strategies were able to produce them and (ii) why introducing compression can improve "effective robustness" [49] (e.g. Conv and VGG19 BP and EP models in Figure 1). Second, the spectral relationship of train and test data (as considered in this work) is not the only interaction determining the performance of a compressed model. It will be worthwhile to take a more holistic approach that also takes spectral behavior of the compressed model (e.g., using intermediate features) into account, which could possibly benefit from using `CARD`s compressed via different strategies when building a "winning hand". Third, we only derived an upper bound on the amount of overparameterization needed to approximate a target dense network in our theoretical analysis; it will also be interesting to explore a lower bound (a necessary condition) on the same which may indicate scenarios where the proposed approach will not work (e.g., underparameterized NNs). Fourth, "lottery ticket-style" models in theory can be found more efficiently, which was not our focus but is a valuable future direction. Finally, achieving the theoretical memory savings obtained from `CARD`s (reported in this paper) would require their implementation on specialized hardware. We hope that our results will help researchers better understand the limits of compressed neural nets, and motivate future work on `CARD`s and their applications to areas where DL struggles currently due to its parameter-inefficiency and OOD brittleness.

## Acknowledgements

We would like to thank the reviewers for their valuable discussion during the rebuttal period that resulted in improved clarity and presentation of our research. This work was performed under the auspices of the U.S. Department of Energy by the Lawrence Livermore National Laboratory under Contract No. DE-AC52-07NA27344 and LLNL LDRD Program Project No. 20-ER-014 (LLNL-CONF-823802).

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
