# Supplementary Material: *A Winning Hand*: Compressing Deep Networks Can Improve Out-Of-Distribution Robustness

Here we provide a brief outline of the appendices. In Appendix A, we provide details on relevant past works. In Appendix B, we discuss our experimental setting and relevant hyperparameters. In Appendix C, we provide additional experiments with initialization methods and, in part, show that the robustness of the EP method is not only due to binarization but also due to the specific pruning strategy. In Appendix D, we provide Fourier heatmaps for additional pruning rates and architectures. In Appendix E, we provide additional Fourier heatmap results on comparing the rewinding-based schemes with the traditional pruning schemes. In Appendix F, we provide extensive tables for `CARD` and `CARD-Deck` experiments performed in Section 3. In Appendix G, we provide remaining proof details for our theoretical justification of our `CARD-Deck` approach. We show the universal approximation power of `CARD-Decks` and prove that `CARD-Deck` with a suitable gating function is provably better than using a single classifier.

## A  Background

### A.1  Accuracy preserving model compression

Two popular approaches for model compression are: pruning and quantization. Here, we discuss these approaches and their effects on accuracy.

**Pruning.** Neural network pruning removes weights [30] or larger structures like filters [31] from neural networks to reduce their computational burden [18, 21] and potentially improve their generalization [53, 35]. As the performance of DNNs has continued to improve with increasing levels of overparameterization [56], production DNNs have grown larger [29, 4], and the need to broadly deploy such models has amplified the importance of compression methods like pruning [18, 17].

In modern networks, pruning the smallest magnitude weights after training then fine-tuning (FT) to recover accuracy lost from the pruning event is surprisingly effective; when the pruning is done iteratively rather than all at once, this approach enables a 9x compression ratio without loss of accuracy [18]. Gradual magnitude pruning (GMP) performs such iterative pruning throughout training rather than after training [37, 60], recovering accuracy lost from pruning events as training proceeds, and matches or exceeds the performance of more complex methods [14].

Another form of magnitude pruning stems from work on the lottery ticket hypothesis (LTH), which posits that the final, sparse subnetwork discovered by training then pruning can be rewound to its state at initialization [12] or early in training [13], then trained in isolation to be comparably accurate to the trained dense network. The associated pruning approach that iteratively trains the network, rewinds the weights (and learning rate schedule) to their values early in training, then trains the subnetwork is referred to here as LTH. A simpler version of this algorithm, learning rate rewinding (LRR) [43], only rewinds the learning rate schedule (not the weights) and achieves a state-of-the-art accuracy-efficiency frontier while being less complex than other competitive approaches [60, 36, 12, 22]. LRR has been shown to offer small improvements to accuracy with not-too-high compression ratio [43]. The authors in [50] proposed calibration mechanisms to find more effective lottery tickets.

Building on the lottery ticket hypothesis, the edgepopup (EP) algorithm introduced a way to find sparse subnetworks at initialization that achieve good performance without any further training [41]. Diffenderfer and Kailkhura [9] introduced a similar pruning approach, biprop (BP), which also performs weight binarization.

**Binarization.** Typical post-training schemes have not been successful in binarizing pretrained models with or without retraining to achieve reasonable accuracy. Most existing post-training works [17, 58] are limited to ternary weight quantization. To overcome this limitation, there have been several efforts to improve the performance of binary neural network (BNN) training. This is challenging due to the discontinuities introduced by the binarization, which makes back-propagation difficult. Binaryconnect [7] first showed how to train networks with binary weights within the familiar back-propagation paradigm. Unfortunately, this early scheme resulted in a significant drop in accuracy compared to its full precision counterparts. To improve performance, XNOR-Net [42] proposed adding a real-valued channel-wise scaling factor to improve capacity. Dorefa-Net [59] extended

XNOR-Net to accelerate the training process via quantized gradients. ABC-Net [33] improved performance by using more weight bases and activation bases at the cost of increased memory.

Notably, one can exploit the complementary nature of pruning and binarization to combine their strengths. For example, Diffenderfer and Kailkhura [9] produced an algorithm for finding multi-prize lottery tickets (MPTs): sparse, binary subnetworks present at initialization that don't require training.

**Pruning algorithm framework.** The following pruning algorithm framework, inspired by those in [43, 51], covers traditional-through-emerging pruning methodologies. Specifically, we define the trained subnetwork created by one pruning-retraining cycle (i.e., one pruning iteration) as:

$$W_{\text{sparse}} = F_1(W_{k_1}; \mathcal{D}) \odot F_2(W_i; \mathcal{D}, \mathcal{M}, k_2), \tag{3}$$

where $\mathcal{D}$ denotes the training dataset, $W_i$ denotes the weight vector at the start of training iteration $i$,[4] $F_1$ represents the function that finds and returns the weight-masking vector $\mathcal{M}$, $F_2$ represents the function that retrains the weights after $\mathcal{M}$ is found, $k_i$ is the earliest training iteration that $F_i$ requires information from (e.g., weight-vector or learning-rate values), $F_1$ and $F_2$ are each applied at the beginning of iteration $k_1$, and $\odot$ is the Hadamard (element-wise) product. Using this, the pruning paradigms and representative techniques from these categories considered in this paper are as follows:

- **Traditional** $k_1 = k_2$ and $F_2 \neq I$ (identity function). Particular techniques:
  - Fine-Tuning (FT) [18]:
    $W_{\text{sparse}} = F_1(W_T) \odot F_2(W_T; \mathcal{D}, \mathcal{M}, T)$
  - Gradual Magnitude Pruning (GMP) [60]:
    $W_{\text{sparse}} = F_1(W_t) \odot F_2(W_t; \mathcal{D}, \mathcal{M}, t)$, where $t \in [t_1, t_2, ..., t_n]$ and $t_n < T$
- **Rewinding-based Lottery Ticket** $k_1 = aT - (a-1)r$ and $k_2 = r$, where $a \in [1 .. n]$ and $r \ll T$. Particular techniques:
  - Weight Rewinding (LTH) [12, 13]:
    $W_{\text{sparse}} = F_1(W_{aT-(a-1)r}) \odot F_2(W_r; \mathcal{D}, \mathcal{M}, r)$
  - Learning Rate Rewinding (LRR) [43]:
    $W_{\text{sparse}} = F_1(W_{aT-(a-1)r}) \odot F_2(W_{aT-(a-1)r}; \mathcal{D}, \mathcal{M}, r)$
- **Initialization-based (Strong) Lottery Ticket** $k_1 = k_2 = 0$ and $F_2 = I$. Particular techniques:
  - Edgepopup (EP) [41]:
    $W_{\text{sparse}} = F_1(W_{0, \text{binary}}; \mathcal{D}) \odot I(W_{0, \text{binary}})$
  - Biprop (BP) [9]:
    $W_{\text{sparse}} = F_1(W_0; \mathcal{D}) \odot I(W_{0, \text{binarized by biprop}})$

Note that GMP, LTH, and LRR are all iterative. Further, since rewinding schemes apply $F_1$ and $F_2$ at the beginning of iterations $k_1 = aT - (a-1)r$, $a \in [1 .. n]$, it's true that $k_1 > k_2 = r$, so $F_2$ needs to store information from iteration $k_2 = r$ in order to (at $k_1$) perform the training iterations that determine $W_i$, $i \geq T$. As opposed to traditional and rewinding schemes, strong lottery ticket [41] schemes do not require any weight training before or after pruning—a performant network is found at initialization via $F_1$. In other words, learning occurs simply by pruning a randomly initialized neural network. Furthermore, by design BP performs binarization of the weights to reduce the memory footprint. We note that the precision of the weights in networks trained using EP maintain the same precision as the randomly initialized weights. Hence, EP can also be used to identify binarized networks by randomly initializing the weights to binary values. To take advantage of additional compression, in our experiments with EP the mask $\mathcal{M}$ is learned from a binary-initialized weight vector $W_{0, \text{binary}}$. As BP performs binarization during pruning, a full-precision weight vector $W_0$ is used when finding $\mathcal{M}$. In all of these methods, we make use of global unstructured pruning which allows for different pruning percentages at each layer of the network.

---

[4]During training, $i < T$ for most pruning approaches, where $T$ is the default number of training iterations. However, fine-tuning trains for an additional set of iterations after pruning takes place at iteration $T$. Additionally, rewinding-based lottery ticket approaches (when accounting for training done by $F_2$) use $(n+1)T - nr$ training iterations, where $n$ is the number of pruning iterations or "shots" in an n-shot pruning procedure, and $r$ is the iteration rewound to after each pruning iteration (note that when $r = 0$, the network is rewound to its state from initialization after each pruning iteration and $(n + 1)T$ total iterations are required by this approach).

## A.2 Accuracy preserving robust training

While DNN models show impressive generalization in I.I.D. data scenarios [48, 10], the robustness of such models on OOD data (e.g., common corruptions – blurring from camera movement, or noise from low-lighting conditions) is critical to the successful deployment of DL in the wild. To evaluate performance in the presence of such common corruptions, Hendrycks and Dietterich [23] introduced the CIFAR-10-C dataset, which comprises validation images from CIFAR-10 [28] that were exposed to 15 diverse corruption types applied at 5 severity levels.

To achieve high OOD robustness and accuracy, AugMix [24] creates data augmentations at training time by composing randomly-selected augmentation operations from a diverse set, which notably excludes augmentations overlapping with those used to create CIFAR-10-C. Additionally, AugMix utilizes a Jensen-Shannon Divergence consistency loss term to match the predictions between different augmentations of a given image. This approach is expanded on by DeepAugment [25], which inputs clean images to a pretrained image-to-image model, corrupts this model's weights and activations with various operations that distort the typical forward pass, then uses the output images as augmented data. AdversarialAugment (AdA) builds on DeepAugment by generating the weight perturbations performed on the image-to-image models via adversarial training [6]. Also, when used with an appropriately selected perturbation radius and distance metric, adversarial training can serve as a strong baseline against common corruptions [16, 27].

Notably, the state-of-the-art in OOD robustness has historically evolved by leveraging more advanced data augmentation schemes and larger models than prior works [8].

## A.3 Methods to design compact-accurate-robust models

Despite its critical need, efforts towards achieving model compactness, high accuracy, and OOD (natural corruption) robustness simultaneously have mostly been unsuccessful, to the best of our knowledge. Note that some recent works have shown successful attempts for different use cases, e.g., adversarial example robustness [52], additive white noise robustness [1], and domain generalization [57].

Hooker et al. [26] analyzed traditional compression techniques [60] and showed that pruned and quantized models have comparable accuracy to the original dense network *but* are far more brittle than non-compressed models in response to small distributional changes that humans are robust to. It is well known that even non-compressed models are very brittle to the OOD shifts. The authors in [26] showed that this brittleness is amplified at higher levels of compression.

Liebenwein et al. [32] corroborated that a pruned [43, 2] model can have similar predictive power to the original one when it comes to test accuracy, while being more brittle when faced with out of distribution data points. They further showed that this phenomenon holds even when considering robust training objectives (e.g., data augmentation). Their results suggest that robustness advances discussed in Sec. A.2 may be suboptimal with model compression approaches unless OOD shifts are known at train time.

Notably, the aforementioned papers only analyze a limited class of pruning approaches. Our findings with traditional pruning approaches are consistent with the findings of [26], which involved a traditional pruning approach. Additionally, when Liebenwein et al. [32] employ a lottery ticket-style pruning approach, they find pruning harms robustness more when using smaller networks, which is consistent with our `CARD` hypothesis that states that the starting network must be sufficiently overparameterized.

## B  Experiment settings

All codes were written in Python using Pytorch and were run on IBM Power9 CPU with 256 GB of RAM and one to two NVIDIA V100 GPUs. Publicly available code was used as the base for each pruning method for models pruned with FT and GMP[5], LTH and LRR[6], EP[7] and BP[8]. We

---

[5] https://github.com/RAIVNLab/STR
[6] https://github.com/facebookresearch/open_lth
[7] https://github.com/allenai/hidden-networks
[8] https://github.com/chrundle/biprop

| | Learning Rate | | | LR Schedule | | Optimizer | | Weight Decay | | Epochs | | Pruning Details | |
|---|---|---|---|---|---|---|---|---|---|---|---|---|---|
| | Conv2 | Conv4/6/8 | Rest | Conv2/4/6/8 | Rest | Conv2/4/6/8 | Rest | Conv2/4/6/8 | Rest | Conv2/4/6/8 | Rest | Conv2/4/6/8 | Rest |
| Dense | 2e-4 | 3e-4 | 0.1 | None | LR160 | Adam | SGD | 0 | 1e-4 | 100 | 160 | N/A | N/A |
| FT | $\leftarrow$ 0.01 $\rightarrow$ | | 0.1 | Cosine | LR160 | $\leftarrow$ SGD $\rightarrow$ | | $\leftarrow$ 1e-4 $\rightarrow$ | | $\leftarrow$ 200 $\rightarrow$ | | $\longleftarrow$ Prune at epoch 160 then fine tune 40 epochs $\longrightarrow$ | |
| GMP | $\leftarrow$ 0.01 $\rightarrow$ | | 0.1 | Cosine | LR160 | $\leftarrow$ SGD $\rightarrow$ | | $\leftarrow$ 1e-4 $\rightarrow$ | | $\leftarrow$ 160 $\rightarrow$ | | $\longleftarrow (s_i, t, n, \Delta t) = (0, 5, 105, 1) \longrightarrow$ | |
| LTH | 5e-3 | 1e-2 | 0.1 | $\leftarrow$ LR160 $\rightarrow$ | | $\leftarrow$ SGD $\rightarrow$ | | $\leftarrow$ 1e-4 $\rightarrow$ | | $\leftarrow$ 160 $\rightarrow$ | | rewind it.: 1000, rate: 20% | rewind it.: 5000, rate: 20% |
| LRR | 5e-3 | 1e-2 | 0.1 | $\leftarrow$ LR160 $\rightarrow$ | | $\leftarrow$ SGD $\rightarrow$ | | $\leftarrow$ 1e-4 $\rightarrow$ | | $\leftarrow$ 160 $\rightarrow$ | | rewind it.: 1000, rate: 20% | rewind it.: 5000, rate: 20% |
| BP | $\longleftarrow$ 0.1 $\longrightarrow$ | | | $\leftarrow$ Cosine $\rightarrow$ | | $\leftarrow$ SGD $\rightarrow$ | | $\leftarrow$ 1e-4 $\rightarrow$ | | $\leftarrow$ 250 $\rightarrow$ | | $\longleftarrow$ All Epochs $\longrightarrow$ | |
| EP | $\longleftarrow$ 0.1 $\longrightarrow$ | | | $\leftarrow$ Cosine $\rightarrow$ | | $\leftarrow$ SGD $\rightarrow$ | | $\leftarrow$ 1e-4 $\rightarrow$ | | $\leftarrow$ 250 $\rightarrow$ | | $\longleftarrow$ All Epochs $\longrightarrow$ | |

Table 1: Hyperparameters used when training dense baselines and each pruning method by model. Note that "Rest" refers to all other models trained in our experiments, such as VGG and ResNet.

added functionality for global pruning in FT, GMP, EP and BP as it was not implemented in existing repositories.

ResNet-18 results for rewinding strategies, LRR and LTH, make use of regular ResNet-18 [19] models while all other methods, including dense, make use of PreAct ResNet-18 [20] as it provided improved performance in terms of accuracy and robustness.

A breakdown of hyperparameters by model and pruning method is provided in Table 1. As mentioned in Section 2, for each pruning method we used hyperparameters tuned specifically for that method. The dense Conv2/4/6/8 models used a batch size of 60, as specified in Figure 2 of the original Lottery Ticket Hypothesis paper [12]. All pruned models and the remaining dense models were trained using a batch size of 128. In the LR schedule column, *Cosine* denotes cosine decay while *LR160* denotes a schedule that sets the learning rate to 0.01 at epoch 80 and 0.001 at epoch 120. All models trained using SGD use a momentum of 0.9.

We first note details of experiments using traditional pruning methods, fine-tuning (FT) and gradual magnitude pruning (GMP). For FT models, unpruned training takes place for 160 epochs at which point pruning to the full sparsity level takes place using global magnitude pruning. After pruning, fine-tuning of the pruned network takes place over 40 epochs where the learning rate is kept at the final value after pruning at epoch 160 [34, 43]. For GMP models, the sparsity level gradually increases over the course of the training process. In our experiments, the sparsity level at training step $t$ increases in accordance with equation (1) from [60] which we include here to interpret the GMP pruning details from Table 1:

$$s_t = s_f + (s_i - s_f)\left(1 - \frac{t - t_0}{n\Delta t}\right)^3, \text{ for } t \in \{t_0 + k\Delta t\}_{k=0}^n. \tag{4}$$

Here, $s_i$ denotes the initial sparsity level, $s_f$ denotes the final sparsity level, $n$ denotes the number of pruning steps, $t_0$ denotes the first training step where pruning is performed, and $s_t$ denotes the sparsity level at the current training step. Note that the values for $s_i$, $t_0$, $n$, and $\Delta t$ are provided in Table 1.

For rewinding methods, LTH and LRR, hyperparameters were chosen based on details from [12, 13, 11, 43]. Notably, our rewinding-iteration choices stemmed from the hyperparameter study shown in Figure 7 of [13], and the fact that the small Conv models performed well when rewound to iteration 0 in [12]. All LTH/LRR runs were implemented using a modified version of the OpenLTH repository [11].

For initialization methods, edgepopup (EP) and biprop (BP), pruning is achieved by learning a pruning mask that is applied to the randomly initialized networks weights and, in the case of BP, binarization is applied to the weights of the resulting pruned network. For EP networks, weights were initialized using the signed constant initialization from [41] which offered the best performance. As an added benefit for compactness, this initialization also yields a binary weight network. For BP networks, weights were initialized using the kaiming normal initialization as in [9] and the biprop algorithm performs binarization during training resulting in a binary-weight network. Due to the binary weights in both the EP and BP CARDs we trained, these CARDs provided further reductions in on-device memory consumption over rewinding based pruning strategies. For both EP and BP, we used the same number of epochs for training as in [9].

# C  Additional Experiments

## C.1  Effect of global vs. layerwise pruning in lottery ticket initialization methods.

The lottery ticket initialization methods analyzed in the Section 2 were originally developed to prune a percentage of weights uniformly across all layers of the network. In contrast, global pruning methods are considered to be more flexible as they can prune some layers more heavily than others while still meeting a user-specified sparsity level for the entire network. By analyzing these initialization methods using both layerwise and global pruning, we notice certain peculiar patterns. Figure 5 provides the accuracy and robustness of models trained with BP and EP using global and layerwise pruning. For each model, the maximum CIFAR-10 accuracy was achieved by a layerwise pruned model at one of the six sparsity levels. However, the globally-pruned models consistently outperform the layerwise pruned models on robustness at nearly every sparsity level. Furthermore, the globally-pruned models typically achieve higher or comparable accuracy at higher sparsity levels, indicating that initialization methods utilizing global pruning are more suitable when a high-level of sparsity is desired.

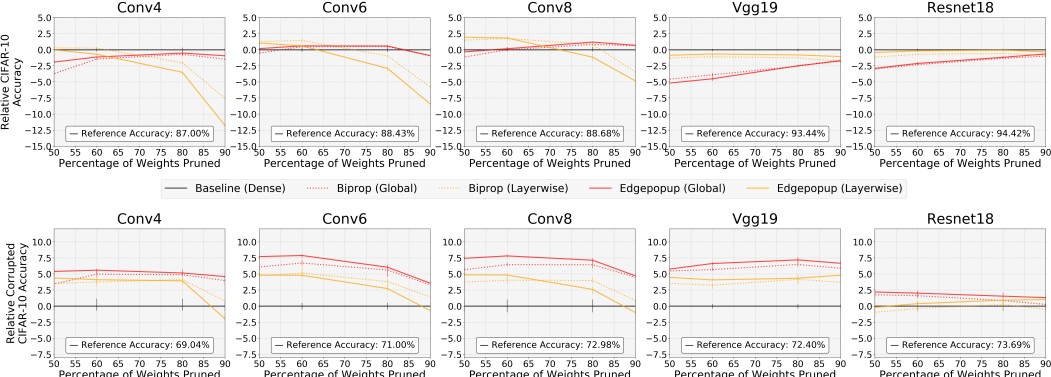

Figure 5: **Global pruning in lottery ticket Initialization methods provides greater robustness gains**: While layerwise pruning is able to achieve the highest accuracy across all sparsity levels in initialization methods, global pruning provides more significant robustness gains at all sparsity levels.

## C.2  Comparison of full-precision-weight Edgepopup pruning with binary-weight Edgepopup pruning

The models pruned using EP in our experiments are pruned using weights initialized from a scaled binary initialization, as specified in [41]. Additionally, models pruned with BP contain binary weights regardless of the initialization used. To demonstrate that the robustness gains afforded are a feature of initialization based pruning methods and not binarization, we provide some results for full-precision initialization based pruning models. In particular, by using the kaiming normal initialization with EP the resulting network has full-precision weights. In Figure 6, we visualize the accuracy of these models on CIFAR-10 and CIFAR-10-C. These experiments demonstrate the the robustness of the initialization based CARDs is not exclusive to binary weight networks as the full-precision weight networks can achieve comparable accuracy to the binary weight networks at some prune percentages.

# D  Additional heatmaps

Here we provide additional heatmaps (varying sparsity levels) for Conv8 (see Figures 7 and 8) and for ResNet18 models (see Figures 9, 10 and 11). By comparing the heatmaps of rewinding and initialization based pruning methods to baselines, we find that these models are more resilient to perturbations of varying severity.

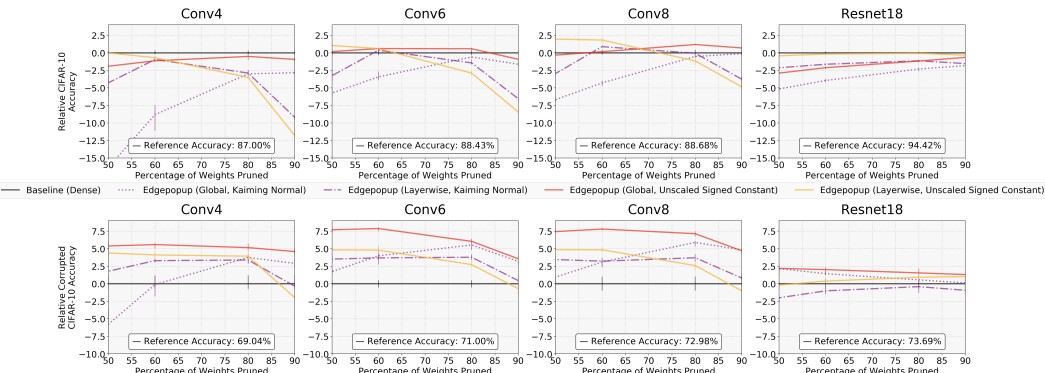

Figure 6: **Using Full Precision weights in lottery ticket Initialization still provides robustness gains**: While initialization pruning methods with binary weights yield the greatest robustness gains over the baseline, randomly initialized networks with full-precision weights pruned using Edgepopup are capable of providing improved robustness over the dense baseline.

## D.1 Additional Conv8 heatmaps

In the Conv8 models, differences in the heatmaps of initialization methods and the baseline model persist up to the highest sparsity level of 95%, as seen in Figure 8. The top three rows in each figure provide the Fourier heatmaps for each model while the bottom three rows provide the difference to the dense baseline. In the difference heatmaps, blue pixels are where the compressed model has an error rate lower than the dense model and red pixels are where the compressed model has an error rate higher than the dense model.

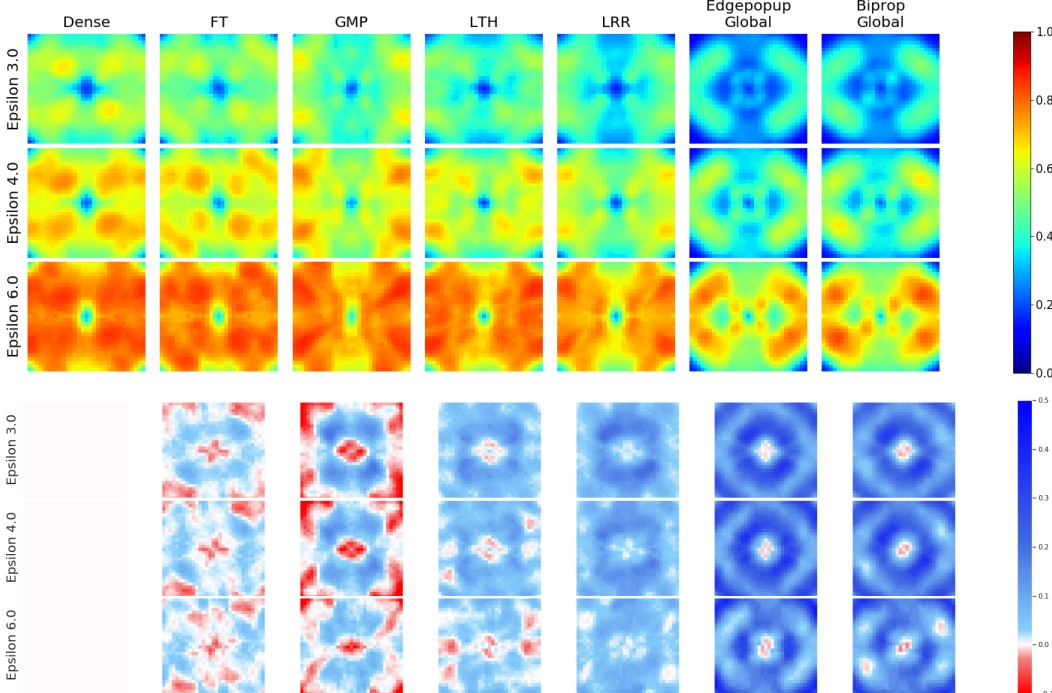

Figure 7: **Visualizing the response of compressed models to perturbations at different frequencies**: The top three rows are Fourier heatmaps for error rate of Conv8 models trained on CIFAR-10 with 90% of weights pruned. The bottom three rows are difference to the baseline with blue regions indicating lower error rate than baseline.

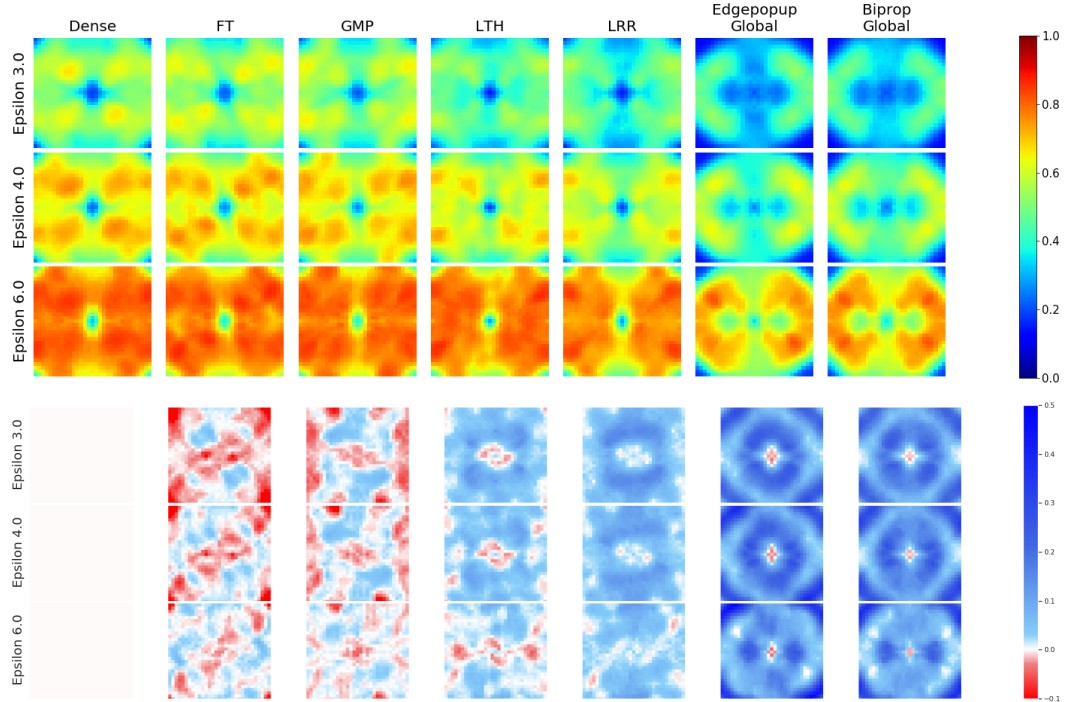

Figure 8: **Visualizing the response of compressed models to perturbations at different frequencies**: The top three rows are Fourier heatmaps for error rate of Conv8 models trained on CIFAR-10 with 95% of weights pruned. The bottom three rows are difference to the baseline with blue regions indicating lower error rate than baseline.

## D.2 ResNet-18 heatmaps

Here we provide Fourier error rate heatmaps for the ResNet-18 architecture trained using different pruning methods. As in the Conv8 heatmap figures, we include heatmaps for a trained dense ResNet-18 model for reference and the difference heatmaps clearly conveying the difference of each compression method to the baseline.

## E Constructing iterative pruning with rewinding from fine-tuning

While LTH [12] and LRR [43] offer unsurpassed performance, such approaches also greatly extend the training duration, pruning just 20% of the remaining weights every $T - r$ epochs, where $T$ is the initial training duration and $r$ is the epoch the weights/learning-rate are rewound to after each pruning event (here, $r = 12$ and $T = 160$). This raises the question: *Is longer training and the multi-shot pruning procedure critical to the robustness improvements LTH/LRR offer relative to FT/GMP?*

To test this, we gradually construct the LTH/LRR pruning approaches used in this paper by starting from a fine-tuning approach and adding modifications until we produce the LTH/LRR method that prunes the network 13 times to reach 95% sparsity. The phases of this construction for LRR are illustrated in Figure 12, wherein we plot a column of Fourier heatmaps for each phase. Specifically, the first column is our FT approach, the second column extends the fine-tuning duration, the third column adds learning-rate rewinding to this fine-tuning period, the fourth column decreases the iterative prune rate to achieve 95% sparsity in 4 shots rather than 1, and subsequent columns continue to increase the number of pruning shots. In this construction process, we find a notable benefit of adding learning-rate rewinding (70.9% to 72.6% CIFAR-10-C accuracy moving from column 2 to column 3), but the biggest benefits of LTH/LRR come from combining this rewinding with multiple iterations (i.e., all columns from 4 onward display at least 75% robust accuracy). Interestingly, our results also indicate that it may be possible to achieve the robustness benefits of LTH/LRR with

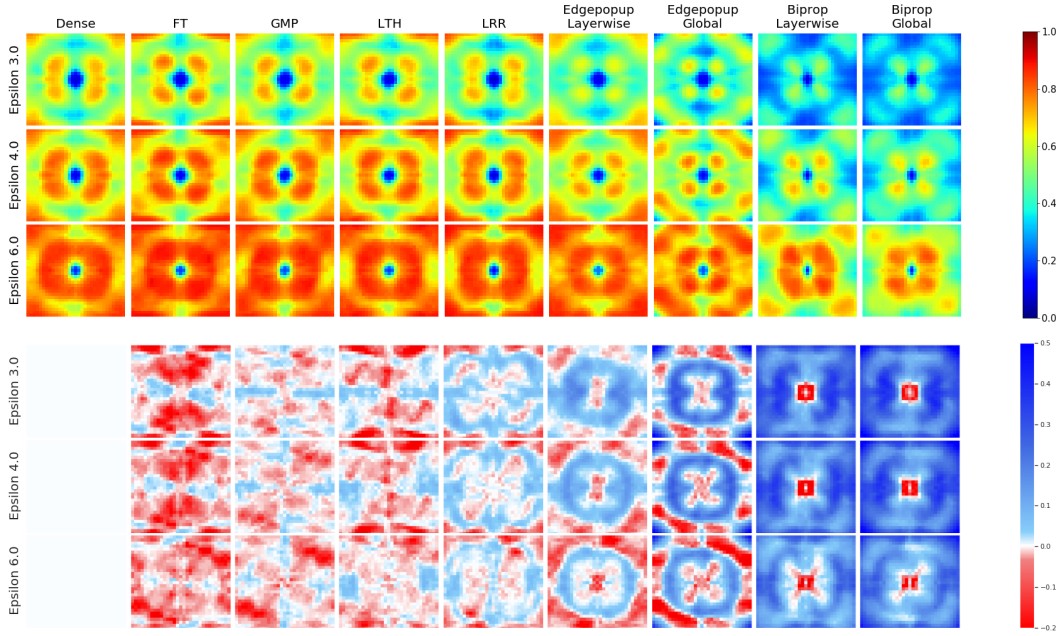

Figure 9: **Visualizing the response of compressed models to perturbations at different frequencies**: The top three rows are Fourier heatmaps for error rate of ResNet-18 models trained on CIFAR-10 with 80% of weights pruned. The bottom three rows are difference to the baseline with blue regions indicating lower error rate than baseline.

a higher iterative pruning rate and thus fewer pruning shots/iterations than what is standard in the literature [12, 43].

We now repeat this experiment using 90% sparsity (Figure 14), using LTH instead of LRR at 95% sparsity (Figure 13), and using LTH and 90% sparsity (Figure 15).

At 95% sparsity, we observe the same pattern: adding multiple shots of pruning is critical to improving the LTH heatmaps and robustnesses of the rewinding-based methods (Figure 13). That is to say, adding rewinding and a longer post-pruning fine-tuning duration to our FT method is not sufficient to obtain the results achievable with LTH/LRR—multiple iterations are needed. Interestingly, as especially visible at epsilon 6.0 in the Fourier heatmaps, LRR (Figure 12) is clearly more resilient to perturbations than LTH, which is consistent with the improved performance of LRR relative to LTH.

At 90% sparsity, for both LTH (Figure 15) and LRR (Figure 14), the Fourier heatmaps reflect benefits of multiple shots and rewinding (particularly near the centers of the images for all epsilons). For LRR, there is greater similarity among the Rewinding and Initialization Fourier heatmaps at 90% sparsity than at 95% sparsity, and this is reflected in their robustnesses in the captions, which are less separated in the 90% sparsity case. Notably, however, all these robustness figures are consistent with the aforementioned heatmap improvements in that they show the benefits of combining rewinding with multiple pruning shots. Note that 10-shot pruning corresponds to the scheme / iterative pruning rate (20%) we use to reach 90% sparsity in other sections (e.g., Figure 10).

# F   Additional results with CARDS and CARD-Deck

In this section, we provide tables for all experimental results from Section 3. This includes tables for individual CARDs on CIFAR-10 for ResNet-18 (Table 2), ResNeXt-29 (Table 13), ResNet-50 (Table 14), and WideResNet-18-2 (Table 15). Additionally, we provide tables for CIFAR-10 CARD-Decks using ResNet-18 (Table 3), WideResNet-18-2 (Table 4), and CIFAR-100 CARD-Decks using WideResNet-18-2 (Table 5). Breakdowns for the performance of CIFAR-10 ResNet-18 CARDs and CARD-Decks on each of the 15 corruption types in CIFAR-10-C are provided in Tables 6 – Tables 11. As a reference, tables for individual CARDs provide results for dense baseline models. Due

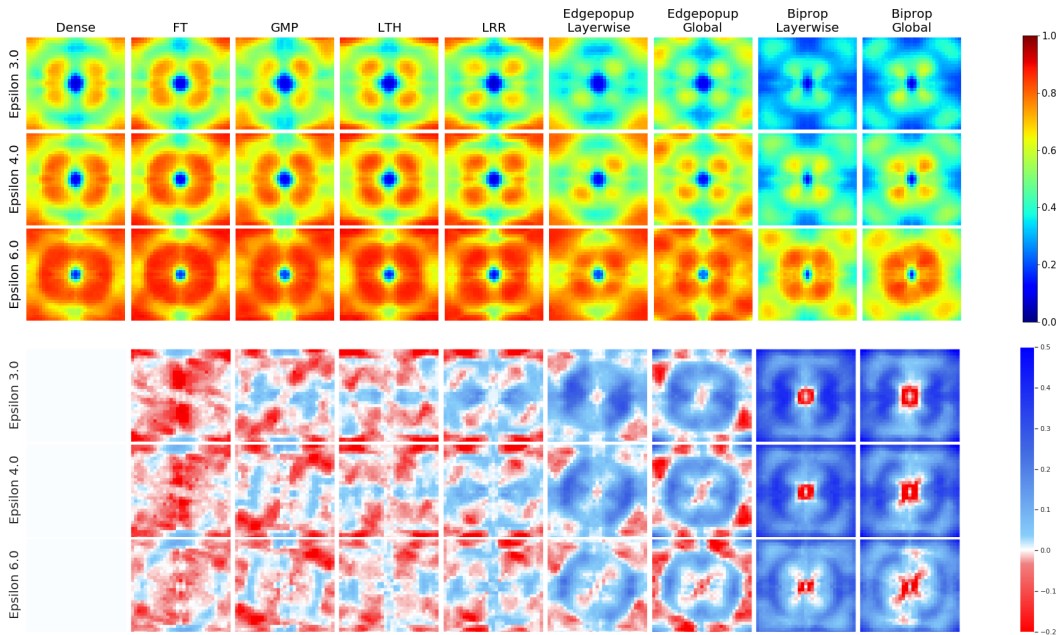

Figure 10: **Visualizing the response of compressed models to perturbations at different frequencies**: The top three rows are Fourier heatmaps for error rate of ResNet-18 models trained on CIFAR-10 with 90% of weights pruned. The bottom three rows are difference to the baseline with blue regions indicating lower error rate than baseline.

| | | Baseline | | | CARD | | | | | | | | | | | | | | | | | |
| | | Dense | | | Edgepopup | | | | | | LRR | | | LTH | | | Biprop | | | | | |
| | | - | | | Layerwise | | | Global | | | Global | | | Global | | | Layerwise | | | Global | | |
| | | Augmix | Clean | Gauss. | Augmix | Clean | Gauss. | Augmix | Clean | Gauss. | Augmix | Clean | Gauss. | Augmix | Clean | Gauss. | Augmix | Clean | Gaussian | Augmix | Clean | Gauss. |
|---|---|---|---|---|---|---|---|---|---|---|---|---|---|---|---|---|---|---|---|---|---|---|
| **80%** | Clean Acc. | **95.5** | 95.1 | 93.9 | 94.3 | 93.7 | 92.4 | **94.9** | 94.4 | 93 | 96.1 | 95.6 | 93.8 | 95.6 | 94.9 | 93.5 | 93.9 | 93.7 | 92.4 | **94.5** | 94.1 | 93.2 |
| | Robust Acc. | **89.2** | 73.7 | 85.6 | 87.8 | 76.7 | 85.1 | **88.4** | 74.4 | 85.7 | 89.8 | 75.7 | 86.4 | 89.4 | 74.4 | 85.8 | 87.5 | 76.1 | 85.1 | **87.8** | 74.3 | 85.3 |
| | Memory (Mbit) | **358** | 358 | 358 | 2.23 | 2.23 | 2.23 | 2.23 | 2.23 | 2.23 | 71.5 | 71.5 | 71.5 | 71.5 | 71.5 | 71.5 | 2.23 | 2.23 | 2.23 | 2.23 | 2.23 | 2.23 |
| **90%** | Clean Acc. | **95.5** | 95.1 | 93.9 | 94.4 | 93.9 | 92.9 | 94.4 | 94.1 | 92.8 | **96.3** | 95.6 | 93.9 | 95.7 | 95.2 | 93.6 | **94.4** | 94.1 | 92.7 | 93.7 | 93.6 | 93 |
| | Robust Acc. | **89.2** | 73.7 | 85.6 | **88** | 75.6 | 85.2 | 87.9 | 76 | 85.4 | **89.8** | 76.1 | 86.3 | 89.7 | 74.3 | 86 | **87.8** | 75.1 | 85 | 87.1 | 74.6 | 84.2 |
| | Memory (Mbit) | **358** | 358 | 358 | 1.12 | 1.12 | 1.12 | 1.12 | 1.12 | 1.12 | 35.8 | 35.8 | 35.8 | 35.8 | 35.8 | 35.8 | 1.12 | 1.12 | 1.12 | 1.12 | 1.12 | 1.12 |
| **95%** | Clean Acc. | **95.5** | 95.1 | 93.9 | **94.5** | 94 | 92.6 | 93.2 | 92.7 | 91.2 | 96.1 | 95.7 | 93.9 | **95.8** | 95.1 | 93.8 | 94.2 | 93.8 | 92.5 | 92.5 | 92.1 | 91.4 |
| | Robust Acc. | **89.2** | 73.7 | 85.6 | 87.8 | 73.1 | 84.3 | 85.7 | 73.4 | 83.9 | 89.6 | 75.6 | 86.3 | **89.7** | 74.3 | 86 | 87.5 | 73.8 | 84.4 | 84.7 | 73.4 | 83.1 |
| | Memory (Mbit) | **358** | 358 | 358 | **0.56** | 0.56 | 0.56 | **0.56** | 0.56 | 0.56 | 17.9 | 17.9 | 17.9 | **17.9** | 17.9 | 17.9 | **0.56** | 0.56 | 0.56 | **0.56** | 0.56 | 0.56 |

Table 2: Performance comparison between dense baselines and CARDs using ResNet-18 architecture. Clean and Robust Acc. refer to accuracy on CIFAR-10 and CIFAR-10-C, respectively. The best performance for each method is shown in bold.

to the structure of the table these results are intentionally repeated at each sparsity level (the dense baselines are not pruned so their performance remains constant).

## F.1 Tables of CARD and CARD-Decks results for ResNet-18 and WideResNet-18

The clean and robust accuracies (averaged across three realizations) of CARDs for each pruning scheme are provided in Table 2. We find that CARDs perform comparably to (and in some cases better than) their dense counterparts in terms of accuracy and robustness but have a significantly smaller memory footprint.

| | | CARD-Deck (Agnostic/Adaptive) | | | | | | | | | | | |
|---|---|---|---|---|---|---|---|---|---|---|---|---|---|
| | | Edgepopup (Global) | | | LRR | | | LTH | | | Biprop (Global) | | |
| | | 2 | 4 | 6 | 2 | 4 | 6 | 2 | 4 | 6 | 2 | 4 | 6 |
| 80% | Clean Acc. | 92.1/94.1 | 94/94.5 | 94.2/94.8 | 96/96 | 96.3/96.4 | 96.4/96.6 | 95.5/95.5 | 96/96.1 | 96.1/96.1 | 93.8/93.8 | 94.3/94.3 | 94.3/94.4 |
| | Robust Acc. | 85/88.9 | 88.4/89.8 | 88.9/90 | 89.7/90.9 | 91.7/91.8 | 91.9/92 | 89.4/90.4 | 91/91.2 | 91.2/91.4 | 87.5/88.6 | 88.8/89.3 | 89/89.6 |
| | Memory (Mbit) | 4.47 | 8.94 | 13.4 | 143 | 286 | 429 | 143 | 286 | 429 | 4.47 | 8.94 | 13.4 |
| 90% | Clean Acc. | 92.9/94.4 | 94.6/94.8 | 94.7/94.8 | 96.3/96.3 | 96.4/96.4 | 96.4/**96.6** | 95.7/95.7 | 95.9/95.7 | 96.2/**96.2** | 94/94 | 94.6/94.4 | 94.5/94.6 |
| | Robust Acc. | 85.2/89.2 | 88.6/90.1 | 89.3/**90.4** | 89.8/91.1 | 91.7/91.8 | 92/**92.1** | 89.4/90.5 | 91/91.3 | 91.3/**91.5** | 87.4/88.6 | 89.2/89.5 | 89.4/**89.9** |
| | Memory (Mbit) | 2.23 | 4.47 | 6.70 | 71.5 | 143 | 215 | 71.5 | 143 | 215 | 2.23 | 4.47 | 6.70 |
| 95% | Clean Acc. | 92.6/94.5 | 94.2/94.8 | 94.5/**95.1** | 96.1/96.1 | 96.3/96.4 | 96.3/96.5 | 95.8/95.8 | 96/96.1 | 96.1/96.2 | 94/94 | 94.7/94.5 | **94.7**/94.6 |
| | Robust Acc. | 84.3/88.6 | 87.7/89.5 | 88.4/89.9 | 89.6/91 | 91.6/91.8 | 91.9/92 | 89/90.3 | 90.8/91.1 | 91.1/91.4 | 87.2/88.5 | 88.9/89.4 | 89.2/89.7 |
| | Memory (Mbit) | **1.12** | 2.23 | 3.35 | **35.8** | 71.5 | 107 | **35.8** | 71.5 | 107 | **1.12** | 2.23 | 3.35 |

Table 3: Performance of domain-agnostic and domain-adaptive ResNet-18 `CARD-Decks`. Clean and Robust Acc. refer CIFAR-10 and CIFAR-10-C accuracy, respectively.

| | | CARD-Deck (Agnostic/Adaptive) | | | | | | | | | | | |
|---|---|---|---|---|---|---|---|---|---|---|---|---|---|
| | | Edgepopup (Global) | | | LRR | | | LTH | | | Biprop (Global) | | |
| | | 2 | 4 | 6 | 2 | 4 | 6 | 2 | 4 | 6 | 2 | 4 | 6 |
| 90% | Clean Acc. | 92.4/ 92.9 | 94.0/ 94.8 | 94.8/ 95.1 | 96.3/ 96.3 | 96.7/ 96.7 | 96.7/ 96.8 | 96.1/ 96.1 | 96.5/ 96.4 | 96.6/ 96.6 | 92.4/ 93.1 | 94.3/ 94.5 | 94.9/ 95.0 |
| | Robust Acc. | 85.1/ 86.2 | 88.6/ 90.0 | 90.1/ **90.6** | 90.6/ 91.7 | 92.3/ 92.3 | 92.5/ 92.6 | 90.1/ 91.2 | 91.6/ 91.8 | 91.9/ 92.1 | 85.3/ 86.2 | 88.7/ 89.8 | 89.9/ 90.5 |
| | Memory (Mbit) | 8.93 | 17.86 | 26.79 | 285.8 | 571.6 | 857.5 | 285.8 | 571.6 | 857.5 | 8.93 | 17.86 | 26.79 |
| 95% | Clean Acc. | 92.8/ 93.4 | 94.6/ 94.9 | 95.1/ **95.3** | 96.3/ 96.3 | 96.8/ 96.8 | 96.6/**96.8** | 96.1/ 96.1 | 96.5/ 96.4 | 96.6/ **96.7** | 92.3/ 92.8 | 94.2/ 94.5 | 95.0/ **95.2** |
| | Robust Acc. | 85.2/ 86.1 | 88.6/ 89.9 | 90.0/ **90.6** | 90.8/ 91.8 | 92.4/ 92.5 | 92.7/**92.75** | 89.9/ 91.4 | 91.6/ 91.9 | 91.9/ **92.2** | 84.9/ 86.0 | 88.4/ 89.5 | 90.0/ **90.5** |
| | Memory (Mbit) | **4.46** | 8.93 | 13.39 | **142.9** | 285.8 | 428.7 | **142.9** | 285.8 | 428.7 | **4.46** | 8.93 | 13.39 |

Table 4: Performance of domain-agnostic and domain-adaptive WideResNet-18 `CARD-Decks`. Clean and Robust Acc. refer to CIFAR-10 and CIFAR-10-C accuracy, respectively. The best performance for each method is shown in bold. For reference, dense WideResNet-18 (AugMix) model achieves (Clean Acc., Robust Acc., Memory) = (95.6%, 89.3%, 1429 Mbit).

| | | CARD-Deck (Agnostic/Adaptive) | | | | | | | | | | | |
|---|---|---|---|---|---|---|---|---|---|---|---|---|---|
| | | Edgepopup (Global) | | | LRR | | | LTH | | | Biprop (Global) | | |
| | | 2 | 4 | 6 | 2 | 4 | 6 | 2 | 4 | 6 | 2 | 4 | 6 |
| 90% | Clean Acc. | 77.1/77.1 | 78.5/78.4 | 78.6/78.7 | 78.3/78.3 | 79.6/79.7 | 79.6/80.2 | 78.2/78.2 | 79.6/79.7 | 79.9/80.3 | 76.9/76.9 | 77.8/78.1 | 78.0/**78.3** |
| | Robust Acc. | 66.3/67.8 | 69.5/69.6 | 69.9/**70.3** | 66.9/68.8 | 70.5/70.7 | 71.0/71.2 | 66.2/68.6 | 69.8/70.4 | 70.6/71.0 | 65.8/67.3 | 68.7/69.0 | 69.1/**69.6** |
| | Memory (Mbit) | 8.95 | 17.90 | 26.85 | 286.5 | 572.9 | 859.3 | 286.5 | 572.9 | 859.3 | 8.95 | 17.90 | 26.85 |
| 95% | Clean Acc. | 77.1/77.1 | 78.5/78.5 | 78.7/**79.1** | 78.7/78.7 | 79.9/80.2 | 80.1/**80.6** | 78.2/78.2 | 79.7/80.0 | 79.8/**80.4** | 76.0/76.0 | 77.4/77.1 | 77.9/77.8 |
| | Robust Acc. | 65.6/67.1 | 68.9/69.0 | 69.4/69.7 | 67.1/68.8 | 70.6/70.7 | 71.1/**71.3** | 66.5/68.6 | 70.1/70.3 | 70.7/**71.0** | 64.8/66.5 | 67.9/68.1 | 68.4/68.7 |
| | Memory (Mbit) | **4.48** | 8.95 | 13.43 | **143.2** | 286.5 | 429.7 | **143.2** | 286.5 | 429.7 | **4.48** | 8.95 | 13.43 |

Table 5: Performance of domain-agnostic and domain-adaptive WideResNet-18 `CARD-Decks`. Clean and Robust Acc. refer to CIFAR-100 and CIFAR-100-C accuracy, respectively. The best performance for each method is shown in bold. For reference, dense WideResNet-18 (AugMix) model achieves (Clean Acc., Robust Acc., Memory) = (77.5%, 66.1%, 1433 Mbit).

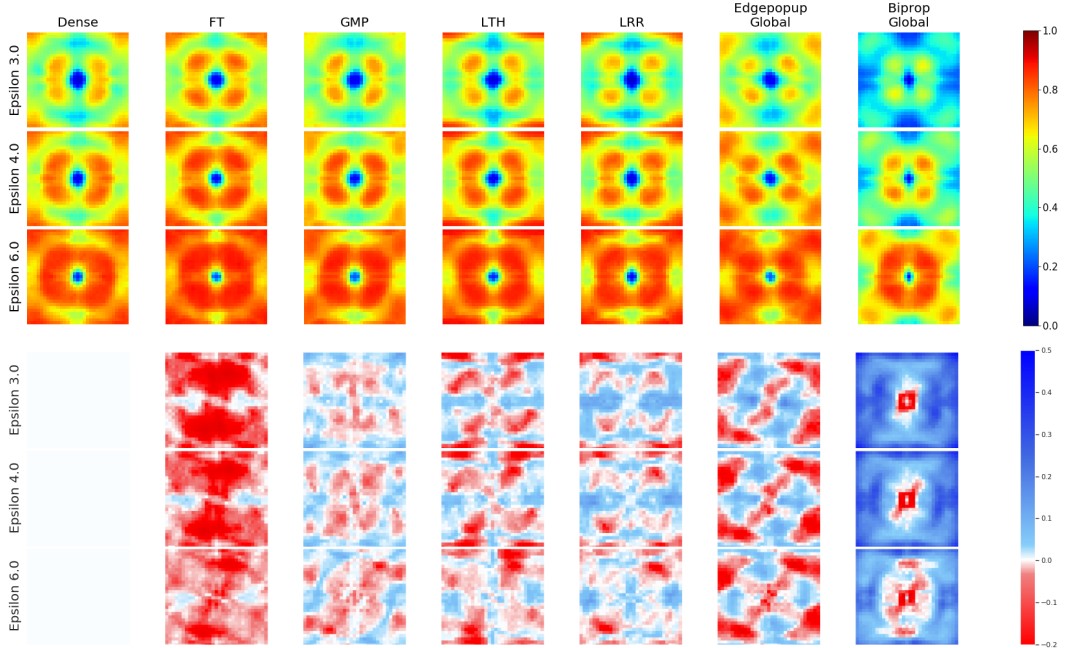

Figure 11: **Visualizing the response of compressed models to perturbations at different frequencies**: The top three rows are Fourier heatmaps for error rate of ResNet-18 models trained on CIFAR-10 with 95% of weights pruned. The bottom three rows are difference to the baseline with blue regions indicating lower error rate than baseline.

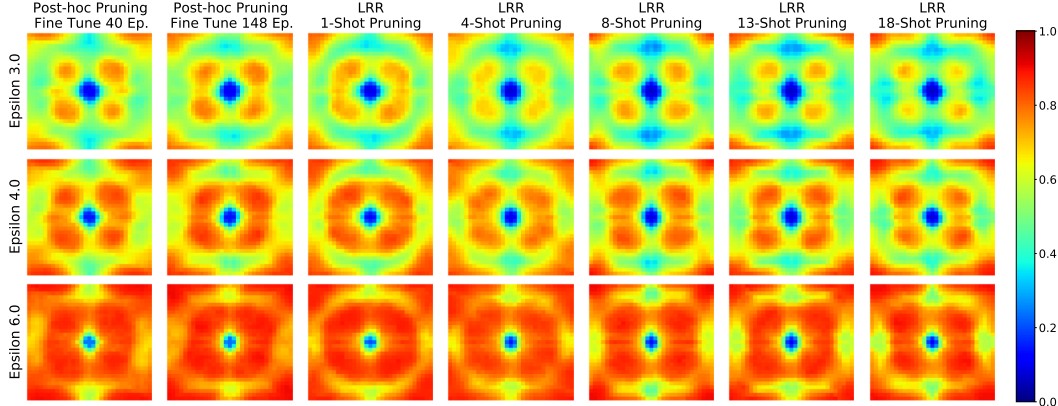

Figure 12: **Comparing the resiliencies of Rewinding and Traditional methods to perturbations at different frequencies**: Fourier heatmaps for error rate of ResNet18 models trained on CIFAR-10 in which 95% of the weights are pruned.

## F.2 Additional results for CARDs and CARD-Decks with ResNet-18

In this section, we provide a breakdown of the accuracy of ResNet-18 `CARDs` and `CARD-Decks` by CIFAR-10-C corruption types. In particular, Tables 6 - 8 contain the performance of `CARDs` trained on clean, Augmix, and Gaussian augmentations when tested on CIFAR-10-C corruption types. Tables 9 to 11 contain the performance of LTH, LRR, EP, and BP `CARD-Decks` on individual CIFAR-10-C corruptions.

As a note of interest, we found that the best performance on different CIFAR-10-C corruptions changes for individual `CARDs` as the sparsity level increases. At 80% sparsity, a Gaussian `CARD` yields the highest accuracy on impulse noise but at 90% and 95% sparsity levels Augmix `CARDs` deliver the

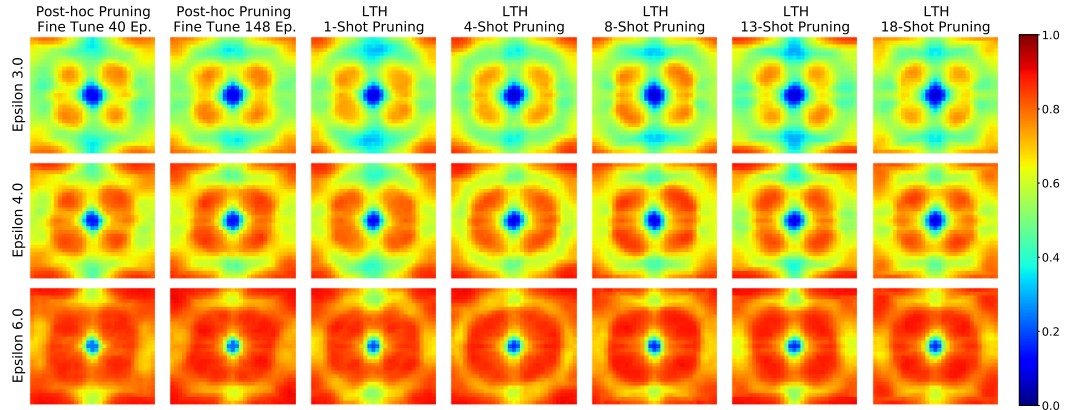

Figure 13: **Comparing the resiliencies of Rewinding and Traditional methods to perturbations at different frequencies**: Fourier heatmaps for error rate of ResNet18 models trained on CIFAR-10 in which 95% of the weights are pruned via LTH.

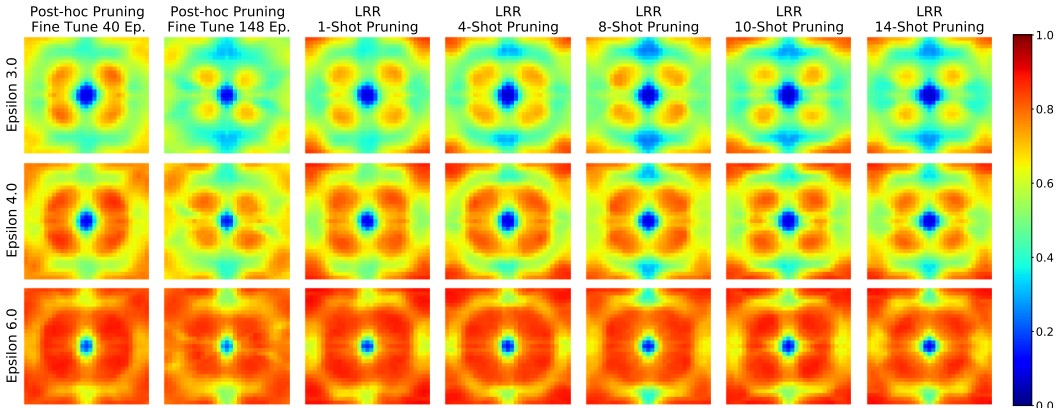

Figure 14: **Comparing the resiliencies of Rewinding and Traditional methods to perturbations at different frequencies**: Fourier heatmaps for error rate of ResNet18 models trained on CIFAR-10 in which 90% of the weights are pruned via LRR.

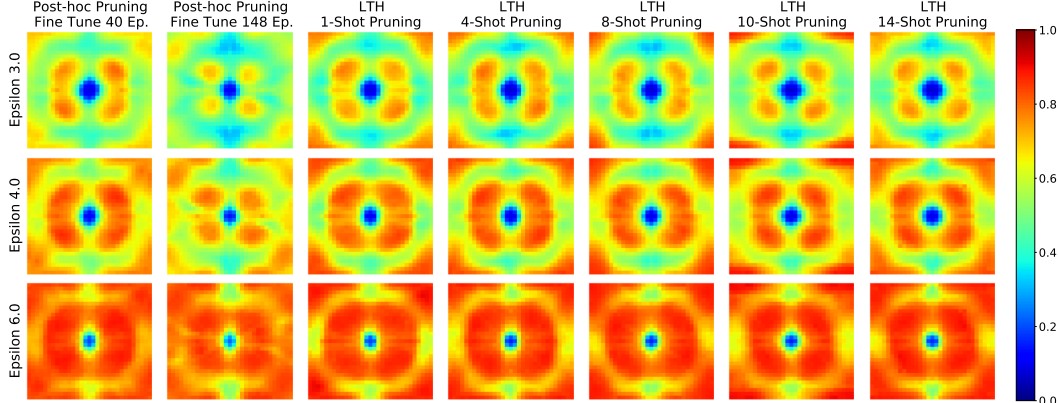

Figure 15: **Comparing the resiliencies of Rewinding and Traditional methods to perturbations at different frequencies**: Fourier heatmaps for error rate of ResNet18 models trained on CIFAR-10 in which 90% of the weights are pruned via LTH.

highest accuracy on impulse noise. Further, at 95% sparsity the margin of difference in accuracy on impulse noise provided by the Augmix CARD over the Gaussian CARD is more significant.

While pruning using FT and GMP were unable to yield CARDs, note that we include the accuracy and robustness of ResNet-18 models pruned using FT and GMP with the same augmentation schemes in Table 12 for comparison against the performance of ResNet-18 CARDs in Table 2.

### F.3 Achieving state-of-the-art performance on CIFAR-10-C using larger models

We report these results in Tables 13, 14 and 15. To summarize, our results highlight the fact that the accuracy/robustness gains due to the model compression (and ensembling) are compatible with the gains from the existing strategies, i.e., data augmentation and the use of larger models. By combining these strategies with the scheme proposed in this paper, we achieve even larger gains in terms of robustness and accuracy, in turn, establishing a new SOTA. Note that we include performance of WideResNet-18 CARD-Decks composed of layerwise pruned BP and EP models in Table 16.

### F.4 Note on gating function performance

In Table 17, we provide a break down of the performance of the spectral-similarity based gating function by CIFAR-10-C corruption type. For each augmentation scheme and corruption type, the corresponding number indicates the percetage of data from that corruption selected by the gating function averaged across the 5 severity levels in CIFAR-10-C. Based on the performance of Augmix and Gaussian CARDs by CIFAR-10-C corruption type in Tables $6-8$, entries in the table are marked in bold whenever a model pruned to sparsity 80%, 90%, or 95% trained using that data augmentation scheme achieved the highest accuracy averaged over all severity levels of that corruption type. Bolding these entries in Table 17 indicates that the gating function typically selects the best performing augmentation scheme, and thereby the CARDs in the deck trained on data most similar to the incoming test data, for the domian-adaptive CARD-Decks. Improvements could be made by determining a gating function that is more accurate on the frost and jpeg corruptions. As noted in Section F.2, the augmentation scheme yielding the best performing models on impulse and glass corruptions varies with the sparsity level of the pruned network. This observation indicates that an alternative similarity metric that takes into account features of the trained CARDs, such as sparsity level, could provide a gating function that offers improved performance on CIFAR-10-C corruptions.

## G Theory

### G.1 Proof of Corollary 2

Using the triangle inequality, we have that

$$\left\| \sum_{k=1}^{n} \lambda_i f_k(x) - \sum_{k=1}^{n} \lambda_i F_k(\ell, \boldsymbol{w})(x) \right\| \leq \sum_{k=1}^{n} \lambda_i \| f_k(x) - F_k(\ell, \boldsymbol{w})(x) \|, \tag{5}$$

for any $x \in \mathcal{X}$. Hence, if

$$\sup_{x \in \mathcal{X}} \| f_k(x) - F_k(\ell, \boldsymbol{w})(x) \| \leq \varepsilon \tag{6}$$

for each $1 \leq k \leq n$, then it immediately follows that

$$\left\| \sum_{k=1}^{n} \lambda_i f_k(x) - \sum_{k=1}^{n} \lambda_i F_k(\ell, \boldsymbol{w})(x) \right\| \leq \sum_{k=1}^{n} \lambda_i \varepsilon = \varepsilon, \tag{7}$$

for all $x \in \mathcal{X}$. Under the hypotheses of Theorem 3 in [38] (Theorem 2 in [9]), for each $k \in \{1, \dots, n\}$ we have that with probability $(1 - \delta)$ there exists a full-precision (binary-weight) CARD satisfying (6). Thus, with probability $(1 - \delta)^n$ there exists a collection of full-precision (binary-weight) networks $\{f_k\}$ satisfying (1).

## G.2 OOD Robustness analysis

To understand the average OOD robustness better, we derive the following decomposition:

$$
\begin{aligned}
Rob(\mathcal{D}_c, f^a) &= Rob(\hat{S}_a, f^a) \\
&+ [Rob(\mathcal{D}_a, f^a) - Rob(\hat{S}_a, f^a)] \\
&+ [Rob(\mathcal{D}_c, f^a) - Rob(\mathcal{D}_a, f^a)].
\end{aligned}
$$

Next, using the triangle inequality $a + (b - a) \geq a - \|b - a\|$ which is true because $\|b - a\| \geq a - b$ for $a, b \geq 0$, we have

$$
\begin{aligned}
Rob(\mathcal{D}_c, f^a) &\geq Rob(\hat{S}_a, f^a) \\
&- \|Rob(\mathcal{D}_a, f^a) - Rob(\hat{S}_a, f^a)\| \\
&- \|Rob(\mathcal{D}_c, f^a) - Rob(\mathcal{D}_a, f^a)\|.
\end{aligned}
$$

By linearity of expectation, we can bound (2) from below

$$
Rob(\mathcal{D}^C, f^{Deck}) \geq \sum_{a=1}^{|A|} \sum_{c=1}^{|C|} w_c^a Rob(\hat{S}_a, f^a) \tag{8}
$$

$$
- \sum_{a=1}^{|A|} \sum_{c=1}^{|C|} w_c^a \|Rob(\mathcal{D}_a, f^a) - Rob(\hat{S}_a, f^a)\| \tag{9}
$$

$$
- \sum_{a=1}^{|A|} \sum_{c=1}^{|C|} w_c^a \|Rob(\mathcal{D}_c, f^a) - Rob(\mathcal{D}_a, f^a)\|. \tag{10}
$$

Note that we have bounded (2) in terms of the following three error terms for a classifier-corruption pair weighted by their gating (or selection) probabilities: 1) empirical robustness (8), 2) generalization gap (9), and 3) OOD-shift (10).

Next, we aim to provide a bound on the OOD-shift that is independent of the classifiers in hand and is only related to the properties of the distributions. To facilitate this, we define a notion of distance between two distributions.

**Definition 2** ( (Conditional Wasserstein distance). *For two labeled distributions $\mathcal{D}$ and $\mathcal{D}'$ with supports on $X \times Y$, we define conditional Wasserstein distance according to a distance metric $d$ as follows:*

$$
W(\mathcal{D}, \mathcal{D}') = \mathop{\mathbb{E}}_{(.,y)\sim\mathcal{D}} \left[ \inf_{J \in \mathcal{J}(\mathcal{D}|y, \mathcal{D}'|y)} \mathop{\mathbb{E}}_{(x,x')\sim J} d(x, x') \right], \tag{11}
$$

*where $\mathcal{J}(\mathcal{D}, \mathcal{D}')$ is the set of joint distributions whose marginals are identical to $\mathcal{D}$ and $\mathcal{D}'$.*

Conditional Wasserstein distance between the two distributions is simply the expectation of Wasserstein distance between conditional distributions for each class.

| Model | | Clean | Avg. Robust | Memory (Mbit) | Noise | | | Blur | | | | Weather | | | | Digital | | | |
|---|---|---|---|---|---|---|---|---|---|---|---|---|---|---|---|---|---|---|---|
| | | | | | gaussian | shot | impulse | defocus | glass | motion | zoom | snow | frost | fog | brightness | contrast | elastic | pixelate | jpeg |
| **Baseline** | Dense (Augmix) | 95.5 | 89.2 | 358 | 81.7 | 85.9 | 86.7 | 94.3 | 80.8 | 92.4 | 93.2 | 90.0 | 89.7 | 92.2 | 94.7 | 91.0 | 89.4 | 88.7 | 88.7 |
| | Dense (Gaussian) | 93.9 | 85.6 | 358 | 91.3 | 91.8 | 88.3 | 85.8 | 81.0 | 80.9 | 84.9 | 86.4 | 88.3 | 81.9 | 92.3 | 90.6 | 86.6 | 88.1 | 87.7 |
| | Dense (Clean) | 95.1 | 73.7 | 358 | 46.5 | 59.1 | 54.0 | 81.8 | 55.1 | 78.1 | 76.4 | 82.4 | 78.2 | 88.2 | 93.5 | 78.2 | 84.0 | 76.1 | 79.3 |
| **CARDs** | LRR (Augmix) | **96.1** | **89.8** | 71.5 | 78.5 | 85.0 | 88.6 | **95.1** | 81.3 | **93.6** | **94.2** | **91.4** | **90.8** | **93.2** | **95.2** | **92.1** | **91.7** | 89.4 | 88.7 |
| | LTH (Augmix) | 95.6 | 89.4 | 71.5 | 80.5 | 86.0 | 85.3 | 94.6 | 80.9 | 93.1 | 93.6 | 90.5 | 90.3 | 92.5 | 94.8 | 91.4 | 91.2 | **89.5** | 88.3 |
| | EP (Layerwise Augmix) | 94.9 | 88.4 | 2.23 | 83.4 | 87.3 | 84.3 | 93.6 | 77.8 | 91.6 | 92.3 | 88.3 | 88.2 | 90.8 | 93.9 | 89.5 | 90.0 | 88.9 | 88.5 |
| | BP (Layerwise Augmix) | 94.5 | 87.8 | 2.23 | 82.3 | 86.3 | 82.6 | 93.0 | 77.4 | 90.9 | 91.9 | 87.6 | 87.4 | 90.5 | 93.5 | 89.1 | 89.7 | 87.8 | 88.1 |
| | EP (Global Augmix) | 94.3 | 87.8 | 2.23 | 83.5 | 87.1 | 82.8 | 92.9 | 78.1 | 90.9 | 91.7 | 87.4 | 87.2 | 90.7 | 93.0 | 89.6 | 89.7 | 88.6 | 88.3 |
| | BP (Global Augmix) | 93.9 | 87.5 | 2.23 | 82.9 | 86.6 | 83.3 | 92.4 | 77.8 | 90.3 | 91.2 | 87.0 | 87.1 | 89.4 | 92.7 | 89.1 | 89.1 | 88.6 | 87.9 |
| | LRR (Gaussian) | 93.8 | 86.4 | 71.5 | **91.7** | **92.2** | **89.1** | 87.9 | **81.8** | 83.2 | 86.7 | 87.5 | 88.8 | 79.8 | 92.4 | 68.8 | 87.5 | 88.4 | **90.9** |
| | LTH (Gaussian) | 93.5 | 85.8 | 71.5 | 92.0 | 92.0 | 88.5 | 87.0 | 79.0 | 82.5 | 86.0 | 86.7 | 87.9 | 80.9 | 92.1 | 69.5 | 87.2 | 88.7 | 90.4 |
| | EP (Layerwise Gaussian) | 93.0 | 85.7 | 2.23 | 91.6 | 91.4 | 88.5 | 86.6 | 81.6 | 81.6 | 85.7 | 86.2 | 87.9 | 80.8 | 91.0 | 71.1 | 85.9 | 88.7 | 89.8 |
| | BP (Layerwise Gaussian) | 93.2 | 85.3 | 2.23 | 90.7 | 91.1 | 88.5 | 86.0 | 79.9 | 79.9 | 84.9 | 86.2 | 87.6 | 82.1 | 91.5 | 85.7 | 85.7 | 87.4 | 89.5 |
| | EP (Global Gaussian) | 93.2 | 85.1 | 2.23 | 90.3 | 90.8 | 88.5 | 85.8 | 80.3 | 80.6 | 84.5 | 85.2 | 86.8 | 81.8 | 90.0 | 70.3 | 85.5 | 87.5 | 89.2 |
| | BP (Global Gaussian) | 92.4 | 85.1 | 2.23 | 90.4 | 90.9 | 88.6 | 86.2 | 81.4 | 81.0 | 85.2 | 85.1 | 87.2 | 80.4 | 90.2 | 68.9 | 85.5 | 88.4 | 89.4 |
| | EP (Global Gaussian) | 92.4 | 85.1 | 2.23 | 91.7 | 91.6 | 88.6 | 86.2 | 81.8 | 81.0 | 85.2 | 85.1 | 87.5 | 79.8 | 92.7 | 86.8 | 89.1 | 88.6 | 89.4 |
| | EP (Global Clean) | 93.7 | 76.7 | 2.23 | 71.6 | 71.6 | 64.4 | 82.8 | 56.8 | 77.7 | 78.2 | 80.4 | 78.8 | 87.6 | 92.0 | 77.1 | 84.8 | 77.6 | 83.9 |
| | LRR (Clean) | 92.4 | 76.3 | 71.5 | 64.4 | 64.4 | 55.7 | 82.8 | 58.5 | 77.7 | 78.2 | 80.4 | 78.8 | 87.6 | 92.0 | 77.1 | 84.8 | 77.6 | 83.9 |
| | EP (Global Clean) | 93.7 | 76.7 | 2.23 | 71.6 | 71.6 | 64.4 | 85.2 | 56.8 | 81.0 | 85.2 | 85.1 | 80.4 | 80.4 | 90.2 | 84.8 | 85.5 | 88.4 | 89.4 |
| | BP (Global Clean) | 95.6 | 76.3 | 2.23 | 61.0 | 61.0 | 55.7 | 85.2 | 58.5 | 77.7 | 81.5 | 84.5 | 80.5 | 78.8 | 92.0 | 77.1 | 84.8 | 77.6 | 83.9 |
| | LTH (Clean) | 93.7 | 76.1 | 71.5 | 61.2 | 62.0 | 62.9 | 82.9 | 55.9 | 78.2 | 78.9 | 80.5 | 79.7 | 88.6 | 91.7 | 75.7 | 84.4 | 76.6 | 81.5 |
| | EP (Layerwise Clean) | 94.9 | 74.4 | 2.23 | 59.3 | 62.0 | 51.8 | 82.8 | 55.9 | 78.8 | 78.9 | 83.2 | 79.7 | 88.8 | 93.5 | 78.7 | 85.7 | 77.9 | 80.7 |
| | BP (Layerwise Clean) | 94.1 | 74.3 | 2.23 | 53.0 | 64.1 | 56.2 | 82.8 | 50.8 | 77.4 | 78.5 | 80.1 | 76.7 | 87.9 | 92.5 | 78.2 | 83.9 | 75.6 | 81.2 |

Table 6: Performance when using ResNet-18 architecture with 80% of weights pruned evaluated on CIFAR-10 and CIFAR-10-C. Note: LTH and LRR model prune rates are 79%.

| | Model | Clean | Avg. Robust | Memory (Mbit) | Noise | | | Blur | | | | Weather | | | | Digital | | | |
|---|---|---|---|---|---|---|---|---|---|---|---|---|---|---|---|---|---|---|---|
| | | | | | gaussian | shot | impulse | defocus | glass | motion | zoom | snow | frost | fog | brightness | contrast | elastic | pixelate | jpeg |
| **Baseline** | Dense (Augmix) | 95.5 | 89.2 | 358 | 81.7 | 85.9 | 86.7 | 94.3 | 80.8 | 92.4 | 93.2 | 90.0 | 89.7 | 92.2 | 94.7 | 91.0 | 90.5 | 88.7 | 87.7 |
| | Dense (Gaussian) | 93.9 | 85.6 | 358 | 91.3 | 91.8 | 88.3 | 85.8 | 81.0 | 80.9 | 84.9 | 86.4 | 88.3 | 81.9 | 92.3 | 70.6 | 86.6 | 88.1 | 89.9 |
| | Dense (Clean) | 95.1 | 73.7 | 358 | 46.5 | 59.1 | 54.0 | 81.8 | 55.1 | 78.1 | 76.4 | 82.4 | 78.2 | 88.2 | 93.5 | 78.2 | 84.0 | 76.1 | 79.3 |
| **CARDs** | LRR (Augmix) | **96.3** | **90.1** | 35.8 | 79.2 | 85.4 | **89.3** | **95.3** | **81.6** | **93.8** | **94.4** | **91.5** | **91.3** | **93.3** | **95.5** | **92.5** | **91.8** | 89.8 | 88.7 |
| | LTH (Augmix) | 95.7 | 89.4 | 35.8 | 79.9 | 85.5 | 86.5 | 94.8 | 81.1 | 93.2 | 93.9 | 91.0 | 90.6 | 92.8 | 94.9 | 91.7 | 91.4 | **89.9** | 88.6 |
| | EP (Global Augmix) | 94.4 | 88.0 | 1.12 | 83.2 | 87.1 | 82.9 | 93.0 | 77.6 | 91.0 | 92.0 | 87.7 | 88.0 | 90.4 | 93.4 | 88.8 | 89.6 | 88.9 | 88.2 |
| | EP (Layerwise Augmix) | 94.4 | 87.9 | 1.12 | 83.1 | 86.5 | 84.5 | 93.0 | 77.6 | 90.9 | 91.9 | 87.7 | 87.5 | 90.1 | 93.3 | 88.5 | 89.5 | 88.1 | 88.1 |
| | BP (Global Augmix) | 94.4 | 87.8 | 1.12 | 83.0 | 86.9 | 83.8 | 92.9 | 76.6 | 90.5 | 91.6 | 87.6 | 87.5 | 89.9 | 93.2 | 88.7 | 89.5 | 88.1 | 87.8 |
| | BP (Layerwise Augmix) | 93.7 | 87.1 | 1.12 | 81.9 | 85.8 | 82.3 | 92.2 | 76.8 | 90.0 | 91.0 | 87.0 | 86.5 | 89.5 | 92.5 | 86.9 | 89.0 | 87.2 | 87.8 |
| | LRR (Gaussian) | 94.0 | 86.4 | 35.8 | **91.9** | **92.5** | 89.1 | 88.0 | 81.3 | 82.9 | 86.7 | 87.8 | 88.7 | 90.9 | 92.3 | 89.0 | 87.6 | 89.6 | **91.0** |
| | LTH (Gaussian) | 93.8 | 86.0 | 35.8 | 91.8 | 92.2 | 88.5 | 87.4 | 79.2 | 82.8 | 86.6 | 87.0 | 88.4 | 91.8 | 92.4 | 86.9 | 87.4 | 89.6 | 90.3 |
| | EP (Layerwise Gaussian) | 92.8 | 85.4 | 1.12 | 90.6 | 91.1 | 88.3 | 86.2 | 81.0 | 80.9 | 85.2 | 85.6 | 87.4 | 80.9 | 90.7 | 71.2 | 85.6 | 87.7 | 89.6 |
| | EP (Global Gaussian) | 92.9 | 85.2 | 1.12 | 90.8 | 91.1 | 88.7 | 86.4 | 80.5 | 80.8 | 85.2 | 85.4 | 87.0 | 80.9 | 90.8 | 69.8 | 85.8 | 87.4 | 89.4 |
| | BP (Global Gaussian) | 92.7 | 85.0 | 1.12 | 90.4 | 90.9 | 88.5 | 85.1 | 80.2 | 79.8 | 83.9 | 87.0 | 80.9 | 80.9 | 90.4 | 71.1 | 85.2 | 86.6 | 89.2 |
| | BP (Layerwise Gaussian) | 93.0 | 84.2 | 1.12 | 89.9 | 90.4 | 87.0 | 85.2 | 79.5 | 79.5 | 83.8 | 84.6 | 87.5 | 81.5 | 91.1 | 69.3 | 84.5 | 86.6 | 88.9 |
| | LRR (Clean) | 95.6 | 76.6 | 35.8 | 47.6 | 60.5 | 55.2 | 85.1 | 61.1 | 82.0 | 81.4 | 84.9 | 81.2 | 90.3 | 94.3 | 81.4 | 86.4 | 79.7 | 81.3 |
| | EP (Layerwise Clean) | 94.1 | 76.0 | 1.12 | 60.7 | 69.3 | 62.5 | 83.5 | 52.7 | 77.9 | 79.1 | 79.8 | 78.1 | 87.5 | 92.3 | 78.0 | 84.2 | 75.7 | 82.1 |
| | EP (Global Clean) | 93.9 | 75.6 | 1.12 | 55.5 | 66.1 | 59.9 | 84.2 | 52.3 | 78.6 | 80.4 | 81.1 | 77.9 | 88.4 | 92.4 | 79.1 | 84.1 | 76.7 | 82.3 |
| | BP (Global Clean) | 94.1 | 75.1 | 1.12 | 56.9 | 66.8 | 62.0 | 82.6 | 54.3 | 77.2 | 77.9 | 80.6 | 78.5 | 87.9 | 92.2 | 76.9 | 83.4 | 74.9 | 81.3 |
| | BP (Layerwise Clean) | 93.6 | 74.6 | 1.12 | 54.2 | 64.7 | 59.3 | 83.0 | 50.8 | 77.0 | 78.2 | 79.5 | 76.6 | 87.5 | 91.9 | 77.0 | 83.5 | 75.0 | 81.8 |
| | LTH (Clean) | 95.2 | 74.5 | 35.8 | 45.5 | 58.4 | 53.2 | 83.5 | 56.1 | 79.6 | 79.9 | 83.3 | 79.4 | 89.1 | 93.9 | 78.6 | 85.6 | 77.6 | 80.4 |

Table 7: Performance when using ResNet-18 architecture with 90% of weights pruned evaluated on CIFAR-10 and CIFAR-10-C. Note: LTH and LRR model prune rates are 89%.

| Model | Clean | Avg. Robust | Memory (Mbit) | Noise | | | Blur | | | | Weather | | | | Digital | | | |
|---|---|---|---|---|---|---|---|---|---|---|---|---|---|---|---|---|---|---|
| | | | | gaussian | shot | impulse | defocus | glass | motion | zoom | snow | frost | fog | brightness | contrast | elastic | pixelate | jpeg |
| **Baseline** | | | | | | | | | | | | | | | | | | |
| Dense (Augmix) | 95.5 | 89.2 | 358 | 81.7 | 85.9 | 86.7 | 94.3 | 80.8 | 92.4 | 93.2 | 90.0 | 89.7 | 92.2 | 94.7 | 90.5 | 91.0 | 88.7 | 87.7 |
| Dense (Gaussian) | 93.9 | 85.6 | 358 | 91.3 | 91.8 | 88.3 | 85.8 | 81.0 | 80.9 | 84.9 | 86.4 | 88.3 | 81.9 | 92.3 | 70.6 | 86.6 | 88.1 | 89.9 |
| Dense (Clean) | 95.1 | 73.7 | 358 | 46.5 | 59.1 | 54.0 | 81.8 | 55.1 | 78.1 | 76.4 | 82.4 | 78.2 | 88.2 | 93.5 | 78.2 | 84.0 | 76.1 | 79.3 |
| **CARDs** | | | | | | | | | | | | | | | | | | |
| LRR (Augmix) | **96.1** | **90.1** | 17.9 | 79.2 | 85.3 | **89.6** | **95.2** | **81.5** | **93.8** | **94.3** | **91.6** | **91.4** | **93.3** | **95.4** | **92.4** | **91.8** | **89.7** | 88.8 |
| LTH (Augmix) | 95.9 | 89.5 | 17.9 | 78.0 | 84.5 | 87.0 | 94.7 | 81.1 | 93.1 | 93.8 | 90.9 | 90.5 | 92.8 | 94.9 | 91.5 | 91.6 | 89.3 | 88.4 |
| EP (Global Augmix) | 94.5 | 87.8 | 0.56 | 82.1 | 86.5 | 83.1 | 93.2 | 77.1 | 90.9 | 92.0 | 87.8 | 87.7 | 90.3 | 93.3 | 88.5 | 89.5 | 88.1 | 87.5 |
| BP (Global Augmix) | 94.2 | 87.5 | 0.56 | 82.0 | 86.1 | 83.4 | 92.8 | 76.5 | 90.2 | 91.5 | 87.2 | 87.4 | 89.8 | 93.0 | 87.9 | 89.2 | 87.5 | 87.4 |
| LRR (Gaussian) | 94.0 | 86.5 | 17.9 | **91.9** | **92.3** | 88.9 | 87.7 | 80.4 | 83.3 | 86.7 | 87.7 | 88.8 | 81.1 | 92.6 | 70.2 | 87.8 | 89.2 | **90.7** |
| LTH (Gaussian) | 93.8 | 86.0 | 17.9 | 91.7 | 92.1 | 88.4 | 87.5 | 77.6 | 82.9 | 86.5 | 86.7 | 87.8 | 82.2 | 92.3 | 70.5 | 87.5 | 87.6 | 90.3 |
| EP (Layerwise Augmix) | 93.2 | 85.7 | 0.56 | 81.7 | 85.1 | 82.9 | 91.3 | 73.5 | 88.2 | 89.8 | 85.6 | 85.4 | 88.4 | 91.8 | 85.5 | 87.7 | 85.4 | 86.9 |
| EP (Global Gaussian) | 92.6 | 84.3 | 0.56 | 90.3 | 90.7 | 88.2 | 85.2 | 78.6 | 78.9 | 84.2 | 85.1 | 86.6 | 80.7 | 90.2 | 69.1 | 84.8 | 85.7 | 89.0 |
| BP (Layerwise Augmix) | 92.5 | 84.7 | 0.56 | 80.6 | 84.2 | 81.3 | 90.7 | 73.0 | 87.4 | 89.0 | 84.1 | 83.6 | 87.2 | 90.9 | 83.6 | 86.9 | 85.4 | 86.8 |
| BP (Global Gaussian) | 92.5 | 84.4 | 0.56 | 90.1 | 90.7 | 88.2 | 85.3 | 78.8 | 79.0 | 84.0 | 85.0 | 86.5 | 80.7 | 90.2 | 69.1 | 84.9 | 85.6 | 89.0 |
| EP (Layerwise Gaussian) | 91.2 | 83.9 | 0.56 | 88.8 | 89.6 | 86.5 | 85.6 | 79.5 | 79.6 | 84.5 | 84.2 | 85.7 | 79.8 | 89.2 | 67.9 | 84.4 | 85.8 | 88.2 |
| BP (Layerwise Gaussian) | 91.4 | 83.1 | 0.56 | 88.6 | 89.2 | 85.8 | 84.8 | 78.5 | 79.0 | 83.4 | 83.8 | 85.3 | 80.9 | 89.7 | 67.3 | 83.6 | 85.5 | 87.9 |
| LRR (Clean) | 95.7 | 75.8 | 17.9 | 46.1 | 59.4 | 56.4 | 85.0 | 58.9 | 81.1 | 81.0 | 84.5 | 80.3 | 89.5 | 94.4 | 80.9 | 86.3 | 79.5 | 80.7 |
| LTH (Clean) | 95.2 | 75.1 | 17.9 | 48.0 | 60.8 | 57.2 | 83.2 | 56.0 | 78.0 | 78.7 | 83.4 | 79.8 | 88.9 | 93.9 | 79.1 | 85.3 | 78.3 | 80.9 |
| BP (Global Clean) | 93.8 | 73.8 | 0.56 | 49.9 | 61.8 | 53.7 | 83.2 | 53.7 | 77.0 | 78.1 | 79.7 | 76.8 | 87.6 | 92.1 | 78.4 | 82.9 | 74.9 | 80.7 |
| EP (Global Clean) | 92.7 | 73.4 | 0.56 | 56.1 | 64.9 | 63.3 | 82.1 | 48.6 | 76.9 | 77.2 | 78.1 | 73.5 | 86.3 | 90.9 | 77.2 | 82.4 | 73.8 | 81.4 |
| BP (Layerwise Clean) | 93.8 | 73.8 | 0.56 | 61.8 | 69.3 | 64.1 | 82.2 | 51.1 | 77.0 | 78.7 | 79.7 | 76.8 | 87.6 | 92.1 | 78.4 | 82.9 | 74.9 | 80.7 |
| EP (Layerwise Clean) | 92.1 | 73.4 | 0.56 | 61.4 | 60.2 | 64.1 | 80.0 | 51.5 | 74.6 | 74.0 | 77.8 | 74.5 | 85.6 | 90.1 | 72.7 | 81.5 | 72.2 | 81.5 |
| EP (Global Clean) | 94.0 | 73.1 | 0.56 | 49.3 | 60.2 | 57.6 | 82.0 | 52.5 | 75.7 | 77.6 | 79.2 | 75.9 | 87.0 | 92.1 | 74.8 | 82.9 | 74.3 | 80.6 |

Table 8: Performance when using ResNet-18 architecture with 95% of weights pruned evaluated on CIFAR-10 and CIFAR-10-C.

| Model | Clean | Avg. Robust | Memory (Mbit) | Noise | | | Blur | | | | Weather | | | | Digital | | | |
|---|---|---|---|---|---|---|---|---|---|---|---|---|---|---|---|---|---|---|
| | | | | gaussian | shot | impulse | defocus | glass | motion | zoom | snow | frost | fog | brightness | contrast | elastic | pixelate | jpeg |
| **Baseline** | | | | | | | | | | | | | | | | | | |
| Dense (Augmix) | 95.5 | 89.2 | 358 | 81.7 | 85.9 | 86.7 | 94.3 | 80.8 | 92.4 | 93.2 | 90.0 | 89.7 | 92.2 | 94.7 | 91.0 | 90.5 | 88.7 | 87.7 |
| Dense (Gaussian) | 93.9 | 85.6 | 358 | 91.3 | 91.8 | 88.3 | 85.8 | 81.0 | 80.9 | 84.9 | 86.4 | 88.3 | 81.9 | 92.3 | 70.6 | 86.6 | 88.1 | 89.9 |
| Dense (Clean) | 95.1 | 73.7 | 358 | 46.5 | 59.1 | 54.0 | 81.8 | 55.1 | 78.1 | 76.4 | 82.4 | 78.2 | 88.2 | 93.5 | 78.2 | 84.0 | 76.1 | 79.3 |
| **CARD-Deck (Agnostic)** | | | | | | | | | | | | | | | | | | |
| LRR (6) | **96.4** | **91.9** | 429 | 88.8 | **91.5** | **91.6** | **95.1** | **84.7** | 93.3 | 94.3 | **92.2** | 92.3 | 92.3 | **95.5** | 90.5 | **92.6** | **91.6** | **92.1** |
| LRR (4) | 96.3 | 91.7 | 286 | 87.1 | 90.5 | 91.0 | **95.1** | 84.6 | 93.4 | **94.4** | **92.2** | **92.3** | 92.6 | **95.5** | 91.1 | 92.5 | 91.2 | 91.7 |
| LTH (6) | 96.1 | 91.2 | 429 | 88.8 | **91.5** | 89.8 | 94.4 | 83.6 | 92.5 | 93.6 | 91.4 | 91.6 | 91.6 | 95.1 | 89.6 | 92.1 | 90.9 | 91.6 |
| LTH (4) | 96.0 | 91.0 | 286 | 87.7 | 90.6 | 88.9 | 94.5 | 83.4 | 92.7 | 93.8 | 91.4 | 91.5 | 91.9 | 95.0 | 90.3 | 92.0 | 90.7 | 91.1 |
| LRR (2) | 96.0 | 89.7 | 143 | 77.9 | 84.3 | 87.1 | **95.1** | 81.3 | **93.6** | 94.1 | 91.4 | 90.8 | **93.1** | 95.2 | **91.9** | 91.5 | 89.2 | 88.7 |
| LTH (2) | 95.5 | 89.4 | 143 | 80.5 | 86.0 | 84.9 | 94.6 | 80.9 | 93.0 | 93.6 | 90.5 | 90.2 | 92.4 | 94.8 | 91.4 | 91.1 | 89.5 | 87.9 |
| BP (6) | 94.3 | 89.0 | 13.4 | **89.7** | 91.2 | 88.9 | 91.9 | 82.2 | 89.1 | 90.6 | 88.3 | 89.2 | 88.2 | 93.0 | 83.1 | 89.8 | 89.4 | 90.4 |
| BP (4) | 94.3 | 88.8 | 8.94 | 88.9 | 90.6 | 88.1 | 92.0 | 81.4 | 89.4 | 90.8 | 88.1 | 88.9 | 88.4 | 92.9 | 83.9 | 89.7 | 88.9 | 90.1 |
| BP (2) | 93.8 | 87.5 | 4.47 | 82.9 | 86.6 | 83.3 | 92.4 | 77.8 | 90.3 | 91.2 | 87.0 | 87.1 | 89.1 | 92.7 | 86.8 | 89.1 | 87.7 | 87.9 |

Table 9: Performance of data-agnostic CARD-Deck ensembles composed of 80% sparse ResNet-18 CARDs on individual CIFAR-10-C corruptions. For each corruption, entry is average over severity levels 1 through 5.

| Model | Clean | Avg. Robust | Memory (Mbit) | Noise | | | Blur | | | | Weather | | | | Digital | | | |
|---|---|---|---|---|---|---|---|---|---|---|---|---|---|---|---|---|---|---|
| | | | | gaussian | shot | impulse | defocus | glass | motion | zoom | snow | frost | fog | brightness | contrast | elastic | pixelate | jpeg |
| **CARD-Deck (Agnostic)** | | | | | | | | | | | | | | | | | | |
| LRR (6) | **96.4** | **92.0** | 215 | 89.4 | **91.9** | **91.9** | 95.0 | **85.0** | 93.3 | 94.2 | **92.2** | **92.4** | 92.2 | **95.6** | 90.5 | **92.6** | **91.5** | **92.1** |
| LRR (4) | **96.4** | 91.7 | 143 | 87.6 | 90.8 | 91.2 | 95.1 | 84.4 | 93.5 | 94.3 | 92.1 | 92.1 | 92.6 | 95.5 | 91.1 | 92.3 | 91.1 | 91.6 |
| LTH (6) | 96.2 | 91.3 | 215 | 88.9 | 91.6 | 89.8 | 94.4 | 83.7 | 92.5 | 93.7 | 91.6 | 91.8 | 91.7 | 95.2 | 89.8 | 92.1 | 91.0 | 91.7 |
| LTH (4) | 95.9 | 91.0 | 143 | 87.4 | 90.5 | 88.3 | 94.5 | 83.1 | 92.9 | 93.8 | 91.6 | 91.6 | 92.1 | 95.2 | 90.4 | 92.0 | 90.7 | 91.1 |
| LRR (2) | 96.3 | 89.8 | 71.5 | 77.7 | 84.3 | 87.2 | **95.3** | 81.4 | **93.8** | **94.4** | 91.4 | 91.0 | **93.1** | 95.5 | **91.9** | 91.7 | 89.8 | 88.7 |
| LTH (2) | 95.7 | 89.4 | 71.5 | 79.9 | 85.5 | 83.2 | 94.7 | 81.1 | 93.2 | 93.9 | 90.8 | 90.6 | 92.5 | 94.9 | 91.7 | 91.2 | 89.9 | 87.8 |
| BP (6) | 94.5 | 89.4 | 6.70 | **89.9** | 91.5 | 89.1 | 92.1 | 81.9 | 89.3 | 91.1 | 89.0 | 89.7 | 88.8 | 93.5 | 85.1 | 90.0 | 89.5 | 90.7 |
| BP (4) | 94.6 | 89.2 | 4.47 | 88.8 | 90.7 | 88.2 | 92.3 | 81.0 | 89.7 | 91.2 | 88.7 | 89.4 | 89.2 | 93.4 | 85.7 | 90.0 | 89.3 | 90.2 |
| BP (2) | 94.0 | 87.4 | 2.23 | 82.0 | 86.3 | 82.7 | 92.5 | 76.5 | 90.3 | 91.5 | 87.3 | 86.8 | 89.7 | 92.8 | 87.8 | 89.5 | 88.1 | 87.6 |
| **Baseline** | | | | | | | | | | | | | | | | | | |
| Dense (Augmix) | 95.5 | 89.2 | 358 | 81.7 | 85.9 | 86.7 | 94.3 | 80.8 | 92.4 | 93.2 | 90.0 | 89.7 | 92.2 | 94.7 | 91.0 | 90.5 | 88.7 | 87.7 |
| Dense (Gaussian) | 93.9 | 85.6 | 358 | 91.3 | 91.8 | 88.3 | 85.8 | 81.0 | 80.9 | 84.9 | 86.4 | 88.3 | 81.9 | 92.3 | 70.6 | 86.6 | 88.1 | 89.9 |
| Dense (Clean) | 95.1 | 73.7 | 358 | 46.5 | 59.1 | 54.0 | 81.8 | 55.1 | 78.1 | 76.4 | 82.4 | 78.2 | 88.2 | 93.5 | 78.2 | 84.0 | 76.1 | 79.3 |

Table 10: Accuracy of data-agnostic `CARD-Deck` ensembles composed of 90% sparse ResNet-18 `CARD`s on individual CIFAR-10-C corruptions. For each corruption, entry is average over severity levels 1 through 5.

| | Model | Clean | Avg. Robust | Memory (Mbit) | Noise | | | Blur | | | | Weather | | | | Digital | | | |
|---|---|---|---|---|---|---|---|---|---|---|---|---|---|---|---|---|---|---|---|
| | | | | | gaussian | shot | impulse | defocus | glass | motion | zoom | snow | frost | fog | brightness | contrast | elastic | pixelate | jpeg |
| **Baseline** | Dense (Augmix) | 95.5 | 89.2 | 358 | 81.7 | 85.9 | 86.7 | 94.3 | 80.8 | 92.4 | 93.2 | 90.0 | 89.7 | 92.2 | 94.7 | 91.0 | 90.5 | 88.7 | 87.7 |
| | Dense (Gaussian) | 93.9 | 85.6 | 358 | 91.3 | 91.8 | 88.3 | 85.8 | 81.0 | 80.9 | 84.9 | 86.4 | 88.3 | 81.9 | 92.3 | 70.6 | 86.6 | 88.1 | 89.9 |
| | Dense (Clean) | 95.1 | 73.7 | 358 | 46.5 | 59.1 | 54.0 | 81.8 | 55.1 | 78.1 | 76.4 | 82.4 | 78.2 | 88.2 | 93.5 | 78.2 | 84.0 | 76.1 | 79.3 |
| **CARD-Deck (Agnostic)** | LRR (6) | **96.3** | **91.9** | 107 | 89.3 | **91.6** | **91.9** | 94.9 | **84.5** | 93.2 | 94.1 | **92.2** | **92.2** | 92.1 | **95.6** | 90.3 | **92.4** | **91.5** | **92.0** |
| | LRR (4) | **96.3** | 91.6 | 71.5 | 87.0 | 90.3 | 91.3 | 95.0 | 84.0 | 93.4 | 94.2 | **92.2** | **92.2** | 92.6 | 95.5 | 90.8 | **92.4** | 91.4 | 91.6 |
| | LTH (6) | 96.1 | 91.1 | 107 | 89.0 | **91.6** | 90.1 | 94.3 | 83.2 | 92.3 | 93.6 | 91.5 | 91.4 | 91.7 | 95.1 | 89.4 | 92.0 | 90.5 | 91.4 |
| | LTH (4) | 96.0 | 90.8 | 71.5 | 87.3 | 90.6 | 88.8 | 94.5 | 82.8 | 92.7 | 93.6 | 91.3 | 91.3 | 91.9 | 95.0 | 90.1 | 92.0 | 90.0 | 90.8 |
| | LRR (2) | 96.1 | 89.6 | 35.8 | 75.8 | 83.4 | 86.8 | **95.2** | 80.9 | **93.8** | **94.3** | 91.5 | 90.9 | **93.3** | 95.4 | **91.9** | 91.8 | 88.8 | 88.7 |
| | BP (6) | 94.7 | 89.2 | 3.35 | **89.5** | 91.2 | 88.7 | 92.4 | 81.0 | 89.2 | 91.3 | 88.7 | 89.5 | 88.7 | 93.5 | 84.8 | 89.9 | 88.8 | 90.6 |
| | LTH (2) | 95.8 | 89.0 | 35.8 | 77.8 | 84.5 | 83.6 | 94.6 | 80.5 | 93.1 | 93.6 | 91.5 | 91.3 | 91.9 | 95.0 | 90.1 | 92.0 | 90.0 | 90.8 |
| | BP (4) | 94.7 | 88.9 | 2.23 | 88.3 | 90.5 | 87.8 | 92.4 | 80.3 | 89.3 | 91.2 | 88.4 | 89.1 | 88.9 | 93.3 | 85.4 | 89.7 | 89.3 | 87.7 |
| | BP (2) | 94.0 | 87.2 | 1.12 | 81.2 | 85.7 | 82.5 | 92.5 | 76.0 | 89.9 | 91.5 | 87.0 | 87.1 | 89.7 | 92.8 | 87.6 | 88.9 | 87.4 | 87.4 |

Table 11: Accuracy of data-agnostic CARD-Deck ensembles composed of 95% sparse ResNet-18 CARDs on individual CIFAR-10-C corruptions. For each corruption, entry is average over severity levels 1 through 5.

| | | Baseline | | | ResNet-18 | | | | | |
| | | Dense | | | FT | | | GMP | | |
| | | - | | | Global | | | Global | | |
| | | Augmix | Clean | Gauss. | Augmix | Clean | Gauss. | Augmix | Clean | Gauss. |
|---|---|---|---|---|---|---|---|---|---|---|
| 80% | Clean Acc. | 95.5 | 95.1 | 93.9 | 95.6 | 94.6 | 94.1 | 95.1 | 94.3 | 93.4 |
| | Robust Acc. | 89.2 | 73.7 | 85.6 | 88.7 | 73.6 | 84.5 | 88.2 | 73 | 84.5 |
| | Memory (Mbit) | 358 | 358 | 358 | 143 | 286 | 429 | 143 | 286 | 429 |
| 90% | Clean Acc. | 95.5 | 95.1 | 93.9 | 94.7 | 94.1 | 93.1 | 94.8 | 94 | 93.4 |
| | Robust Acc. | 89.2 | 73.7 | 85.6 | 87.2 | 71.5 | 83.5 | 87.7 | 72.9 | 83.7 |
| | Memory (Mbit) | 358 | 358 | 358 | 71.5 | 143 | 215 | 71.5 | 143 | 215 |
| 95% | Clean Acc. | 95.5 | 95.1 | 93.9 | 93.8 | 93.4 | 92.2 | 94.6 | 93.9 | 92.9 |
| | Robust Acc. | 89.2 | 73.7 | 85.6 | 85.4 | 68 | 81.1 | 87.2 | 73 | 83.7 |
| | Memory (Mbit) | 358 | 358 | 358 | 35.8 | 71.5 | 107 | 35.8 | 71.5 | 107 |

Table 12: Performance comparison between dense baselines and ResNet-18 models pruned using fine-tuning (FT) and gradual magnitude pruning (GMP). Clean and Robust Acc. refer to accuracy on CIFAR-10 and CIFAR-10-C, respectively.

| | | Baseline | | | CARD | | | | | | | | | | | | | | | | | | |
| | | Dense | | | Edgepopup | | | | | | LRR | | | LTH | | | Biprop | | | | | |
| | | - | | | Layerwise | | | Global | | | Global | | | Global | | | Layerwise | | | Global | | |
| | | Augmix | Clean | Gauss. | Augmix | Clean | Gauss. | Augmix | Clean | Gauss. | Augmix | Clean | Gauss. | Augmix | Clean | Gauss. | Augmix | Clean | Gaussian | Augmix | Clean | Gauss. |
|---|---|---|---|---|---|---|---|---|---|---|---|---|---|---|---|---|---|---|---|---|---|---|
| 90% | Clean Acc. | 95.9 | 95.4 | 93.8 | 94.2 | 94.2 | 93 | 94.4 | 94 | 92.9 | 96.4 | 95.8 | 94.1 | 96.2 | 95.3 | 93.9 | 94.2 | 94 | 92.8 | 93.7 | 93.6 | 92.4 |
| | Robust Acc. | 88.2 | 71.2 | 84.3 | 86.5 | 72 | 83 | 86.7 | 72.1 | 83 | 89.1 | 72.4 | 85.2 | 88.8 | 71.5 | 84.2 | 86.2 | 71.5 | 83.1 | 86 | 71.5 | 82.7 |
| | Memory (Mbit) | 153 | 153 | 153 | 0.48 | 0.48 | 0.48 | 0.48 | 0.48 | 0.48 | 15.28 | 15.28 | 15.28 | 15.28 | 15.28 | 15.28 | 0.48 | 0.48 | 0.48 | 0.48 | 0.48 | 0.48 |
| 95% | Clean Acc. | 95.9 | 95.4 | 93.8 | 93.7 | 93.7 | 93 | 91.7 | 91.5 | 90.4 | 96.2 | 95.6 | 94.1 | 96.1 | 95.1 | 93.8 | 93.6 | 93.5 | 92.4 | 90.9 | 91.2 | 90.2 |
| | Robust Acc. | 88.2 | 71.2 | 84.3 | 85.8 | 69.9 | 82.3 | 82.8 | 69 | 80.3 | 89.3 | 72.1 | 85.2 | 88.4 | 71.1 | 84.3 | 85.1 | 70.3 | 81.8 | 82.4 | 70.4 | 80.2 |
| | Memory (Mbit) | 153 | 153 | 153 | 0.24 | 0.24 | 0.24 | 0.24 | 0.24 | 0.24 | 7.64 | 7.64 | 7.64 | 7.64 | 7.64 | 7.64 | 0.24 | 0.24 | 0.24 | 0.24 | 0.24 | 0.24 |

Table 13: Performance comparison between dense baselines and CARDs for ResNeXt-29. Clean and Robust Acc. refer to accuracy on CIFAR-10 and CIFAR-10-C, respectively.

| | | Baseline | | | CARD | | | | | | | | | | | | | | | | | | |
| | | Dense | | | Edgepopup | | | | | | LRR | | | LTH | | | Biprop | | | | | |
| | | - | | | Layerwise | | | Global | | | Global | | | Global | | | Layerwise | | | Global | | |
| | | Augmix | Clean | Gauss. | Augmix | Clean | Gauss. | Augmix | Clean | Gauss. | Augmix | Clean | Gauss. | Augmix | Clean | Gauss. | Augmix | Clean | Gaussian | Augmix | Clean | Gauss. |
|---|---|---|---|---|---|---|---|---|---|---|---|---|---|---|---|---|---|---|---|---|---|---|
| 90% | Clean Acc. | 95.9 | 95.1 | 93.6 | 94.8 | 94.4 | 93.3 | 95.4 | 95 | 94 | 96.4 | 95.9 | 94.7 | 95.8 | 95.4 | 94.2 | 94.7 | 94.4 | 93 | 94.9 | 94.7 | 93.2 |
| | Robust Acc. | 88.9 | 74.4 | 85.2 | 88 | 75.8 | 84.8 | 89.1 | 76.6 | 86.3 | 90.3 | 76.1 | 86.9 | 89.8 | 75.6 | 86 | 87.9 | 74.6 | 85.1 | 88.6 | 75.6 | 85.6 |
| | Memory (Mbit) | 753 | 753 | 753 | 2.35 | 2.35 | 2.35 | 2.35 | 2.35 | 2.35 | 75.27 | 75.27 | 75.27 | 75.27 | 75.27 | 75.27 | 2.35 | 2.35 | 2.35 | 2.35 | 2.35 | 2.35 |
| 95% | Clean Acc. | 95.9 | 95.1 | 93.6 | 92.4 | 93.6 | 92.2 | 95 | 94.8 | 93.4 | 96.4 | 95.8 | 94.7 | 95.8 | 95.4 | 94.2 | 94.6 | 94.6 | 93.2 | 94.4 | 93.9 | 92.6 |
| | Robust Acc. | 88.9 | 74.4 | 85.2 | 79.2 | 70.9 | 82.6 | 88.9 | 76.6 | 85.9 | 90.4 | 75.7 | 86.8 | 89.6 | 75.3 | 86 | 87.4 | 73.9 | 84.8 | 87.6 | 74.4 | 84.6 |
| | Memory (Mbit) | 753 | 753 | 753 | 1.18 | 1.18 | 1.18 | 1.18 | 1.18 | 1.18 | 37.63 | 37.63 | 37.63 | 37.63 | 37.63 | 37.63 | 1.18 | 1.18 | 1.18 | 1.18 | 1.18 | 1.18 |

Table 14: Performance comparison between dense baselines and CARDs for ResNet-50. Clean and Robust Acc. refer to accuracy on CIFAR-10 and CIFAR-10-C, respectively.

| | | Baseline | | | CARD | | | | | | | | | | | | | | | | |
|---|---|---|---|---|---|---|---|---|---|---|---|---|---|---|---|---|---|---|---|---|---|
| | | Dense | | | Edgepopup | | | | | | LRR | | | LTH | | | Biprop | | | | | |
| | | - | | | Layerwise | | | Global | | | Global | | | Global | | | Layerwise | | | Global | | |
| | | Augmix | Clean | Gauss. | Augmix | Clean | Gauss. | Augmix | Clean | Gauss. | Augmix | Clean | Gauss. | Augmix | Clean | Gauss. | Augmix | Clean | Gaussian | Augmix | Clean | Gauss. |
| 90% | Clean Acc. | 95.6 | 95.2 | 93.3 | 94.8 | 94.2 | 92.4 | 95.0 | 94.7 | 92.8 | 96.4 | 95.9 | 94 | 96.1 | 95.6 | 94.2 | 94.5 | 94.4 | 92.7 | 94.6 | 94.1 | 92.6 |
| | Robust Acc. | 89.3 | 74.6 | 85.7 | 88.6 | 78.2 | 85.5 | 88.8 | 76.5 | 85.7 | 90.5 | 76.8 | 86.6 | 89.8 | 75.5 | 86.5 | 88.4 | 77.4 | 85.6 | 88.3 | 75.7 | 85.3 |
| | Memory (Mbit) | 1429 | 1429 | 1429 | 4.47 | 4.47 | 4.47 | 4.47 | 4.47 | 4.47 | 142.9 | 142.9 | 142.9 | 142.9 | 142.9 | 142.9 | 4.47 | 4.47 | 4.47 | 4.47 | 4.47 | 4.47 |
| 95% | Clean Acc. | 95.6 | 95.2 | 93.3 | 94.9 | 94.4 | 92.8 | 94.2 | 93.8 | 92.1 | 96.5 | 95.8 | 94.2 | 96.1 | 95.8 | 94.3 | 94.7 | 94.3 | 92.7 | 93.6 | 93.3 | 91.5 |
| | Robust Acc. | 89.3 | 74.6 | 85.7 | 88.7 | 75.6 | 85.2 | 87.6 | 75.9 | 85.0 | 90.8 | 77.1 | 87.0 | 90.0 | 76.2 | 86.6 | 88.6 | 76.1 | 84.9 | 87.3 | 76.0 | 84.6 |
| | Memory (Mbit) | 1429 | 1429 | 1429 | 2.23 | 2.23 | 2.23 | 2.23 | 2.23 | 2.23 | 71.46 | 71.46 | 71.46 | 71.46 | 71.46 | 71.46 | 2.23 | 2.23 | 2.23 | 2.23 | 2.23 | 2.23 |

Table 15: Performance comparison between dense baselines and `CARDs` for WideResNet-18 (2x). Clean and Robust Acc. refer to accuracy on CIFAR-10 and CIFAR-10-C, respectively.

| | | CARD-Deck (Agnostic/Adaptive) | | | | | |
|---|---|---|---|---|---|---|---|
| | | Biprop (Layerwise) | | | Edgepopup (Layerwise) | | |
| | | 2 | 4 | 6 | 2 | 4 | 6 |
| 90% | Clean Acc. | 92.3/ 93.0 | 94.0/ 94.5 | 94.8/**95.1** | 92.6/ 93.3 | 94.3/ 95.0 | 95.2/**95.4** |
| | Robust Acc. | 85.2/ 86.2 | 88.4/ 89.8 | 89.9/**90.5** | 85.3/ 86.6 | 89.0/ 90.4 | 90.6/**91.0** |
| | Memory (Mbit) | 8.93 | 17.86 | 26.79 | 8.93 | 17.86 | 26.79 |
| 95% | Clean Acc. | 91.4/ 92.0 | 93.3/ 93.6 | 94.0/ 94.1 | 92.1/ 92.2 | 93.8/ 94.1 | 94.7/94.8 |
| | Robust Acc. | 84.6/ 85.2 | 87.4/ 88.7 | 88.8/ 89.3 | 85.0/ 85.9 | 88.1/ 89.3 | 89.7/90.0 |
| | Memory (Mbit) | 4.46 | 8.93 | 13.39 | 4.46 | 8.93 | 13.39 |

Table 16: Performance of additional domain-agnostic and domain-adaptive `CARD-Decks` containing `CARDs` using WideResNet-18 architecture.

| | Noise | | | Blur | | | | Weather | | | | Digital | | | |
|---|---|---|---|---|---|---|---|---|---|---|---|---|---|---|---|
| Augmentation | gaussian | shot | impulse | defocus | glass | motion | zoom | snow | frost | fog | brightness | contrast | elastic | pixelate | jpeg |
| Augmix | 0 | 3.2 | **0** | **89** | **0** | **100** | **96.2** | **100** | 38.6 | **100** | **99.8** | **100** | **98.2** | **99.6** | 100 |
| Gaussian | **100** | **96.8** | 100 | 11 | **100** | 0 | 3.8 | 0 | **61.4** | 0 | 0.2 | 0 | 1.8 | 0.4 | **0** |

Table 17: **Gating Function Selection by Corruption Type on CIFAR-10-C**: The `CARD-Decks` make use of models trained on Augmix and Gaussian augmented datasets. Here we provide the percentage of CIFAR-10-C test data that was gated to the Augmix and Gaussian models in the `CARD-Deck` based on the type of corruption. For each corruption type, the bold number indicates which `CARD` achieves higher performance (on average) on that corruption. This highlights that our spectral similarity based gating function typically selects the best performing model. Note that for impulse and glass corruptions, the best performing models vary between those trained on Augmix and Gaussian corruptions based on the sparsity level of the model. For each corruption type, the percentage in the table gated to each augmentation method is an average over the gating percentages on CIFAR-10-C corruption severity levels 1 through 5.