# OpenReview forum: "A Winning Hand: Compressing Deep Networks Can Improve Out-of-Distribution Robustness"
_NeurIPS.cc/2021/Conference — NeurIPS 2021 Poster_

### Official Review · Reviewer_kbDD · 2021-06-25

**Rating:** 6
**Confidence:** 4

**Summary:**

The authors evaluate several pruning techniques and their impact on accuracy and robustness on CIFAR10. They perform a spectral analysis of Fourier domain errors and attempt to link those to OOD robustness. They propose an ensembling approach based on the spectral properties of unlabeled test images.

**Limitations And Societal Impact:**

The authors discussed limitations of their work. They did not discuss potential negative societal impacts of their work, but I would not expect this paper to lead to such.

**Main Review:**

### Pros:

I think the question of achieving robustness and efficiency at the same time is interesting and very relevant. I also think that this is an under-explored question and agree with the authors’ statement in line 53: “Perhaps surprisingly, the current solution in the robust ML community to improve the OOD robustness (and accuracy) is to increase the model size (e.g., [21, 13] and [6]).” It would be great if we had small and efficient models that are robust and accurate.

### Cons:

The authors oversell their results in all performed experiments. They speak of “drastic” gains, “dramatically more robust” models etc. Looking at the results, the robustness gains are on the order of 6-8 percent points for the smaller models and on the order of 1-2 percent points for the larger and thus, more interesting models. Given that their findings do not produce significant gains for their best model, I doubt that this analysis would actually scale to ImageNet. Since their small models with the larger gains have around 88% clean accuracy on CIFAR10, I do not find an analysis on those models particularly interesting, because the clean result is very far from the state of the art on CIFAR10, and the ultimate goal of the authors is to show a new state of the art on CIFAR10. Thus, they maybe should focus on using better models from the beginning.

Similar to the overselling of results, the authors seemed to have an “agenda”, a “hypothesis” of what they wanted to show. Irrespective of experimental evidence, they still claimed to see the desired outcome in the experiment, even if the experiment showed the exact opposite, which is a very dangerous and unscientific practice.
Examples:

Line 156: “While layerwise pruned models consistently achieved the maximum accuracy (across prune percentages), the globally pruned models consistently provided significant robustness gains over the layerwise models. Furthermore, the globally-pruned models typically achieve higher or comparable accuracy at higher sparsity levels, indicating that using initialization methods with global pruning can yield CARDs with better accuracy and robustness at the same level of compactness.” I am confused. 1. The layerwise pruning methods (yellow) lose quite a lot of clean accuracy in the conv models (up to 10-12 percent points) while the global ones degrade the clean accuracy less. Thus, the first part of the first sentence “While layerwise pruned models consistently achieved the maximum accuracy (across prune percentages)” is wrong. 2. On both VGG and ResNet18, the global pruning methods show the same loss in clean accuracy as their robustness gain. Thus, the models are not “CARDS”, because they are less accurate than the baseline *for all considered pruning percentages*! Thus, the statement that “initialization methods with global pruning can yield CARDs with better accuracy and robustness at the same level of compactness” is incorrect. Furthermore, for the ResNet18, the gains/losses are on the order of 1-2 percent points which I would not call significant in the first place. This symmetry in losing clean accuracy and gaining the “same amount” in robustness makes me wonder whether there is some weird robustness-accuracy trade-off at play?

Line 186: “In particular, by comparing the center of the heatmaps for layerwise and global methods, it follows that initialization methods leveraging layerwise pruning are more robust to low frequency perturbations, which translates to a higher accuracy on the clean data.” -> This is plain wrong. We look at conv8 with 80% weights pruned. Conv8 at 80% pruned weights has a 2% lower clean accuracy for Edgepopup and maybe a 0.5% increase in clean accuracy for Biprop (both initialization methods). Virtually all the remaining models have lower clean accuracy compared to baseline at all pruning levels for layerwise pruning.


Line 189: “On the other hand, global pruning in initialization methods yields CARDs that are less sensitive to mid to high frequency perturbations when compared to layerwise pruned CARDs, which translates to a
higher robustness to the OOD data.” To me, the layerwise pruning heat maps definitely have higher robustness values in the mid-to-high range frequencies compared to the global pruning methods. Following the authors’ argumentation, layerwise pruning methods should then have higher OOD robustness compared to layerwise pruning methods which they don’t.

Line 216: “Our analysis in Section 3 highlights that the robustness of the compressed model on frequency
perturbed data is crucially dependent on the interaction between the frequencies the model is biased towards and the frequencies on which the test-time corruption is concentrated.” This is wrong. The authors did not evaluate the individual corruptions in CIFAR-C, but only differentiated between clean (low frequency) and corrupted (high frequency) corruptions. Considering that CIFAR-C corruptions have very different frequencies, I find the analysis too coarse and the statement here far too bold.

The results in Table 1 look very strange. The vanilla trained model has the highest baseline accuracy which is significantly degraded by Gaussian noise training and Augmix. There must some optimization issue. Since the authors use an adaptive method, I would suggest comparing to TENT (https://arxiv.org/abs/2006.10726v1) who also use an adaptive technique at test time and show results on CIFAR-C. I would then either use the same architecture as them or reimplement their method on a ResNet50.

The results are only done on CIFAR10. I would suggest performing the analysis on ImageNet since this is the dataset robustness researchers, i.e. the target audience for this paper, are most interested in. I doubt robustness researchers would pay attention to this work if the analysis is only done on CIFAR10.
In general, the used models are very far from state-of-the-art models on CIFAR10. Checking the CIFAR10 leaderboard (https://paperswithcode.com/sota/image-classification-on-cifar-10), models with a clean accuracy of 94.4% are around place 104 in the leaderboard. The VGG net has a top-1 accuracy of 93.4%. Given that CIFAR10 is a very small dataset and thus, does not impose huge computation requirements, I would suggest using a state-of-the-art model. Even larger state-of-the-art CIFAR10 models can still be run on one GPU with a reasonable computation time.
Line 262: Gaussian noise is part of CIFAR-C. The authors overfit to the test set rendering their results meaningless. In any case, the performance gains over the baseline are moderate at best (2 percent points max).

This was a rough summary of my impression of the paper and a justification of my rating. Please see the general comments section for further comments.

### General comments:

Line 96ff. “F1 represents the function to find the masks, F2 represents the function to train the weights after pruning, ki is the earliest training iteration that Fi requires information from”, “Rewinding-based Lottery Tickets: k1 = T and k2 = r; r < T;” I am confused by this definition. I understand F2 to be the function to be trained after pruning. This means, F2 needs to know F1 (i.e. the masks). Then, the definition k1=T, k2=r; r<T does not make sense because this would lead to the opposite: We would first train the already pruned function at k2=r and then find the pruning masks at k1=T. Please rephrase this paragraph for clarity.
Which heuristic is used to determine the final network? Is it the accuracy on the clean training set?

Line 116: „For a reliable comparison, we make use of convolutional [9], VGG [39] and ResNet [16] style architectures of varying size and train them on CIFAR-10 [24]“. The distinction is unclear: VGG and ResNet architectures also involve convolutions.

Line 125: “For each pruning method, we plot the mean accuracy or robustness value over all realizations relative to the mean of the dense baseline realizations with error bars extending to the minimum and maximum realization value at each of the aforementioned sparsity levels.” This sentence is very hard to parse, please rephrase. Also: unclear which “realizations”.
Does the y-axis of Figure 1 show percent points?

Line 134: “However, we find that rewinding and initialization based pruning approaches consistently produce dramatic gains in robustness relative to dense baselines while matching (and sometimes surpassing) the accuracy of the dense baseline.” -> “consistently dramatic gains” is a strong claim and for the ResNet18 which is a popular model for CIFAR10 and which is the one with the highest clean accuracy of 94.4% here, the gain seems to be at 2 percent points which is rather small.

Line 137: “In particular, the rewinding class of methods provide a consistent, moderate improvement to both accuracy and robustness while the initialization class provides more substantial gains in robustness even when the accuracy is slightly below the baseline accuracy.” -> The second statement about the initialization class strongly oversells: For the ResNet18, the drop in clean accuracy is actually larger than the gains in robustness! For the VGG network, we see a loss of 3% in clean accuracy and a gain of 6% in robustness.

Line 139: “The significance of overparameterization to finding highly compact CARDs using initialization methods is evident for all architecture types, as the robustness of these models in higher pruning regimes improves at increasing levels of parameterization.” I am confused by this sentence. A ResNet18 has more parameters than a Conv4, but sees fewer gains from pruning? Additionally, the gains diminish for all models in higher pruning regimes. Please rephrase/ explain.

Line 162: “drastically different levels of OOD” -> very bold statement. I would not call a gain of 2 percent points for the best and thus most interesting model drastic.

Line 173: “usually corresponding to clean data” -> what does “usually” mean here? Very vague, be more specific please.

Line 180: “Figure 3 illustrate how rewinding and initialization pruning methods reduce the error rate across the whole frequency spectrum of perturbations when compared to the dense model.” I do not see major differences for the rewinding based approaches. I would highly suggest to the authors to show ResNet18 results here since it is their best and therefore, it is the most interesting and relevant model. Looking at Figs. 8 and 9 and the ResNet18 heat maps, I do not see any differences between the dense heat map and the rewinding methods’ heat maps. -> Maybe it would be helpful if the authors also reported an aggregated value because a qualitative comparison merely by looking at the heat maps can be ambiguous.


line 191: “Finally, traditional pruning methods yield a Fourier heatmap with a slightly altered structure when compared to the dense baseline and it slightly resembles a higher error rate version of heatmaps for rewinding methods.” Very vague, using “slightly” twice in this sentence and I don’t agree. I don’t see structure differences I would consider significant here. As discussed above, I interpret this sentence as authors wanting to follow their agenda and over-interpreting their results.

I don’t understand what Figure4 is supposed to tell me. I don’t see significant differences between the different columns.

The results in Table 1 seem wrong: In the first column, “clean” training leads to a robust accuracy of 89.2% while Gaussian noise training only leads to 73.7%. Given that the authors overfit to Gaussian noise on CIFAR-C, the accuracy with Gaussian noise training should be much higher than with clean training. It also seems wrong that Augmix seems to degrade accuracy over clean training. In addition, the high accuracy of 89.2% achieved with clean training is surpassed by not even one percent point when using the ensembling strategy. I would suggest to the authors to mark the best value in bold to make parsing the table easier.

Section 5.1. I do not understand how Theorem 1 relates to the experiments/ the rest of the paper. None of the pruning methods are about binary networks, yet Theorem 1 considers binary networks. In addition, in theorem 1, the pruned network is twice as deep as the target non-compressed network which also doesn’t hold for any of the pruning approaches above. The theoretical considerations seem like an after-thought and did not get how they supported the other insights from the paper.

### Updates to the Review post rebuttal

I think the authors did a great job addressing my comments and participating in the discussion with me. I specifically appreciate them removing and modifying instances where their results are "oversold". I think their last response to my question as to how the Fourier analysis and the OOD robustness analysis are connected is very rigorous and should be reflected in the paper as was presented in their comment.  Due to their last comment, I now understand the connection much better and it does make sense to me to include the Fourier analysis even though a direct correspondence to CIFAR10-C is not given. I urge the authors to include all points from their last comment in the manuscript to avoid confusion similar to what I experienced.
I appreciate that they included results from the robustbench benchmark and that they show that they actually get SotA results with their method. As stated before, I think the main research question is interesting and important, that is how to design compact DNNs that are accurate and robust at the same time.

I really like the current rebuttal setting of NeurIPS and I think that the submission has been improved a lot due to the discussion. I wish it was possible to upload a new version of the paper. I would like to reread it with the new changes, as it is hard to judge whether any incosistencies remain.

I have raised my rating.

**Time Spent Reviewing:**

11-12

---

> ### Author Response · Authors · 2021-08-10
> **Our focus is on simultaneously achieving compactness, accuracy, and robustness (further we achieve new SOTA for models simultaneously targeting accuracy and robustness) 3/3**
>
> >Line 262: Gaussian noise is part of CIFAR-C. The authors overfit to the test set rendering their results meaningless. In any case, the performance gains over the baseline are moderate at best (2 percent points max).
>
> The objective here was to determine if pruning was compatible with state-of-the-art data augmentation techniques for OOD robustness. The technique leveraging Gaussian noise [2] is one of the top performing methods on RobustBench [1] and our results show that pruning can be used in a complementary fashion with this technique. Additionally, 1) CIFAR-10-C has 15 corruption types, each with 5 levels of severity -- training uses only one of these corruption types, i.e., Gaussian at one given severity level -- and 2) the training set, CIFAR-10, and test data, CIFAR-10-C, do not have any overlap in terms of clean images. Hence, the claim that Gaussian augmentation from [2] “overfit to the test set rendering their results meaningless” is too strong. Further, we show that adding pruning to AugMix, which does not make use of Gaussian-noise-based data augmentation, improves clean and robust accuracy even at 95% sparsity, when compared to the same dense architecture trained using Augmix. Finally, your comment on the performance does not account for the compactness of the CARDs.
>
>
> >Line 96ff. “F1 represents the function to find the masks, F2 represents the function to train the weights after pruning, ki is the earliest training iteration that Fi requires information from”, “Rewinding-based Lottery Tickets: k1 = T and k2 = r; r < T;” I am confused by this definition. I understand F2 to be the function to be trained after pruning. This means, F2 needs to know F1 (i.e. the masks). Then, the definition k1=T, k2=r; r<T does not make sense because this would lead to the opposite: We would first train the already pruned function at k2=r and then find the pruning masks at k1=T. Please rephrase this paragraph for clarity. Which heuristic is used to determine the final network? Is it the accuracy on the clean training set?
>
> We optimize and prune all networks using only performance on the clean training set, and we will clarify this; note that we stated that we use the hyperparameters used by the developers of each method (lines 115-116). Also, we will state that in *rewinding*-based lottery ticket methods, F2 is applied after F1. Note that F2 requires information *from* iteration r (where r<T), however that does not mean that F2 is applied *on* iteration r (i.e., you can bring information from iteration r to iteration T), which we will clarify.
>
>
> >Line 125: “For each pruning method, we plot the mean accuracy or robustness value over all realizations relative to the mean of the dense baseline realizations with error bars extending to the minimum and maximum realization value at each of the aforementioned sparsity levels.” This sentence is very hard to parse, please rephrase. Also: unclear which “realizations”. Does the y-axis of Figure 1 show percent points?
>
> The y-axis of Figure 1 is percentage points. For each architecture, we did the following:
> 1) As a baseline, we trained 5 realizations without pruning
> 2) For each pruning method and pruning percentage we trained 5 realizations
> 3) The CIFAR-10 (CIFAR-10-C) test accuracies were averaged for the 5 baseline realizations then subtracted from the average CIFAR-10 (CIFAR-10-C) test accuracies for the 5 realizations from each pruning method and pruning percentage. These are the values that are plotted as lines in row 1 (row 2).
> 4) The error bars extend to the maximum and minimum values from the 5 realizations minus the average baseline accuracy.
>
> We can clarify this sentence in the final version of the paper.
>
>
> >Line 180: “Figure 3 illustrate how rewinding and initialization pruning methods reduce the error rate across the whole frequency spectrum of perturbations when compared to the dense model.” I do not see major differences for the rewinding based approaches. I would highly suggest to the authors to show ResNet18 results here since it is their best and therefore, it is the most interesting and relevant model. Looking at Figs. 8 and 9 and the ResNet18 heat maps, I do not see any differences between the dense heat map and the rewinding methods’ heat maps. -> Maybe it would be helpful if the authors also reported an aggregated value because a qualitative comparison merely by looking at the heat maps can be ambiguous.
>
> Referring to the heatmap legend, note that even minor color differences correspond to large error rate differences (i.e. red to orange is ~0.1, 10 percentage points, difference). Using the legend when comparing LTH/LRR to the dense baseline, there are several visible regions of the heatmap corresponding to a difference in the error rate of at least 0.2 (i.e. 20 percentage points). We can try an alternative colormap scheme to emphasize the differences. Thank you for your suggestion. We will also consider an appropriate aggregate value that could be used.
>
> >I don’t understand what Figure4 is supposed to tell me. I don’t see significant differences between the different columns.
>
> Our earlier results suggested that LRR was superior to fine-tuning at producing robust networks via pruning, which was intriguing because you can build the LRR approach by modifying just a few aspects of the fine-tuning approach (lines 201-203). Accordingly, producing heatmaps for each intermediate pruning method that results as these modifications are made (as we do in Figure 4) reveals whether robustness in the Fourier domain visibly improves at some point in the modification process. Interestingly, at just 4 shots of pruning (4th column), heatmaps were notably cooler than in the fine-tuning case. In other words, a shorter-duration LRR training scheme (fewer shots) can yield similar error rates across the frequency domain to a longer-duration LRR scheme, which suggests the possibility of reduced training time for robust LRR models (lines 211-213), as each shot of pruning requires retraining the model.
> As mentioned in the previous response, potentially changing the colormap used for the heatmap could display the differences more clearly.
>
> >The results in Table 1 seem wrong: In the first column, “clean” training leads to a robust accuracy of 89.2% while Gaussian noise training only leads to 73.7%. Given that the authors overfit to Gaussian noise on CIFAR-C, the accuracy with Gaussian noise training should be much higher than with clean training. It also seems wrong that Augmix seems to degrade accuracy over clean training. In addition, the high accuracy of 89.2% achieved with clean training is surpassed by not even one percent point when using the ensembling strategy. I would suggest to the authors to mark the best value in bold to make parsing the table easier.
>
> This was the result of a LaTeX formatting issue in Table 1 which caused all accuracy values to be shifted to the right by one position in the table (as pointed out by another reviewer). We will be sure to mark the best value in bold in the revised version of the paper.
>
> >Section 5.1. I do not understand how Theorem 1 relates to the experiments/ the rest of the paper. None of the pruning methods are about binary networks, yet Theorem 1 considers binary networks.
>
> We will clarify that all of the networks trained using biprop and edgepopup algorithms throughout the paper have binary weights (noted on L143 - 145, also stated on L671 in Section C of appendix and Section D). Note that we have also provided theoretical results for real-value weighted networks in the appendix (Section H.1).
>
> &nbsp;
>
> Thank you again for taking the time to review our submission. We hope that you have found our responses to be meaningful and clarifying and that you will reconsider your original score of our work. We welcome any additional questions that you may have.
>
> &nbsp;
>
>
> *References*:
>
> [1] Croce et. al., *RobustBench: A Standardized Adversarial Robustness Benchmark*, 2020
>
> [2] Kireev et. al., *On the effectiveness of adversarial training against common corruptions*, 2021
>
> [3] Yin et. al., *A Fourier Perspective on Model Robustness in Computer Vision*, NeurIPS 2019
>
> [4] Zhang et. al., *Lottery Jackpots Exist in Pre-trained Models*, 2021
>
> [5] Diffenderfer et. al., *Multi-Prize Lottery Ticket Hypothesis: Finding Accurate Binary Neural Networks by Pruning A Randomly Weighted Network*, ICLR 2021
>
> [6] Hooker et. al., *What do compressed deep neural networks forget?*, 2020
>
> [7] Liebenwein et. al., *Lost in Pruning: The Effects of Pruning Neural Networks beyond Test Accuracy*, 2021

---

> > ### Comment · Reviewer_kbDD · 2021-08-11
> > **Summary response to the rebuttal (addressing the authors' comments to my review), asking for discussion**
> >
> > First of all, I would like to thank the authors for providing such a detailed answer to my comments, I really appreciate it. I would also like to engage in a discussion with them.
> >
> > First, I agree that the corruption numbers on the RobustBench benchmark are the relevant ones for this analysis and not the ones on clean CIFAR10. This was an oversight on my part and I apologize. I appreciate the authors including the top number from that benchmark in their new Table in the rebuttal.
> >
> > Considering the comment about line 156: Yes, I misunderstood the Figure, thanks for the clarification. Please fix this in the manuscript to avoid others also misunderstanding it.
> >
> > Considering the comment on “While layerwise pruned models consistently achieved the maximum accuracy (across prune percentages), the globally pruned models consistently provided significant robustness gains over the layerwise models.”: I understand the point now and I think the issue is that in line 94, one compares pruned models to unpruned models while the statement in line 158 compares global vs layerwise pruning (specifically the line “better accuracy and robustness”). I would suggest clarifying this in the text. I agree that the findings in Figure 2 support the CARDS hypothesis (specifically the “comparable (or higher) accuracy and robustness” line) if a loss of 1-2 percent points in clean accuracy is “comparable”. It is probably nitpicking but the loss in clean accuracy for a ResNet18 is of the same magnitude as the gain in robust accuracy at a prune percentage of 90%. So it is a bit fishy calling the 1-2 percent points loss in clean accuracy “comparable”, while at the same time calling the 1-2 percent points gain in robust accuracy “better”.
> >
> > Could the authors please post an updated Table 1 without the LaTeX bug? The authors write “The objective here was to determine if pruning was compatible with state-of-the-art data augmentation techniques for OOD robustness.” This is fine and should be communicated as such. My comment “In any case, the performance gains over the baseline are moderate at best (2 percent points max).” targeted the line “We find that both the data-agnostic and adaptive methods are capable of **significantly** improving the performance.” After all, a loss of two percent points in clean accuracy has been considered “comparable” In Figure 2 ;).
> >
> > Commenting on the "overselling the results" issue: The CARDS hypothesis assumes a "comparable or higher" accuracy and robustness. The larger globally pruned models in Fig. 2 then have ~2 percent points lower clean accuracy which is considered "comparable" I suppose. This is fine but then a 2 percent point gain in robustness is also "comparable" and not "dramatically more robust" (abstract), "drastically different" (line 162). Please use more consistent words to describe similar findings.
> >
> > ### The heatmaps discussion ###
> >
> > The authors write about line 216: “Note that this is a remark about Fourier-basis perturbations, not CIFAR-10-C corruptions; i.e., the “test-time corruption” in the case of the Fourier heatmaps is a set of Fourier-basis perturbations.“ -> Yes, in this case it is fine. But then the authors should remove all connections between the Fourier heatmap analysis and OOD robustness from Section 3., e.g. the statement in line 189. If their statement in line 216 is not targeted at CIFAR-C corruptions at all, it is essentially redundant as all it says is: “Robustness to Fourier test-time corruptions depends on the model’s sensitivity to these Fourier test-time corruptions” which is obvious. Also, in [3], the authors differentiated between different corruption types and observed very different behavior to low-frequency perturbations such as fog and contrast versus high-frequency perturbations such as various noise perturbations.
> >
> > My comment about the incorrect observations from line 189 was not addressed but it should be. My comment is: Line 189: “On the other hand, global pruning in initialization methods yields CARDs that are less sensitive to mid to high frequency perturbations when compared to layerwise pruned CARDs, which translates to a higher robustness to the OOD data.” To me, the layerwise pruning heat maps definitely have higher robustness values in the mid-to-high range frequencies compared to the global pruning methods. Following the authors’ argumentation, layerwise pruning methods should then have higher OOD robustness compared to layerwise pruning methods which they don’t.
> > -> It is not clear to me whether the authors attempt to link the Fourier heatmap results to the results on CIFAR-C or not. It still seems to me like they do, e.g. in line 162 they write “In order to understand what causes pruning methods to have drastically different levels of OOD robustness, we leverage the Fourier sensitivity method [43] to perform a frequency-domain analysis of models trained using various pruning techniques.”
> > In summary, I think there is no evidence for a connection between the heatmaps and the results on CIFAR10-C or there is even evidence for the opposite, see my comment in line 189. In the current form, the presentation of the results is misleading and suggests there might be a connection. I would like the authors to comment on this issue.
> >
> > Comment to line 180: The authors write “Using the legend when comparing LTH/LRR to the dense baseline, there are several visible regions of the heatmap corresponding to a difference in the error rate of at least 0.2 (i.e. 20 percentage points).” I still think that the heatmaps in Figs. 8 and 9 look virtually the same for the dense heat map and the rewinding methods. Maybe the authors could plot the difference between the heatmaps? Currently, this point still stands for me. The same goes for Figure 4: The columns look pretty much the same to me. I think plotting the difference to the dense model might be a better choice here.

---

> > > ### Author Response · Authors · 2021-08-14
> > > **Heatmap Discussion (with quantitative support) 3/3**
> > >
> > > *Table C1*: Table 1 from paper without LaTeX formatting issue.
> > >
> > > |Prune Percentage|       |Unpruned (Dense Baseline)|-------|--->    |Edgepopup (Layerwise)|-------|--->    |Edgepopup (Global)|-------|--->    |LRR (Global)|-------|--->    |LTH (Global)|-------|--->    |Biprop (Layerwise)|-------|--->    |Biprop (Global)|-------|--->    |
> > > |------|--------------|-------------------------|-------|--------|---------------------|-------|--------|------------------|-------|--------|------------|-------|--------|------------|-------|--------|------------------|-------|--------|---------------|-------|--------|
> > > |      |              |Augmix                   |Clean  |Gaussian|Augmix               |Clean  |Gaussian|Augmix            |Clean  |Gaussian|Augmix      |Clean  |Gaussian|Augmix      |Clean  |Gaussian|Augmix            |Clean  |Gaussian|Augmix         |Clean  |Gaussian|
> > > |      |Clean Acc     |95.5                     |95.1   |93.9    |94.3                 |93.7   |92.4    |94.9              |94.4   |93      |96.1        |95.6   |93.8    |95.6        |94.9   |93.5    |93.9              |93.7   |92.4    |94.5           |94.1   |93.2    |
> > > |80%   |Robust Acc    |89.2                     |73.7   |85.6    |87.8                 |76.7   |85.1    |88.4              |74.4   |85.7    |89.8        |76.3   |86.4    |89.4        |74.4   |85.8    |87.5              |76.1   |85.1    |87.8           |74.3   |85.3    |
> > > |      |Memory (Mbits)|358                      |358    |358     |2.23                 |2.23   |2.23    |2.23              |2.23   |2.23    |71.5        |71.5   |71.5    |71.5        |71.5   |71.5    |2.23              |2.23   |2.23    |2.23           |2.23   |2.23    |
> > > |      |              |                         |       |        |                     |       |        |                  |       |        |            |       |        |            |       |        |                  |       |        |               |       |        |
> > > |      |Clean Acc     |95.5                     |95.1   |93.9    |94.4                 |93.9   |92.9    |94.4              |94.1   |92.8    |96.3        |95.6   |93.9    |95.7        |95.2   |93.6    |94.4              |94.1   |92.7    |93.7           |93.6   |93      |
> > > |90%   |Robust Acc    |89.2                     |73.7   |85.6    |88                   |75.6   |85.2    |87.9              |76     |85.4    |89.8        |75.7   |86.3    |89.7        |76.4   |86      |87.8              |75.1   |85      |87.1           |74.6   |84.2    |
> > > |      |Memory (Mbits)|358                      |358    |358     |1.12                 |1.12   |1.12    |1.12              |1.12   |1.12    |35.8        |35.8   |35.8    |35.8        |35.8   |35.8    |1.12              |1.12   |1.12    |1.12           |1.12   |1.12    |
> > > |      |              |                         |       |        |                     |       |        |                  |       |        |            |       |        |            |       |        |                  |       |        |               |       |        |
> > > |      |Clean Acc     |95.5                     |95.1   |93.9    |94.5                 |94     |92.6    |93.2              |92.7   |91.2    |96.1        |95.7   |93.9    |95.8        |95.1   |93.8    |94.2              |93.8   |92.5    |92.5           |92.1   |91.4    |
> > > |95%   |Robust Acc    |89.2                     |73.7   |85.6    |87.8                 |73.1   |84.3    |85.7              |73.4   |83.9    |89.6        |75.6   |86.3    |89.7        |76.5   |86      |87.5              |73.8   |84.4    |84.7           |73.4   |83.1    |
> > > |      |Memory (Mbits)|358                      |358    |358     |0.56                 |0.56   |0.56    |0.56              |0.56   |0.56    |17.9        |17.9   |17.9    |17.9        |17.9   |17.9    |0.56              |0.56   |0.56    |0.56           |0.56   |0.56    |
> > >
> > >
> > >
> > > &nbsp;
> > >
> > > &nbsp;
> > >
> > >
> > >
> > > *Table C2*: Statistics on Difference of LTH/LRR Heatmaps to Baseline Heatmaps in Figures 8, 9, and 10.
> > >
> > > |Method|Heatmap Epsilon|Figure Number|Percentage of Heatmap with Greater Than 5 Percentage Point Difference to Baseline|
> > > |------|---------------|-------------|------------------------|
> > > |LTH   |3              |8            |27 |
> > > |LTH   |4              |8            |22 |
> > > |LTH   |6              |8            |14 |
> > > |LTH   |3              |9            |26 |
> > > |LTH   |4              |9            |26 |
> > > |LTH   |6              |9            |17 |
> > > |LTH   |3              |10           |32 |
> > > |LTH   |4              |10           |28 |
> > > |LTH   |6              |10           |22 |
> > > |      |               |             |         |
> > > |LRR   |3              |8            |31 |
> > > |LRR   |4              |8            |27 |
> > > |LRR   |6              |8            |14 |
> > > |LRR   |3              |9            |39 |
> > > |LRR   |4              |9            |34 |
> > > |LRR   |6              |9            |26 |
> > > |LRR   |3              |10           |35 |
> > > |LRR   |4              |10           |26 |
> > > |LRR   |6              |10           |18 |
> > >
> > > &nbsp;
> > >
> > > &nbsp;
> > >
> > >
> > >
> > > *Table C3*: Statistics on Difference of Heatmaps in Figure 4 to Baseline Heatmaps.
> > >
> > > |Method               |Heatmap Epsilon|Percentage of Heatmap with Greater Than 5 Percentage Point Difference to Baseline|
> > > |---------------------|---------------|---------------------------------------------------------------------|
> > > |Fine Tuning 40 Epoch |3              |35        |
> > > |Fine Tuning 40 Epoch |4              |31  |
> > > |Fine Tuning 40 Epoch |6              |25  |
> > > |Fine Tuning 148 Epoch|3              |37    |
> > > |Fine Tuning 148 Epoch|4              |41   |
> > > |Fine Tuning 148 Epoch|6              |40   |
> > > |LRR 1-Shot           |3              |31     |
> > > |LRR 1-Shot           |4              |30  |
> > > |LRR 1-Shot           |6              |22  |
> > > |LRR 4-Shot           |3              |16  |
> > > |LRR 4-Shot           |4              |17  |
> > > |LRR 4-Shot           |6              |15  |
> > > |LRR 8-Shot           |3              |26  |
> > > |LRR 8-Shot           |4              |26  |
> > > |LRR 8-Shot           |6              |21  |
> > > |LRR 13-Shot          |3              |36  |
> > > |LRR 13-Shot          |4              |25  |
> > > |LRR 13-Shot          |6              |19  |
> > > |LRR 18-Shot          |3              |47  |
> > > |LRR 18-Shot          |4              |42  |
> > > |LRR 18-Shot          |6              |24  |
> > >
> > > &nbsp;
> > >
> > > &nbsp;
> > >
> > >
> > >
> > >
> > > *Table C4: Relationship of Layerwise and Globally Pruned Biprop Conv8 Models on Low and High Frequency Regions.* Note that the mean is computed for the (layerwise - global) heatmap. As the heatmaps specify error rate, a negative mean value indicates the layerwise model has a lower error rate on average while a positive mean value indicates the globally pruned model has a lower error rate on average.
> > >
> > > |         | <------------------------- | ---Low Frequency --- | ---> | <-------------------------   | ---High Frequency--- | ---> |
> > > |---------------|-------------------|---------------|------|---------|--------|------|
> > > |Heatmap Epsilon|Percentage where Global > Layerwise (by at least 5 Percentage Points)|Percentage where Layerwise > Global (by at least 5 Percentage Points)|Mean  |Percentage where Global > Layerwise (by at least 5 Percentage Points)|Percentage where Layerwise > Global (by at least 5 Percentage Points)|Mean  |
> > > |3              |0  |37 |-3.7  |44 |4.1 |4.2   |
> > > |4              |0 |60 |-5.7  |48 |5.4 |5     |
> > > |6              |0 |65 |-7.1  |47 |7.4 |5     |

---

> > > > ### Comment · Reviewer_kbDD · 2021-08-30
> > > > **Reponse to the Heatmaps discussion**
> > > >
> > > > I apologize for replying so late. Hopefully, we can still have an iteration of discussion with the authors. I appreciate the detailed response to my comments. I do still have concerns and questions.
> > > >
> > > > The authors agreed to remove phrases linking OOD robustness as measured by CIFAR10-C and their Fourier analysis. However, if the connections between the Fourier analysis and OOD robustness are indeed removed, the paper will essentially present two different robustness measures and their connection will not be discussed in any way. I think this will be confusing to the reader.
> > > >
> > > > The authors do comment on what they value about the Fourier analysis: “Section 3’s demonstration of model sensitivity differences caused by the choice of pruning approach on the broad range of frequency perturbations examined in the Fourier heatmaps supports a critical finding of Section 2 (pruning approach differences produce OOD robustness differences).” -> The authors essentially say that the heatmaps are different for the various pruning techniques and the OOD robustness results are different for various pruning techniques. I do not understand why this is so surprising or interesting. One does different things to a model and it behaves differently. Further in this paragraph, the authors write: “models pruned using different strategies respond differently to perturbed test data based on the frequency of the perturbation---and in doing so provide guidance for the future development of CARDs.” The latter point is unmotivated. How do the authors intend using the Fourier heatmaps to develop future CARDs?
> > > >
> > > > Ref [3] which the authors base their Fourier analysis on establishes a link between the Fourier range of different corruptions and the model’s accuracy on those. For this, the authors of [3] look at all corruptions individually and this is the reason why they can make this claim. In this paper, the authors do not look at individual corruptions, but only at the agglomerate result which is far too coarse. I think this is the main reason why the link between their Fourier basis perturbations and CIFAR10-C corruptions is so hard to establish.
> > > >
> > > > I think my last unresolved comment can be summarized as: If the OOD robustness and the Fourier basis corruptions are unrelated, what is the benefit of the Fourier analysis? What can we learn from it? How can it help us find new CARDS?
> > > >
> > > > I appreciate the authors’ comments on my comments to line 189 and this issue is now resolved for me.
> > > >
> > > > I appreciate the authors plotting the differences in heatmaps and it’s a shame Openreview does not allow uploading images. I trust the authors to generate and include sensible images in the manuscript, even though they cannot show them to me now. On the other hand, maybe the authors could upload the images anonymously somewhere and just post a link here?

---

> > > > > ### Author Response · Authors · 2021-08-31
> > > > > **Thank you for following up**
> > > > >
> > > > > Thank you for your commitment to improving our submission’s clarity. We are glad that we were able to resolve all of your concerns except for one, which we believe can be addressed with a minor revision and relates to the connection of Section 3 to our other results.
> > > > >
> > > > > First, we believe it may be helpful to quickly recap the manuscript’s current narrative. Prior results in the literature were negative with respect to the ability of pruning to improve robustness (Section 1). Hence, our showing that pruning could improve robustness on CIFAR-10-C was surprising (Section 2), and the apparent explanation for the uniqueness of our results was that the choice of pruning method was critical. As CIFAR-10-C corruptions are limited to certain frequency ranges [3], we went a step further to see if this explanation held on a broader range of frequencies using Fourier sensitivity analysis (Section 3). Each point in the heat map corresponds to a perturbation to a specific frequency; altogether the heatmaps reinforce that our positive results and explanation are not a byproduct of particular frequency combinations that appear in CIFAR-10-C and hold broadly. Next, leveraging the compactness of CARDs (and intuition from Section 3), we showed that this robustness gain can be amplified/enhanced by developing methods to ensemble multiple CARDs, which---together with techniques for improving OoD robustness---set new SOTA performance on CIFAR-10-C (Section 4) and, more recently, CIFAR-100-C. Finally, we concluded with theoretical support for the CARD hypothesis (Section 5).
> > > > >
> > > > > We believe our rewriting of the introductory sentence of Section 3 (line 162) to clarify the information in the above narrative fully addresses your concern regarding the connection between Sections 2 and 3. (Note that we have already removed statements suggesting there may be a one-to-one connection between CIFAR-10-C performance and Fourier heatmaps.)
> > > > >
> > > > > Specifically, regarding your “last unresolved comment”, please note the manuscript now states that the understanding of benefits of particular methods suggested by our Section 2 analysis (CIFAR-10-C) is being corroborated and augmented by the broader view provided in Section 3 (Fourier heatmaps). In other words, heatmaps serve as an analysis/validation tool providing performance understanding beyond that of CIFAR-10-C, and thereby can guide the future development of CARDs.
> > > > >
> > > > > >I trust the authors to generate and include sensible images in the manuscript, even though they cannot show them to me now.
> > > > >
> > > > > Thank you! We will add these to the revised manuscript.

---

> > > > > > ### Comment · Reviewer_kbDD · 2021-08-31
> > > > > > **Reply to last comment on heatmaps**
> > > > > >
> > > > > > Thank you, this response clarified my confusion and it makes sense now. Please incorporate it in the manuscript so others do not experience the same confusion as me.

---

> > > ### Author Response · Authors · 2021-08-14
> > > **Heatmap Discussion (with quantitative support) 2/3**
> > >
> > > >Comment to line 180: The authors write “Using the legend when comparing LTH/LRR to the dense baseline, there are several visible regions of the heatmap corresponding to a difference in the error rate of at least 0.2 (i.e. 20 percentage points).” I still think that the heatmaps in Figs. 8 and 9 look virtually the same for the dense heat map and the rewinding methods. Maybe the authors could plot the difference between the heatmaps? Currently, this point still stands for me.
> > >
> > > To quantitatively note the differences between heatmaps, we have provided a table of results containing statistics indicating the difference between the rewinding heatmaps and the baselin (dense) heatmaps for Figures 8, 9, and 10 (Table C2). In summary, on average ~25% of each heatmap has a difference of 5 percentage points or more when compared to the baseline. Following your suggestion, we have already generated the heatmaps corresponding to the difference with the baseline and these can be used in place of the existing heatmaps for improved visual clarity.
> > >
> > >
> > > >The same goes for Figure 4: The columns look pretty much the same to me. I think plotting the difference to the dense model might be a better choice here.
> > >
> > > Similar to the previous point, we have generated a table of quantitative results comparing the heatmaps in Figure 4 to the baseline (Table C3). In summary, on average ~28% of each heatmap has a difference of 5 percentage points or more when compared to the baseline. We have also generated the heatmaps illustrating the difference to the baseline that can be used in place of the existing heatmaps.
> > >
> > >
> > > >My comment about the incorrect observations from line 189 was not addressed but it should be. My comment is: Line 189: “On the other hand, global pruning in initialization methods yields CARDs that are less sensitive to mid to high frequency perturbations when compared to layerwise pruned CARDs, which translates to a higher robustness to the OOD data.” To me, the layerwise pruning heat maps definitely have higher robustness values in the mid-to-high range frequencies compared to the global pruning methods.
> > >
> > > To support the claims in the paper, here we provide a rigorous definition of low and high frequency regions followed by quantitative results (Table C4). Using the heatmaps for global and layerwise biprop models in FIgure 3, we have generated a table of statistics to address your comment that “layerwise pruning heatmaps definitely have higher robustness values in the mid-to-high range frequencies when compared to the global pruning methods”. As *low* and *high* frequencies are defined in a relative scale to natural/clean images [3], we generated a mask based on clean images to separate regions of the heatmap into *low* and *high* frequencies. To do this, we compute and normalize the 2-dimensional power spectral density for the average of the clean CIFAR-10 images. The low frequency region is composed of pixels that have a power of greater than 0.15. The region containing the low frequency information is in the center of the heatmap (which is consistent with Figure 2 from [3]). We then computed statistics for the layerwise and global biprop heatmaps contained in Figure 3 when restricted to the low and high frequency regions of the heatmap. To summarize the statistics of the table:
> > >
> > > * In the low frequency region, the layerwise model is more accurate than the global model (by an average of 3.7 to 7.1 percentage points depending on heatmap severity)
> > > * In the high frequency region, the global model is more accurate than the layerwise model (by an average of 4.2 to 5 percentage points depending on heatmap severity)
> > > * For all three heatmap severities (epsilon = 3, 4, 6) the layerwise model outperforms the global model by 5 percentage points in 37 - 65% of the low frequency region.
> > > * For all three heatmap severities (epsilon = 3, 4, 6) the global model outperforms the layerwise model by 5 percentage points in 44 - 48% of the high frequency region.
> > >
> > > These patterns were found to be consistent with those at other threshold values as well, e.g., frequencies with 0.1 or 0.2.

---

> > > ### Author Response · Authors · 2021-08-14
> > > **Heatmap Discussion (with quantitative support) 1/3**
> > >
> > > Thank you for following up and agreeing to correspond with us to help improve the clarity of our paper! We are glad that we were able to clarify the major contributions of our paper in our previous response. We believe your remaining points, which we address below, can be clarified and require only minor revisions.
> > >
> > >
> > > >Considering the comment about line 156: Yes, I misunderstood the Figure, thanks for the clarification. Please fix this in the manuscript to avoid others also misunderstanding it.
> > >
> > > We have made this revision to our paper to avoid misunderstandings.
> > >
> > >
> > > >Could the authors please post an updated Table 1 without the LaTeX bug?
> > >
> > > Yes, we have provided these results in a table at the bottom of this response (Table C1).
> > >
> > >
> > > >The authors write “The objective here was to determine if pruning was compatible with state-of-the-art data augmentation techniques for OOD robustness.” This is fine and should be communicated as such.
> > >
> > > We have clarified this point in the revised version of our paper.
> > >
> > >
> > > >My comment “In any case, the performance gains over the baseline are moderate at best (2 percent points max).” targeted the line “We find that both the data-agnostic and adaptive methods are capable of **significantly** improving the performance.” After all, a loss of two percent points in clean accuracy has been considered “comparable” In Figure 2 ;).
> > >
> > > We agree that the language used in this sentence can be interpreted as only regarding accuracy and robustness and, in light of our discussion, should be revised. A minor revision can be made to accurately express the significance with respect to the three metrics: compression, accuracy, and robustness. With respect to all three metrics we believe the results are significant  -- (1) Using 95% pruned CARDS, the LRR/LTH agnostic and adaptive CARD-Decks achieve higher accuracy, robustness, and compression than the dense baseline and (2) 95% pruned Edgepopup/Biprop CARD-Decks achieve comparable accuracy and robustness with compression of ~100x with respect to the baseline. These findings and advancements are significant for ML applications in resource-constrained environments, such as, but not limited to, the motivating application in the introduction.
> > >
> > >
> > >  >Considering the comment on “While layerwise pruned models consistently achieved the maximum accuracy (across prune percentages), the globally pruned models consistently provided significant robustness gains over the layerwise models.”: I understand the point now and I think the issue is that in line 94, one compares pruned models to unpruned models while the statement in line 158 compares global vs layerwise pruning (specifically the line “better accuracy and robustness”). I would suggest clarifying this in the text.
> > >
> > > We have clarified this is in the revised version of our paper.
> > >
> > > &nbsp;
> > >
> > > As we believe the following comments are somewhat related, we are grouping them together to provide a single response.
> > >
> > > >I agree that the findings in Figure 2 support the CARDS hypothesis (specifically the “comparable (or higher) accuracy and robustness” line) if a loss of 1-2 percent points in clean accuracy is “comparable”. It is probably nitpicking but the loss in clean accuracy for a ResNet18 is of the same magnitude as the gain in robust accuracy at a prune percentage of 90%. So, it is a bit fishy calling the 1-2 percent points loss in clean accuracy “comparable”, while at the same time calling the 1-2 percent points gain in robust accuracy “better”.
> > >
> > > >Commenting on the "overselling the results" issue: The CARDS hypothesis assumes a "comparable or higher" accuracy and robustness. The larger globally pruned models in Fig. 2 then have ~2 percent points lower clean accuracy which is considered "comparable" I suppose. This is fine but then a 2 percent point gain in robustness is also "comparable" and not "dramatically more robust" (abstract), "drastically different" (line 162). Please use more consistent words to describe similar findings.
> > >
> > > We agree with you on the use of this type of language and will make this change; as indicated in our initial response to your review and quoted here for convenience: “We agree with your comment on ‘drastic’ and ‘dramatically’ and will simplify the language in these statements in the final version of the paper.”
> > >
> > >
> > > >### The heatmaps discussion ###
> > >
> > > Based on your comments, it appears that you raise three points about the heatmaps in the paper:
> > > Non-perceptible difference between baseline heatmaps and LRR/LTH
> > > Interpretation of heatmaps for layerwise and globally pruned initialization methods
> > > Connection between the heatmaps and the robustness as measured on CIFAR-10-C
> > > We address points 1 and 2 by providing quantitative results in tables at the bottom of this response that support our statements in the paper. We also provide a summary of these quantitative results as responses to specific comments. First, we address point 3:
> > >
> > > >It is not clear to me whether the authors attempt to link the Fourier heatmap results to the results on CIFAR-C or not. It still seems to me like they do, e.g. in line 162 they write “In order to understand what causes pruning methods to have drastically different levels of OOD robustness, we leverage the Fourier sensitivity method [43] to perform a frequency-domain analysis of models trained using various pruning techniques.” In summary, I think there is no evidence for a connection between the heatmaps and the results on CIFAR10-C or there is even evidence for the opposite, see my comment in line 189. In the current form, the presentation of the results is misleading and suggests there might be a connection. I would like the authors to comment on this issue.
> > >
> > > In Section 3, we will revise the text to clarify our approach and prevent the potential misunderstanding that there is a one-to-one relationship between Fourier-perturbation error heatmaps and CIFAR-10-C robustness. In particular, we will: revise line 162 to explain that the Fourier heatmap analysis builds on and supports the conclusion from Section 2 by showing that the particular choice of pruning method is critical to OOD robustness more broadly (we provide quantitative support for this below per your related points); remove the phrase “which translates to a higher robustness [on] the OOD data'' from line 189; and proofread the paper with a focus on enhancing the clarity of the relationship of the Fourier heatmap technique to our results.
> > >
> > > Notably, the Fourier sensitivity analysis [3] (accepted to NeurIPS in 2019) is a valuable tool in the robustness community. While the perturbations in CIFAR-10-C are more complex than the Fourier bases perturbations used by this method, Section 3’s demonstration of model sensitivity differences caused by the choice of pruning approach on the broad range of frequency perturbations examined in the Fourier heatmaps supports a critical finding of Section 2 (pruning approach differences produce OOD robustness differences). Thus, despite the fact that these heatmap findings cannot be said to directly apply to a specific robustness benchmark like CIFAR-10-C, they demonstrate a previously unknown phenomenon---models pruned using different strategies respond differently to perturbed test data based on the frequency of the perturbation---and in doing so provide guidance for the future development of CARDs.
> > >
> > >
> > > >The authors write about line 216: “Note that this is a remark about Fourier-basis perturbations, not CIFAR-10-C corruptions; i.e., the “test-time corruption” in the case of the Fourier heatmaps is a set of Fourier-basis perturbations.“ -> Yes, in this case it is fine. But then the authors should remove all connections between the Fourier heatmap analysis and OOD robustness from Section 3., e.g. the statement in line 189. If their statement in line 216 is not targeted at CIFAR-C corruptions at all, it is essentially redundant as all it says is: “Robustness to Fourier test-time corruptions depends on the model’s sensitivity to these Fourier test-time corruptions” which is obvious.
> > >
> > > Please note that the sentence you reference on line 216 is the introduction of Section 4, which explicitly “[m]otivate[s]” (line 218) our analysis in Section 4 while providing a segue from Section 3. In particular, Section 3 uses an established technique [3] to demonstrate that different pruning approaches produce models with different biases or sensitivities to Fourier-perturbed OOD data (we quantified these differences in accordance with your other comments), not just CIFAR-10-C perturbations (as demonstrated in Section 2), which supports the criticality of the pruning approach to OOD robustness more broadly. Following this finding of Section 3, line 216 provides a segue to Section 4’s analysis. Specifically, by saying in line 216 that “Section 3 highlights” that models with different frequency biases have different robustnesses to particular test-time frequency corruptions, which you agree is obvious, we are providing the intuition for the Section 4 analysis which successfully uses the fact that frequency-based gating can switch to the model with the more appropriate frequency bias for the given test-time OOD data.

---

> ### Author Response · Authors · 2021-08-10
> **Our focus is on simultaneously achieving compactness, accuracy, and robustness (further we achieve new SOTA for models simultaneously targeting accuracy and robustness) 2/3**
>
> >Line 156: “While layerwise pruned models consistently achieved the maximum accuracy (across prune percentages), the globally pruned models consistently provided significant robustness gains over the layerwise models. Furthermore, the globally-pruned models typically achieve higher or comparable accuracy at higher sparsity levels, indicating that using initialization methods with global pruning can yield CARDs with better accuracy and robustness at the same level of compactness.” I am confused. 1. The layerwise pruning methods (yellow) lose quite a lot of clean accuracy in the conv models (up to 10-12 percent points) while the global ones degrade the clean accuracy less. Thus, the first part of the first sentence “While layerwise pruned models consistently achieved the maximum accuracy (across prune percentages)” is wrong.
>
> Our statement is not wrong, and it seems that this is a simple misinterpretation. Across all of the prune percentages tested, the maximum accuracy was achieved by a layerwise pruned model **at one of these** prune percentages. Your point that the globally pruned models degrade the clean accuracy less than the layerwise pruned models is correct from observing the figures but not the interpretation we intended from this statement. We will revise this statement in the final version of the paper to avoid misinterpretation.
>
>
> >2. On both VGG and ResNet18, the global pruning methods show the same loss in clean accuracy as their robustness gain. Thus, the models are not “CARDS”, because they are less accurate than the baseline for all considered pruning percentages! Furthermore, for the ResNet18, the gains/losses are on the order of 1-2 percent points which I would not call significant in the first place. This symmetry in losing clean accuracy and gaining the “same amount” in robustness makes me wonder whether there is some weird robustness-accuracy trade-off at play?
>
> We will clarify in the text that we are not claiming that the binary-weight, initialization-based methods you reference produce CARDs at every prune rate, rather in the instances where the accuracy approaches the dense baseline accuracy. In the case of VGG19, this corresponds to 95% prune percentage while for ResNet-18 this occurs at 90% and 95% prune percentage. The plots demonstrate the trends of each pruning method across all tested sparsity levels to provide a complete picture of how sparsity affects accuracy and robustness, not to say that these initialization based methods will always result in CARDs. Importantly, we wish to emphasize that the claim in our CARD hypothesis is that the accuracy and robustness of the **compressed** model is “comparable (or higher)” (line 94) than the dense model’s given sufficient overparameterization, meaning that the 1% gain/loss of robustness/accuracy you reference that occurs at extreme levels of compression (via pruning+binarization) counts as support for our main hypothesis. Finally, note that the performance of initialization based methods can be improved using various techniques such as starting with a pretrained network (instead of a randomly initialized network) and batchnorm tuning [4,5].
>
>
>
> >Thus, the statement that “initialization methods with global pruning can yield CARDs with better accuracy and robustness at the same level of compactness” is incorrect.
>
> Given that we clarified that initialization-based methods can produce CARDs in our response above, our statement that global pruning can yield CARDs with higher accuracy/robustness than layerwise pruning (lines 158-160) is clearly supported by Figure 2.
>
>
> >Line 186: “In particular, by comparing the center of the heatmaps for layerwise and global methods, it follows that initialization methods leveraging layerwise pruning are more robust to low frequency perturbations, which translates to a higher accuracy on the clean data.” -> This is plain wrong.
>
> We agree and will remove this statement from the paper. Importantly, this incorrect statement does not affect other parts of the paper and thus can be addressed by removing line 186. Relatedly, please see our response to the next comment regarding the usefulness of Fourier heatmap analysis.
>
>
> >Line 216: “Our analysis in Section 3 highlights that the robustness of the compressed model on frequency perturbed data is crucially dependent on the interaction between the frequencies the model is biased towards and the frequencies on which the test-time corruption is concentrated.” This is wrong. The authors did not evaluate the individual corruptions in CIFAR-C, but only differentiated between clean (low frequency) and corrupted (high frequency) corruptions. Considering that CIFAR-C corruptions have very different frequencies, I find the analysis too coarse and the statement here far too bold.
>
> The notion we are highlighting on L216 is well-accepted and has been the subject of prior work at NeurIPS [3]. Note that this is a remark about Fourier-basis perturbations, not CIFAR-10-C corruptions;  i.e., the “test-time corruption” in the case of the Fourier heatmaps is a set of Fourier-basis perturbations. Thus, it is correct  that the robustness of the compressed model on frequency perturbed data is crucially dependent on the interaction between the frequencies the model is biased towards and the frequencies on which the test-time corruption is concentrated, and we corroborate this insight in our empirical (Section 4) and theoretical (Section H.2 in appendix) results.
>
>
> >The results in Table 1 look very strange. The vanilla trained model has the highest baseline accuracy which is significantly degraded by Gaussian noise training and Augmix. There must some optimization issue. Since the authors use an adaptive method, I would suggest comparing to TENT (https://arxiv.org/abs/2006.10726v1) who also use an adaptive technique at test time and show results on CIFAR-C. I would then either use the same architecture as them or reimplement their method on a ResNet50.
>
> This was the result of a LaTeX formatting issue in Table 1 which caused all accuracy values to be shifted to the right by one position in the table (as pointed out by reviewer eMfa). We will be sure to mark the best value in bold in the revised version of the paper. The main objective of this paper was not to design a new method to compare against existing methods. However, we did show that existing pruning methods are complementary to existing robustness-improving methods and can yield comparable or even better performance. As such, TENT could be used in a complementary fashion with the pruning and the adaptive methods provided in our work as opposed to being viewed as a competitor.
>
>
> >The results are only done on CIFAR10. I would suggest performing the analysis on ImageNet since this is the dataset robustness researchers, i.e. the target audience for this paper, are most interested in. I doubt robustness researchers would pay attention to this work if the analysis is only done on CIFAR10. In general, the used models are very far from state-of-the-art models on CIFAR10. Checking the CIFAR10 leaderboard (https://paperswithcode.com/sota/image-classification-on-cifar-10), models with a clean accuracy of 94.4% are around place 104 in the leaderboard. The VGG net has a top-1 accuracy of 93.4%. Given that CIFAR10 is a very small dataset and thus, does not impose huge computation requirements, I would suggest using a state-of-the-art model. Even larger state-of-the-art CIFAR10 models can still be run on one GPU with a reasonable computation time.
>
> Thank you for the suggestion to explore larger datasets (a limitation we acknowledged on line 345). Based on your suggestion, we produced results on CIFAR-100-C, where we also set a SOTA performance level (by an even larger margin). A table containing these results is provided in the joint response (Table B). We will update our manuscript to suggest ImageNet specifically as a topic for future work, but we believe that the demonstrated strengthening of our results on a more complex dataset combined with our theoretical support will encourage researchers to pay attention to CARDs. Additionally, please see our main response to your review, in which we address the fact that we are also interested in robustness and compactness (not just accuracy). Briefly, though, for a fair comparison one needs to look into methods that simultaneously achieve high accuracy, robustness and compression. As mentioned in the motivation of our paper (L51 - 53), past work in achieving this has been negative, i.e., no such method exists and we report the first positive result in this paper.  The closest relevant leaderboard would be RobustBench [1], where we are competing with dense models. However, we show that some compressed models outperform the dense models in terms of both accuracy and OOD robustness while being significantly smaller in size. Robustness benchmark results for CIFAR-10/-C and CIFAR-100/-C are found on RobustBench [1]. Our LRR CARD-Decks outperform CIFAR-10-C and CIFAR-100-C results in both clean and robust accuracy *at a fraction of the memory requirements*. Tables summarizing CIFAR-10-C and CIFAR-100-C results are provided in the joint response.

---

> ### Author Response · Authors · 2021-08-10
> **Our focus is on simultaneously achieving compactness, accuracy, and robustness (further we achieve new SOTA for models simultaneously targeting accuracy and robustness) 1/3**
>
> We appreciate the effort you spent reviewing and your valuable contributions to our work. Respectfully, your review contains critical misunderstandings of our key findings and their significance to the relationship between model compression, accuracy, and robustness. Hence, we will begin our response by clarifying the context and significance of our work. If you have remaining concerns after reading our response we are happy to discuss our work further.
>
> First, prior results in the literature showed that pruning harms robustness [6,7], and our work’s main result is the use of pruning to produce the first empirical confirmation of the existence of models that are simultaneously compact, accurate, and robust (with respect to the dense baseline). The implications of this work are significant as accurate and robust models that are compact are ideal for ML in edge/resource-constrained devices.
>
> Since our stated, primary objective is to determine the existence of compact, accurate, and robust networks, our results should be measured with respect to performing well on these *three metrics* simultaneously. Thus, the closest representative benchmark to compare our results against is RobustBench (https://robustbench.github.io/), which reports accuracy, robustness, and architecture. For example, when critiquing clean/robust accuracy gains as minor, your review ignores the compression metric;  i.e., any such improvements are often achieved in the presence of up to 95% sparsity. In the case of edgepopup/biprop, additional compression is achieved through weight binarization (noted on L143 - 145). Combined pruning and weight binarization can yield memory compression up to ~600x.
>
> Despite your acknowledgement of the trend in the literature suggesting that bigger models provide greater robustness, your review takes issue with the size of any benefits afforded by LRR/LTH relative to the dense baseline or SOTA robustness. Given prior work, the fact that certain pruning strategies do not harm robustness is surprising and our results go further to show that pruning actually produces benefits that are complementary with existing techniques for improving model robustness. These efforts produced new SOTA robustness results on CIFAR-10-C and, more recently, CIFAR-100-C (provided in the table in the joint response for simplicity). Such benefits of LRR/LTH pruned models, which were demonstrated to consistently match or exceed the dense baseline models in terms of accuracy and robustness even when pruned to 95% sparsity, do not appear in your review. Note that these findings are stated throughout the main body of the paper (e.g., L142-143, L276-278), illustrated in Figure 1, provided in tables (Table 1 and 2), and expanded upon with further experimental results using additional architectures (Appendix).
>
> Lastly, we have theoretical and empirical support for our CARD hypothesis which proposes the existence of compact networks with comparable (or higher) accuracy and robustness than the same dense network when it is trained without compression provided that there is a sufficient amount of overparameterization. To try to quantify the level of overparameterization required, our theoretical results provide bounds on the size of a randomly weighted network, say g, that contains a sparse subnetwork approximating some target network, say f, by relating the depth and layer widths of g to those of f. This relationship is from a functional perspective so it is independent from the dataset used when training the model. To provide some context, the bounds on the size of g relative to f ensuring the existence of CARDs are the same as the bounds required to ensure the existence of Strong Lottery Tickets (i.e., compact and accurate networks) established by either of the two spotlight papers from NeurIPS 2020 [2,3]. We also leveraged this work to provide similar assurances for CARD-Deck ensembles (CARD-Deck approximation theorem in the appendix). These results support the CARD hypothesis as they indicate the level of overparameterization required to identify CARD networks. Empirically, we were able to take g = f when searching for subnetworks which emphasizes that the result provides an upper bound on level of overparameterization.
>
> We have addressed the main points from your review below. Please also note our comments to all reviewers in the joint response above. We hope that you will re-evaluate your views based on these facts and revise your final score. We will be happy to answer any additional questions you may have.
>
> &nbsp;
>
> &nbsp;
>
>
> >The authors oversell their results in all performed experiments. They speak of “drastic” gains, “dramatically more robust” models etc. Looking at the results, the robustness gains are on the order of 6-8 percent points for the smaller models and on the order of 1-2 percent points for the larger and thus, more interesting models. Given that their findings do not produce significant gains for their best model, I doubt that this analysis would actually scale to ImageNet. Since their small models with the larger gains have around 88% clean accuracy on CIFAR10, I do not find an analysis on those models particularly interesting, because the clean result is very far from the state of the art on CIFAR10, and the ultimate goal of the authors is to show a new state of the art on CIFAR10. Thus, they maybe should focus on using better models from the beginning.
>
> First, our results should be measured with respect to 3 different metrics: accuracy, robustness, and compression. When pruned to 95% sparsity, LRR demonstrates gains to accuracy and robustness on the order of 1 - 2 percentage points on larger models. Next, to clarify, we have never mentioned setting a new state-of-the-art (SOTA) on CIFAR-10 as our main goal (please see L355). This appears to be a misinterpretation of the motivating question **Q2** on L44. The “performance of SOTA robust-DNNs” we are referring to in **Q2** are tracked and benchmarked as *Common Corruptions* on RobustBench [1] which takes into account clean accuracy and OOD robustness of models. We can clarify this in the final version of the paper. With respect to the intended interpretation of **Q2**, our work answers this question in the affirmative. Specifically, we set a new SOTA on CIFAR-10-C when using pruning as a complementary strategy with existing data augmentation schemes and our CARD-Deck ensembling strategy. Importantly, and based on your remark, we have further verified that the trends observed for CIFAR-10-C hold on CIFAR-100-C, where we also set a new state of the art (5.5% over current SOTA in RobustBench) by using pruning together with SOTA data augmentation schemes designed for OOD robustness and our CARD-Deck strategy. Tables are provided with the best performing CIFAR-10-C and CIFAR-100-C results as well as the previous SOTA results from RobustBench [1] in the joint response (for easy direct comparison). Finally, we agree with your comment on “drastic” and “dramatically” and will simplify the language in these statements in the final version of the paper.
>
>
> >Similar to the overselling of results, the authors seemed to have an “agenda”, a “hypothesis” of what they wanted to show. Irrespective of experimental evidence, they still claimed to see the desired outcome in the experiment, even if the experiment showed the exact opposite, which is a very dangerous and unscientific practice.
>
> We strongly disagree. Our responses below address your individual points. In summary, we provided 1) empirical evidence (Section 2) and 2) theoretical results (Section H.1 in appendix) supporting the CARD hypothesis. For example, the LRR pruned models presented in Figure 1 consistently achieve comparable or better accuracy and robustness than the dense baseline models, even when pruned to 95% sparsity. Furthermore, we comprehensively showed that suitable pruning schemes are complementary with existing OOD robustness strategies -- e.g., use of different model architectures and pruning schemes (Section 2), increasing the model size (Section 4 and Tables 14-16 in appendix), and use of SOTA data augmentation (Section 4).
>
> Our theoretical results provide bounds on the size of a randomly weighted network, say g, that contains a sparse subnetwork approximating some target network, say f, by relating the depth and layer widths of g to those of f. This relationship is from a functional perspective so it is independent from the dataset used when training the model. To provide some context, the bounds on the size of g relative to f ensuring the existence of CARDs is the same as the bounds required to ensure the existence of Strong Lottery Tickets (i.e., compact and accurate networks) established by either of the two spotlight papers from NeurIPS 2020 [2,3]. We also leveraged this work to provide similar assurances for CARD-Deck ensembles (CARD-Deck approximation theorem in the appendix). These results support the CARD hypothesis as they indicate the level of overparameterization required to identify CARD networks. Empirically, we were able to take g = f when searching for subnetworks which emphasizes that the result provides an upper bound on level of overparameterization. We would be glad to move more theoretical details into the main body of the paper (using the additional page) to improve the clarity of and support for the hypothesis. Our empirical work also highlights that the method used to introduce sparsity plays a role in the accuracy and robustness of the model. The hypothesis does not state that a specific pruning strategy will yield comparable accuracy and robustness, rather, that such CARDs can be found. Of relevance, Learning-Rate Rewinding (LRR) was consistently able to outperform the dense baseline in terms of clean and robust accuracy even at 95% sparsity, thereby providing empirical support for the hypothesis.

---

> ### Author Response · Authors · 2021-08-23
> **Encouragement to continue discussion before author response period closes**
>
> As the discussion period for authors will be closing soon, we would be grateful if you could let us know if your position has/hasn’t changed based on our discussion so that we can address any remaining concerns in the next version of our paper. In our most recent response, we provided supplemental quantitative data to address your comments on heatmaps. We stand by our work on compact, accurate, and robust deep learning and assure you that your response and the time spent crafting it will be appreciated by us.

---

### Official Review · Reviewer_eMfa · 2021-07-17

**Rating:** 7
**Confidence:** 3

**Summary:**

This paper explores the possibility and practibility to build compact, accurate, and robust deep neural networks (CARDs). The authors first conduct extensive empirical studies to verify the existence of CARDs. Meanwhile, to better understand CARDs, authors further analyze the Fourier sensitivity in the spectral domain. Based on the empirical observations, a domain-adaptive ensembling approach (CARDs-Deck) is proposed as a practical algorithm to utilize the CARDs models, which is able to achieve new SOTA results for robustness and accuracy on CIFAR-10. The paper is quite informative and solid throughout the text, and the claims are well supported by empirical observations. There are also theoretical analysis that justifies CARDs in depth.

**Limitations And Societal Impact:**

Yes the authors have listed the limitations and potential societal impact.

**Main Review:**


Strengths:

* The paper explores an important problem with promising conclusions: the compact, accurate and robust deep neural networks do empirically exist.

* The authors not only answer the existence of CARDs, but also propose a practical approach to utilize CARDs to improve the accuracy and OOD robustness, which establish new SOTA performance with smaller model sizes on CIFAR-10-C.

* The methodologies involved in both empirical studies (e.g., Fourier sensitivity analysis) and the practical algorithm (domain-adaptive ensembling) are inspiring and novel.


Weakness:

* While extensive experiments and analysis are conducted on CIFAR-10 and CIFAR-10-C, it remains unknown if CARD still exists on large scale datasets such as ImageNet and ImageNet-c.

* It seems the theoretical results (e.g., Theorem 1 and Theorem 3) are less related to pruning, but are general to analyze the approximation and OOD robustness. It may be inspiring to show in what way the compactness (sparsity of network parameters) affect the CARD approximation or the upper bound of OOD robustness.


Detailed comments:

* How about the standard deviation of different methods in Figure 1 and 2? Some lines are highly over-lapped, which could be hard to draw firm conclusions. For example, it is not very clear that layerwise pruned models consistently achieve the maximum accuracy as mentioned in L157.

* The improvement of domain-adaptive CARD-DECKs seems to be incremental, especially when more models are present, according to Table 2.

* In Table 1, there is mis-alignment of the header. Meanwhile, it is kind of misleading that baseline models have the same performance and memory under different pruning ratios in both Table 1 and Table 2.

* The authors can make the paper more self-contained when an additional page is allowed in the revision, especially for theoretical analysis Section 5.


**Time Spent Reviewing:**

5

---

> ### Author Response · Authors · 2021-08-10
> **Domain-adaptive CARD-Decks ~2x faster than domain-agnostic CARD-Decks**
>
> Please also note our comments to all reviewers in the joint response above.
>
> >While extensive experiments and analysis are conducted on CIFAR-10 and CIFAR-10-C, it remains unknown if CARD still exists on large scale datasets such as ImageNet and ImageNet-c.
>
> To overcome this limitation, we have obtained new results on CIFAR-100-C. Some results from these experiments are included in a table in the joint response. In short, we note that these LRR CARD-Decks also set a new SOTA on CIFAR-100-C improving over the existing SOTA (improves robustness by ~5.5 percentage points while improving clean accuracy ~1.5 percentage points). Furthermore, the theoretical results pertaining to the existence of CARDs in our paper which hold regardless of the dataset used.
>
>
> >It seems the theoretical results (e.g., Theorem 1 and Theorem 3) are less related to pruning, but are general to analyze the approximation and OOD robustness. It may be inspiring to show in what way the compactness (sparsity of network parameters) affect the CARD approximation or the upper bound of OOD robustness.
>
>
> We will clarify that theoretically understanding how exactly a given pruning approach improves robustness is a problem that we leave for future work. Note, though, that we took some first steps towards understanding  robustness differences among pruning methods by visualizing how each method impacts the model accuracy when varying perturbations are added to the test data. The associated heatmaps demonstrate a previously unknown phenomenon: models pruned using different strategies respond differently to perturbed test data based on the frequency and orientation of the perturbation. Hence, answering why pruning can bias the model (positively or negatively) towards different perturbations (thereby improving or degrading the OOD robustness) is a non-trivial problem since, as we have demonstrated in our work, the bias is dependent on how the sparsity was introduced into the model. Thus, our work provides this valuable insight when seeking an answer as to "why" pruning can be helpful for robustness: In addition to the need for overparameterization, the method by which sparsity is introduced plays a role in whether it helps or hurts the robustness. We believe that finding an answer to the question of “why” requires deeper consideration and effort and, as such, should be the subject of future work. For some additional context, to the best of our knowledge, there is no existing theoretical work establishing why methods such as weight rewinding (denoted LTH in our paper) or learning rate rewinding (denoted LRR in our paper) are even able to find Lottery Tickets (compact and accurate models) in practice. Further, we hope that these positive findings will ignite interest in the community to theoretically study why the choice of pruning demonstrates this effect.
>
>
> >How about the standard deviation of different methods in Figure 1 and 2? Some lines are highly over-lapped, which could be hard to draw firm conclusions. For example, it is not very clear that layerwise pruned models consistently achieve the maximum accuracy as mentioned in L157.
>
> Thank you for mentioning this point. We can update these figures in the final version of the paper for clarity.
>
>
> >The improvement of domain-adaptive CARD-DECKs seems to be incremental, especially when more models are present, according to Table 2.
>
> The domain-adaptive CARD-Decks only require evaluating a subset of the CARDs in the deck at test time while the domain-agnostic CARD-Decks require evaluating all of the CARDs in the deck. Specifically, for the CARD-Decks presented in this paper **the adaptive CARD-Decks are ~2x faster than the agnostic CARD-Decks** as only half of the CARDs need to be evaluated for each batch of test data. As CARDs are of great interest for resource constrained devices, this reveals that the same (or greater) gains can be achieved by evaluating fewer CARDs at inference. Keep in mind that we are not only interested in model accuracy and robustness but also compactness (and the benefits it provides such as efficiency and energy usage).
>
>
> >In Table 1, there is mis-alignment of the header.
>
> Thank you for pointing out the mis-alignment issue in Table 1. This was caused by a typo in the LaTeX table and will be corrected in the final version of the paper.
>
>
> >Meanwhile, it is kind of misleading that baseline models have the same performance and memory under different pruning ratios in both Table 1 and Table 2.
>
> The baseline models in both Tables 1 and 2 are *not pruned* which is why the performance and memory is persistent across all pruning ratios. We tried to indicate this by including the keyword “Dense” below the header “Baseline” in both tables. We can consider alternate formattings for the table to avoid potential confusion in the final version of the paper.
>
>
> >The authors can make the paper more self-contained when an additional page is allowed in the revision, especially for theoretical analysis Section 5.
>
> Thank you for this suggestion. If accepted, we will use an additional page to make our paper more self-contained.
>
> &nbsp;
>
> Thank you for your time and effort spent reviewing our work. We are excited that you found aspects of our work to be inspiring and novel and we hope that you find the additional results we have provided on CIFAR-100-C (in the joint response) meaningful and exciting as well.
>
> &nbsp;
>
> *References*:
>
> [1] Hooker et. al., *What do compressed deep neural networks forget?*, 2020
>
> [2] Liebenwein et. al., *Lost in Pruning: The Effects of Pruning Neural Networks beyond Test Accuracy*, 2021

---

### Official Review · Reviewer_erM7 · 2021-07-20

**Rating:** 6
**Confidence:** 4

**Summary:**

This paper performs an empirical study on the OOD robustness of compact models and shows the existence of compact, accurate, and robust DNNs (CARDs). It gives a frequency-domain analysis with the Fourier sensitivity method to explain why different pruning methods lead to compact models with varying OOD robustness. Then, the paper proposes CARD-Deck, a domain-adaptive test-time ensembling approach, to dynamically choose an appropriate CARD for each test sample using a spectral-similarity metric and presents some experimental results with CARD-Decks. Finally, it provides some theoretical analysis to support the existence of CARD and the ability to find a CARD with CARD-Deck.

**Limitations And Societal Impact:**

Yes

**Main Review:**

**Originality**:

The paper tries to show that compact, accurate, and robust deep neural networks (CARDs) exist and to develop efficient algorithms to design CARDs that give comparable or better OOD robustness compared to the SOTA dense DNNs. These problems are important and it is novel that the paper points out compression can improve OOD robustness if done properly. Also, this work uses the existing Fourier sensitivity method to study the OOD robustness with different pruning approaches, which is valuable.

**Quality**:

1. The first part of the paper (sections 2 & 3), which discusses the observations on model OOD robustness with different pruning approaches and the Fourier sensitivity studies, is sound and the presented results support most of the claims in the paper.

    * Here is a question regarding FT and GMP results in Figure 1 and Figure 3: in Figure 1, compared to the dense baseline, FT and GMP have higher Top-1 accuracy on CIFAR-10 and lower Top-1 accuracy on CIFAR-10-C. However, in Figure 3, the center of the heatmaps for FT and GMP indicates lower accuracy on clean data and higher robustness to the OOD data following the logic presented in lines 186-191. The conclusions given by the two figures are contradictory. How to explain this?

2.  The second part (section 4) talking about CARD-Deck is very confusing. I have several questions regarding it:
    * What is the difference between CARD-Decks of size 2, 4, and 6?
    * Given the first part of the paper talking about different pruning approaches leads to different OOD robustness, I expected to see a CARD-Deck containing models compressed with different pruning approaches and adaptively selecting models with an approach with the highest OOD robustness. However, according to the results in Table 2, it seems that a CARD-Deck only containing models trained with 2 augmentation schemes and compressed with a single pruning approach. Is there any reason for it? A CARD-Deck containing models compressed with only a single pruning approach is less attractive.
    * It is not clear how CARD-Deck achieves the clean and robust accuracies presented in Table 2. In Table 1, the highest clean accuracy and robust accuracy with LRR is 96.1 and 89.8 given by Augmix. Why with CARD-Deck, LRR can achieve a clean accuracy of 96.6 and robust accuracy of 92? It may be worth spending more space to explain the experiment setups and results.
    * Given the results in Table 2, the improvement brought by Adaptive CARD-Deck is not significant compared to the Agnostic counterparts. Therefore, what is the benefit of CARD-Deck? It would be better to clearly point it out in the paper.

**Clarity**:

The paper is not well organized, and the presentation of the second half (starting from section 4) needs improvement. The description of experimental results is too brief so that it is hard to tell what is the benefit brought by CARD-Deck. The theoretical justifications are also vague. It is not clear what insight or guidance they can provide. I don’t see a strong connection between the CARD approximation theorem and the empirical observations in the paper. It cannot guarantee the existence of CARDs. Also, it cannot explain why “lottery ticket-style” pruning approaches are able to improve the OOD robustness over dense models. Section 5.2 is also unclear. I would recommend putting some formal analysis to the main paper instead of leaving everything in the appendix.

**Significance**:

The idea of studying the existence of CARDs and comparing how different pruning methods affect the OOD robustness of compressed models are valuable and advance our understanding of the connection between model compactness, pruning methods, and OOD robustness. Also, the idea of selecting CARD given the spectral similarity metric is likely to be useful.

**Overall recommendation**:

My overall score for this paper is 4. As stated above, the paper asks important questions and provides valuable observations. However, the presentation needs improvement; the experiments need better design and explanation, and it is better to make the theoretical analysis more relevant to the main ideas of the paper.


**Updates to the Review post rebuttal**

I appreciate the authors' detailed responses to all the reviews. These responses addressed most of my questions and concerns. I have raised my rating.

**Time Spent Reviewing:**

5

---

> ### Author Response · Authors · 2021-08-10
> **Moving theoretical details from appendix to main body will enhance clarity 2/2**
>
> >The theoretical justifications are also vague. It is not clear what insight or guidance they can provide. I don’t see a strong connection between the CARD approximation theorem and the empirical observations in the paper. It cannot guarantee the existence of CARDs.
>
> We respectfully disagree. To provide some context, the level of assurance and theoretical justification for the existence of CARDs is the same level of assurance and theoretical justification for the existence of Strong Lottery Tickets established by either of the two spotlight papers from NeurIPS 2020 [2,3]. We also leveraged this work to provide similar assurances for CARD-Deck ensembles (CARD-Deck approximation theorem in the appendix). We would be glad to move more theoretical details into the main body of the paper (using the additional page) to improve the clarity of the theoretical section.
>
>
> >I don’t see a strong connection between the CARD approximation theorem and the empirical observations in the paper... Also, it cannot explain why “lottery ticket-style” pruning approaches are able to improve the OOD robustness over dense models.
>
> Prior results in the literature showed that pruning harms robustness [4,6], but *our theory highlights that pruning a high-performing network need not harm robustness*, supporting the possibility of CARDs (lines 283-285). It’s true that our empirical results went a step further and showed that “lottery ticket-style” approaches in particular improve robustness relative to dense models, but theory showing why particular pruning approaches are helpful would be difficult to develop, as the analysis would involve the optimization+compression process rather than simply the level of compression. While such theory does not exist yet, then, note that we took a step toward illuminating pruning method differences via our Fourier sensitivity analysis that demonstrated the criticality of the pruning approach to a model’s response to general perturbations, not just those encountered in a particular OOD robustness problem (like CIFAR-10-C and CIFAR-100-C). To make clearer these connections between our theoretical and empirical findings, we will revise our discussion of these topics in the manuscript.
>
>
>
> >Section 5.2 is also unclear. I would recommend putting some formal analysis to the main paper instead of leaving everything in the appendix.
>
> We can move details from the appendix into the main body of the paper in the final version (as we will have an additional page) to improve the clarity. We summarize key insights from Section 5.2 next. Theorem 3 (formally stated in the appendix) provides some key insights into the OOD robustness of classifiers trained on augmented datasets. First, unlike the generalization gap, the OOD-shift does not converge to zero when more augmentation data is provided from the augmented data distributions. This imposes a fundamental limit on the OOD robustness in terms of the distance between augmented train data distribution and corrupted test data distribution. Second, having diverse augmentations is critical to improving the OOD robustness. Also, it highlights that existing solutions trained with a single augmentation scheme might just be getting lucky or overfitting to the corrupted test data. Finally, the domain-adaptive CARD-Deck with a suitable gating function is provably better than using a single classifier because it can achieve the minimum conditional Wasserstein distance (or best achievable OOD robustness) over given augmentation-corruption pairs.
>
> &nbsp;
>
> We thank you for your feedback on our work and for taking the time to review it. Please let us know if you have any additional questions or would like further clarification. If you are confident that we have addressed all of your remarks, we hope that you will update your score and champion our paper for acceptance.
>
> &nbsp;
>
>
> *References*:
>
> [1] Fort et. al., *Deep Ensembles: A Loss Landscape Perspective*, 2020
>
> [2] Orseau et. al., *Logarithmic Pruning is All You Need*, NeurIPS 2020
>
> [3] Pensia et. al., *Optimal Lottery Tickets via Subset Sum: Logarithmic Overparameterization is Sufficient*, NeurIPS 2020
>
> [4] Hooker et. al., *What do compressed deep neural networks forget?*, 2020
>
> [5] Yin et. al., *A Fourier Perspective on Model Robustness in Computer Vision*, NeurIPS 2019
>
> [6] Liebenwein et. al., *Lost in Pruning: The Effects of Pruning Neural Networks beyond Test Accuracy*, 2021

---

> ### Author Response · Authors · 2021-08-10
> **Moving theoretical details from appendix to main body will enhance clarity 1/2**
>
> Please also note our comments to all reviewers in the joint response above.
>
> >Here is a question regarding FT and GMP results in Figure 1 and Figure 3: in Figure 1, compared to the dense baseline, FT and GMP have higher Top-1 accuracy on CIFAR-10 and lower Top-1 accuracy on CIFAR-10-C. However, in Figure 3, the center of the heatmaps for FT and GMP indicates lower accuracy on clean data and higher robustness to the OOD data following the logic presented in lines 186-191. The conclusions given by the two figures are contradictory. How to explain this?
>
> Please note that we did not state there is a one-to-one mapping between heatmaps and CIFAR-10-C performance, and we will clarify their relationship in the revision. While it is true that the center pixel in the heatmap most closely corresponds to the test error of the model on CIFAR-10, it is difficult to have an exact mapping between the error rate heatmap generated by introducing noise in the form of Fourier bases into the test data and the error rate on CIFAR-10-C. This is because the noise for each image, corruption type, and severity in CIFAR-10-C is composed of different Fourier bases elements. To clarify this point further, a visualization showing the average representation of the noise for different CIFAR-10-C corruption types (at severity level 3) in terms of Fourier bases elements can be found in Figure 2 of [5]. The point of Fourier heatmap analysis [5] is not to show one-to-one correspondence but to show that a high level of correlation exists between error on Fourier bases elements and error on real-world OOD data. Of course, one can find corner cases where this won’t hold but that does not invalidate the usefulness of Fourier heatmap analysis. In fact, both our empirical (Section 4) and theoretical (Section H.2 in appendix) results corroborate our findings from Fourier heatmaps.
>
>
> >What is the difference between CARD-Decks of size 2, 4, and 6?
>
> The size of the CARD-Deck indicates the number of CARDs used in the ensemble (based on the CARD-Deck ensemble description in L240 - L243). On L266 we specify how many CARDs in the CARD-Deck ensemble were trained using the different augmentation schemes. To clarify, for these experiments a *2n* card deck ensembles *n* CARDs trained using the Augmix data augmentation scheme and *n* CARDs trained using the Gaussian data augmentation scheme. A domain-agnostic CARD-Deck of size *2n* averages the predictions of all *2n* CARDs. A domain-adaptive CARD-Deck of size *2n* averages the predictions of the *n* CARDs trained using the augmentation scheme most similar (identified using the spectral similarity metric) at test time. We can revise this in the final version of the paper to ensure it is clear.
>
>
> >Given the first part of the paper talking about different pruning approaches leads to different OOD robustness, I expected to see a CARD-Deck containing models compressed with different pruning approaches and adaptively selecting models with an approach with the highest OOD robustness. However, according to the results in Table 2, it seems that a CARD-Deck only containing models trained with 2 augmentation schemes and compressed with a single pruning approach. Is there any reason for it? A CARD-Deck containing models compressed with only a single pruning approach is less attractive.
>
> In Tables 5 - 7 (contained in the supplementary materials), we provide a breakdown of the performance of CARDs trained using current SOTA data augmentation schemes together with the different pruning techniques considered in this work. These tables show that CARDs identified using learning rate rewinding (LRR) achieve the best performance in 14 of the 15 different corruption types contained in CIFAR-10-C when pruned to 80/90% sparsity and achieve the best performance in all 15 corruption types when pruned to 95% sparsity. Hence, in the presence of SOTA data augmentation schemes for improving OOD robustness, selecting the best pruning scheme based on the highest OOD robustness reduces to using LRR pruned models. However, these tables also indicate that the different data augmentation schemes used during training and pruning are able to yield better performance depending on the type of corruption. In light of this, improved performance can be achieved by designing a gating mechanism that selects the model based on the augmentation scheme used when training the model which is exactly what the domain-adaptive CARD-Deck leverages. We agree that our work opens up an opportunity to explore other variants of ensembles as well, such as the one pointed out by the reviewer. This will be mentioned as a worthwhile direction to explore in our future work section.
>
>
> >It is not clear how CARD-Deck achieves the clean and robust accuracies presented in Table 2. In Table 1, the highest clean accuracy and robust accuracy with LRR is 96.1 and 89.8 given by Augmix. Why with CARD-Deck, LRR can achieve a clean accuracy of 96.6 and robust accuracy of 92? It may be worth spending more space to explain the experiment setups and results.
>
> The increased performance can be attributed to prediction averaging [1] and, in the domain-adaptive CARD-Decks, also by the gating function for selecting which CARDs to use. A domain-agnostic CARD-Deck of size 2n averages the predictions of all 2n CARDs. A domain-adaptive CARD-Deck of size 2n averages the predictions of the n CARDs trained using the augmentation scheme most similar (identified using the spectral similarity metric) to the images received at test time. This is outlined in the paragraph starting on L247.
>
>
> >Given the results in Table 2, the improvement brought by Adaptive CARD-Deck is not significant compared to the Agnostic counterparts. Therefore, what is the benefit of CARD-Deck? It would be better to clearly point it out in the paper.
>
> The domain-adaptive CARD-Decks only require evaluating a subset of the CARDs in the deck at test time while the domain-agnostic CARD-Decks require evaluating all of the CARDs in the deck. Specifically, for the CARD-Decks presented in this paper **the adaptive CARD-Decks are ~2x faster than the agnostic CARD-Decks** as only half of the CARDs need to be evaluated for each batch of test data. Additionally, note that in the case of the 2 CARD-Decks there is a larger margin between the accuracy of adaptive and agnostic CARD-Decks. As CARDs are of great interest for resource constrained devices, it is notable that our results show larger accuracy/robustness gains can be achieved with small CARD-Decks/ensembles. We are not only interested in model accuracy and robustness but also compactness (and benefits it provides such as efficiency and energy usage). We will summarize these benefits in the final version of the paper.
>
>
> >Clarity: The paper is not well organized, and the presentation of the second half (starting from section 4) needs improvement.
>
> We agree that the presentation of Sections 4 and 5 could be improved. We will clarify these sections by including details from the appendix. If our submission is accepted, we will be allowed an additional content page for the camera-ready version which will be used to make the paper more self-contained. Due to space constraints in the reviewed version of the paper, we  hope that the reviewer will not consider organizational issues as a weakness.
>
>
> >The description of experimental results is too brief so that it is hard to tell what is the benefit brought by CARD-Deck.
>
> The benefits gained in accuracy, robustness, and memory for CARD-Decks compared to dense models can be inferred from Tables 1 and 2. Table 1 provides clean and robust accuracy as well as memory requirements for an individual CARD network pruned using different strategies as well as the same statistics for dense baseline models. Table 2 provides these statistics for domain-agnostic and domain-adaptive CARD-Decks and demonstrates that clean and robust accuracies can be improved by ensembling sparse models. A summary of benefits is listed here (and will be added to the paper for clarity):
>
> * LRR CARD-Decks containing CARDs pruned to 95% sparsity set a new SOTA performance on CIFAR-10-C in terms of accuracy and robustness at a fraction of the memory usage.
>
> * In CARD-Decks containing CARDs pruned to 90 or 95% sparsity, the total memory is still well below the memory of a single dense network (up to ~105x reduction in memory while maintaining comparable accuracy and robustness when using binary-weight CARDs from edgepopup or biprop).
>
> * For the CARD-Decks presented in this paper **the adaptive CARD-Decks are ~2x faster than the agnostic CARD-Decks** as only half of the CARDs need to be used for inference for each batch of test data.
>
> Further, we have obtained additional results on CIFAR-100-C. The best results are provided in a tables in the joint response. We note that these LRR CARD-Decks also set a new SOTA on CIFAR-100-C, improving over the existing SOTA (improving robustness by ~5.5 percentage points while improving clean accuracy ~1.5 percentage points).

---

> ### Author Response · Authors · 2021-08-23
> **Encouragement to respond before author response period closes**
>
> As the discussion period for authors will be closing soon, we would be grateful if you could let us know if your position has/hasn’t changed based on our response to your review so that we can address any remaining concerns in the next version of our paper. For your convenience, we made our response to your review self-contained by providing in-line responses to excerpts from your original review. We stand by our work on compact, accurate, and robust deep learning and assure you that your response and the time spent crafting it will be appreciated by us.

---

### Official Review · Reviewer_JSRv · 2021-07-20

**Rating:** 4
**Confidence:** 2

**Summary:**

This paper studies on whether pruning is able to improve the model robustness. This paper first show that with a proper pruning algorithm, the robustness can be achieved. Then a spectrum analysis is proposed to visualize what kind of heat map is suitable for robustness. Thirdly, a test-time augmentation approached is proposed to further improve the robustness.

**Main Review:**

The direction this paper studied is interesting while there are several important points regarding the significancy of the result needs discussions.

1. To draw the main conclusion on what kind of pruning algorithm can improve the robustness, only one kind of bench mark (Cifar-10 on label corruption) is tested. This makes the conclusion less convincing. How about adversarial robustness? OOD detection? Calibration? The robustness so far is quite limited and the dataset test is also quite limited.

2. I appreciate the analysis through the heat map, which is very interesting. However, despite that it is a successful visualization, it still didn't give answer to the most important question: why pruning is helpful for robustness. For example, if we train the network use some standard approach for robustness, is the heat map of pruning close to the network using robust training? If so, why this happens. I understand it is hard to really give a very comprehensive explanation but to me, the current analysis is more close to listing the observations.

3. Although pruning can improve the robustness, but more important experiment is to show, combined with the training algorithm that improves the robustness, whether pruning can further enhance the robustness above that. To seriously achieve the robustness, people will use non-standard training approach. How's the pruned network's robustness performance compared with that? How much improve you can further achieve above that?

**Time Spent Reviewing:**

2.5h

---

> ### Author Response · Authors · 2021-08-10
> **Network compression complementary with OOD robustness methods (sets new SOTA on CIFAR-10-C and CIFAR-100-C)**
>
> Please also note our comments to all reviewers in the joint response above.
>
> > To draw the main conclusion on what kind of pruning algorithm can improve the robustness, only one kind of bench mark (Cifar-10 on label corruption) is tested. This makes the conclusion less convincing.
>
> We wish to clarify that the main conclusion of our paper was that *pruning can improve robustness*, and our benchmark (CIFAR-10-C performance) was chosen so that we would be using the same robustness benchmark (or set of test-time image corruptions) as works which found that *pruning harms robustness* [3,4]. To further support this conclusion, our theoretical analysis showed that sufficiently overparameterized networks can be compressed without losing their accuracy/robustness, supporting the existence of compact+accurate+robust DNNs (CARDs). However, we include here and will update the manuscript with results showing that pruning significantly improves the SOTA robustness on CIFAR-100-C as well (see the tables of results provided in the joint response). Also, note that CIFAR-10-C and CIFAR-100-C are not created by adding label corruption, as each contains 15 different common corruptions at five different severity levels applied to the CIFAR-10/CIFAR-100 test images.
>
>
> >How about adversarial robustness? OOD detection? Calibration? The robustness so far is quite limited and the dataset test is also quite limited.
>
> The focus of this work is on OOD robustness (as emphasized in the paper title). Calibration, detection, etc. are interesting but different problems altogether and are out of the scope of this work.
>
>
> >I appreciate the analysis through the heat map, which is very interesting. However, despite that it is a successful visualization, it still didn't give answer to the most important question: why pruning is helpful for robustness. For example, if we train the network use some standard approach for robustness, is the heat map of pruning close to the network using robust training? If so, why this happen? I understand it is hard to really give a very comprehensive explanation but to me, the current analysis is closer to listing the observations.
>
> We are glad that you found the heatmap analysis interesting. In regards to "why pruning is helpful for robustness", we would first like to emphasize that prior work studying the effects of pruning on OOD robustness were mostly negative [3,4] and our findings are the first to demonstrate that pruning can, in fact, improve OOD robustness. Heatmaps were used as a first step to further investigate and visualize how each pruning method impacts the model accuracy when varying perturbations are added to the test data. The heatmaps demonstrate an additional previously unknown phenomenon: models pruned using different strategies respond differently to perturbed test data based on the frequency and orientation of the perturbation. Hence, answering why pruning can bias the model (positively or negatively) towards different perturbations (thereby improving or degrading the OOD robustness) is a non-trivial problem since, as we have demonstrated in our work, the bias is dependent on how the sparsity was introduced into the model. Thus, our work provides this valuable insight when seeking an answer as to "why" pruning can be helpful for robustness: In addition to the need for overparameterization, the method by which sparsity is introduced plays a role in whether it helps or hurts the robustness. We believe that finding an answer to the question of “why” requires deeper consideration and effort and, as such, should be the subject of future work. For some additional context, to the best of our knowledge, there is no existing theoretical work establishing why methods such as weight rewinding (denoted LTH in our paper) or learning rate rewinding (denoted LRR in our paper) are even able to find Lottery Tickets (compact and accurate models) in practice. Further, we hope that these positive findings will ignite interest in the community to theoretically study why the choice of pruning demonstrates this effect.
>
>
> >Although pruning can improve the robustness, but more important experiment is to show, combined with the training algorithm that improves the robustness, whether pruning can further enhance the robustness above that. To seriously achieve the robustness, people will use non-standard training approach. How's the pruned network's robustness performance compared with that? How much improve you can further achieve above that?
>
> In Section 4, we precisely addressed how the model robustness is affected when using "non-standard training approach[es]" together with pruning. Specifically, we considered pruning techniques together with state-of-the-art (SOTA) methods for improving OOD robustness which show that:
>
> 1. **Pruning is complementary with methods for improving OOD robustness** (i.e. compressed models can also benefit from existing OOD robustness methods)
> 2. **Certain pruning techniques consistently provide additional gains in accuracy and robustness** (even when models are pruned to 95% sparsity)
> 3. **Compressed models set a new SOTA for accuracy and robustness on CIFAR-10-C**.
>
> (Note: Another reviewer pointed out a formatting issue in Table 1 in which all of the values are accidentally shifted to the right by one column. Please take this into account when looking into the results in Table 1.) Further, experiments with CIFAR-100-C demonstrate similar trends hold even when models are pruned to 95% sparsity and result in a new SOTA on CIFAR-100-C. For convenience, we have provided the best results (achieved when using LRR to prune WideResnet-18 to 95%) on CIFAR-10-C and CIFAR-100-C in the tables provided in the joint response. Similar trends hold for remaining pruning schemes as well.
>
> &nbsp;
>
>
> We are glad that you found our work interesting. We hope that our responses have addressed your concerns and that you will update your rating to reflect any added perspective from our response.
>
> &nbsp;
>
> *References*:
>
> [1] Hendrycks et. al., *AugMix: A Simple Data Processing Method to Improve Robustness and Uncertainty*, ICLR 2020
>
> [2] Croce et. al., *RobustBench: A Standardized Adversarial Robustness Benchmark*, 2020
>
> [3] Hooker et. al., *What do compressed deep neural networks forget?*, 2020
>
> [4] Liebenwein et. al., *Lost in Pruning: The Effects of Pruning Neural Networks beyond Test Accuracy*, 2021

---

> ### Author Response · Authors · 2021-08-23
> **Encouragement to respond before author response period closes**
>
> As the discussion period for authors will be closing soon, we would be grateful if you could let us know if your position has/hasn’t changed based on our response to your review so that we can address any remaining concerns in the next version of our paper. For your convenience, we made our response to your review self-contained by providing in-line responses to excerpts from your original review. We stand by our work on compact, accurate, and robust deep learning and assure you that your response and the time spent crafting it will be appreciated by us.

---

### Author Response · Authors · 2021-08-10
**Joint Response**

We thank the reviewers for their detailed, valuable reviews. We are happy to know reviewers found our paper’s focus of *simultaneous compactness and robustness* to be an “important” (eMfa, erM7), “valuable” (erM7), “under-explored” (kbDD), “interesting" (JSRv, kbDD), and "very relevant” (kbDD) problem. Critically, reviewers recognized our accomplishment of the first result to show that pruning significantly helps OOD robustness, calling our empirical demonstration of robustness improvement via compression “novel” (erM7),  “promising” (eMfa), and “establish[ing] new SOTA performance” (eMfa).

Indeed, our paper was motivated by the recent results showing that *pruning harms robustness* [2,3], and our finding that this need not be the case was our emphasis. While reviewers found this claim to the contrary to be “well supported by empirical observations” (eMfa, erM7) and our analyses “sound” (erM7), we underperformed with respect to communicating the significance of some of our results.
1. For instance, some said our Fourier-sensitivity study was “very interesting” (JSRv), “valuable” (erM7), and “inspiring” (eMfa), but we did not sufficiently communicate how these results relate to CIFAR-10-C, and that left some reviewers (JSRv, erM7) with questions regarding the related heatmaps and another reviewer under the impression that this Fourier study was “vague” (kbDD); please see our individual responses for answers to the related questions.
2. Similarly, the description of our ensembling approach of compact models (CARD-Deck) is “too brief” such that it is “hard to tell what is the benefit” (​​erM7), and it made our contribution in this area appear “incremental” (eMfa) relative to the domain-agnostic ensembling approach we also used. As our individual responses detail, there are notable benefits of our approach (such as its 2x forward-pass speed-up relative to the domain-agnostic ensemble).
3. Finally, we showed that our empirical finding that *pruning need not harm robustness* was *theoretically* supported by proving that sparse, compact subnetworks exist inside highly accurate and robust networks, but we did not successfully convey this and left reviewers wondering about the relevance of our theory to our empirical results (erM7, ​​eMfa, kbDD); we address specific questions on this matter inside individual responses.

Accordingly, we have updated the discussion to better communicate the significance of these results (explained in detail in the individual responses), and we agree that we “can make the paper more self-contained when an additional page is allowed in the revision” (eMfa).

Another concern shared by reviewers was uncertainty about whether our results would extend to “large scales” (eMfa, kbDD). While a better discussion of our theoretical results (see #3 above) may alleviate this concern, we have run more experiments that may also help. Specifically, we have found that our approach of combining data augmentation techniques for improving OOD robustness (such as the previously-introduced AugMix method [4]) with our ensembling+pruning method confers a greater improvement relative to the previous SOTA on CIFAR-100-C than we observed on CIFAR-10-C. Please see Table B (below) for a quick summary of these results. Some CIFAR-10-C results are also provided (for convenience) in Table A (below).


For answers to the specific questions raised by each reviewer please see our individual  comments below. Based on these answers and clarifications we believe we have addressed the concerns of each of the reviewers. We are hopeful that reviewers would consider our answers and would increase their ratings and recommend acceptance.
&nbsp;

&nbsp;

&nbsp;

*Table A: Results for CIFAR-10-C*

| |**CIFAR-10 Top 1 Acc.**|**CIFAR-10-C Top 1 Acc.**|**Memory (Mbit)**
-----|:-----:|:-----:|:-----:
Dense (WideResNet-18, Augmix)|95.6|89.3|1429
LRR CARD (WideResNet-18, Augmix, 95% sparse)|96.4|90.8|71.5
LRR CARD-Deck (WideResNet-18, 3 Augmix, 3 Gaussian, 95% sparse)|**96.8**|**92.75**|428.7
Edgepopup CARD-Deck (WideResNet-18, 3 Augmix, 3 Gaussian, 95% sparse)|95.3|90.6|**13.4**
Previous SOTA (ResNet-50, from RobustBench)|94.93|92.17|753


&nbsp;

&nbsp;

&nbsp;


*Table B: Results for CIFAR-100-C*

| |**CIFAR-100 Top 1 Acc.**|**CIFAR-100-C Top 1 Acc.**|**Memory (Mbit)**
-----|:-----:|:-----:|:-----:
Dense (WideResNet-18, Augmix)|77.5|66.1|1433
LRR CARD (WideResNet-18, Augmix, 95.6% sparse)|78.5|67|63
LRR CARD-Deck (WideResNet-18, 3 Augmix, 3 Gaussian, 95.6% sparse)|**80.6**|**71.3**|377.9
Edgepopup CARD-Deck (WideResNet-18, 3 Augmix, 3 Gaussian, 95% sparse)|79.1|69.7|**13.5**
Previous SOTA (ResNeXt-29\_32x4d, from RobustBench)|78.9|65.54|157


&nbsp;

&nbsp;

&nbsp;

*References*:

[1] Croce et. al., *RobustBench: A Standardized Adversarial Robustness Benchmark*, 2020

[2] Hooker et. al., *What do compressed deep neural networks forget?*, 2020

[3] Liebenwein et. al., *Lost in Pruning: The Effects of Pruning Neural Networks beyond Test Accuracy*, 2021

[4] Hendrycks et. al., *AugMix: A Simple Data Processing Method to Improve Robustness and Uncertainty*, ICLR 2020

---

### Decision · Program_Chairs · 2021-09-27

**Decision:**

Accept (Poster)

**Comment:**

This work studies whether or not model pruning can be leveraged to improve the out-of-distribution robustness of image models. The work notes that the benefits of pruning strongly depend on details on how pruning is applied. This observation is valuable in light of past literature which were largely unable to improve robustness via model pruning. Reviewers initially had a number of criticisms of the work. For example the question of why model compression sometimes improves robustness remained unanswered and also the work only provides experiments on variants of CIFAR-10 and CIFAR-100. However, the authors had several extended back and forth discussions with reviewers during the rebuttal period and made several improvements to the work while addressing a number of concerns that reviewers have. Although I am quite hesitant to accept a primarily empirical work with experiments only on CIFAR, I overall agree with reviewers that the provided experiments are quite through and recommend accepting the work.